# Hydrogen Peroxide as a Green Oxidant for the Selective Catalytic Oxidation of Benzylic and Heterocyclic Alcohols in Different Media: An Overview

**Majid M. Heravi [1],\*, Nastaran Ghalavand [1] and Elaheh Hashemi [2]**

[1]  Department of Chemistry, Alzahra University, Vanak, Tehran 1993893973, Iran; ghalavand.nastaran@yahoo.com

[2]  Department of Chemistry, Faculty of Science, Shahid Rajaee Teacher Training University, Tehran 1678815811, Iran; e.hashemi@sru.ac.ir

\*  Correspondence: mmheravi@alzahra.ac.ir or mmh1331@yahoo.com; Tel.: +98-218-804-1344

**Abstract:** Among a plethora of known and established oxidant in organic chemistry, hydrogen peroxide stands in a special position. It is commercially and inexpensively available, highly effective, selective, and more importantly it is compatible with current environmental concerns, dictated by principles of green chemistry. Several chemicals or their intermediates that are important in our daily life such as pharmaceuticals, flavors, fragrances, etc. are products of oxidation of alcohols. In this review, we introduce hydrogen peroxide as an effective, selective, green and privileged oxidant for the catalyzed oxidation of primary and secondary benzylic and heterocyclic alcohols to corresponding carbonyl compounds in different media such as aqueous media, under solvent-free conditions, various organic solvent, and dual-phase system.

**Keywords:** selective catalytic oxidation; benzylic alcohols; heterocyclic alcohols; hydrogen peroxide; green oxidant; aqueous media; solvent-free conditions; organic solvents; dual-phase system

---

## 1. Introduction

Nowadays, energy and environmental crises have resulted in the discovery of green technologies for progressive use in the chemical industry. Exploration of green chemical reaction approaches have especially shaped our daily lives. The use of alternative non-toxic and green solvents, benign reagents and substrates, and the application of new catalytic systems with the same efficiency or even superior efficiency have also reduced the risk of chemical reactions and resulted in safety [1,2].

Oxidation of primary and secondary alcohols to their corresponding aldehydes and ketones, respectively, are considered as a pivotal reaction in organic transformations due to the wide applications of these products as precursors or intermediates for construction of several drugs, fragrances, and vitamins. Because of the low selectivity of some reactions such as oxidation, the resulting byproducts generate waste and pollution. Therefore, selective oxidation of organic compounds as a critical chemical transformation has wide-ranging applications in the chemical, industrial, and biological processes which needs more attention. Traditionally, selective oxidation of alcohols has needed toxic, corrosive, and expensive stoichiometric inorganic oxidants such as Cr(VI), Mn(VII), heptavalent iodine, and DMSO-coupled reagents which create both environmental and economic concerns [3,4]. Due to an urgent and important demand for greener approaches, the use of toxic solvents and reagents such as organic peroxides must be avoided. Thus, green solvents such as water or solvent-free systems, clean oxidants such as $O_2$ or $H_2O_2$, and also recyclable catalysts should be considered in order to comply

with the principles of green chemistry proposed by Anastas and Warner [2]. According to the atom economy principle, established by Trost in 1991, in oxidation reactions, with respect to the total mass of the oxidant, the mass amount of oxygen transferred to the substrate should be optimal [5]. Although molecular oxygen is considered to be an ideal oxidant, its use sometimes is problematic since it needs harsh reaction conditions such as high temperature or pressure and it also shows poor selectivity.

Nowadays, among the common oxidants, hydrogen peroxide ($H_2O_2$) with 47% oxygen content has been found to be a more suitable and practical oxidant. Moreover, with the transfer of one oxygen atom to the substrate in the oxidation reaction, one equivalent of $H_2O$ is practically formed as an ideal and green expected byproduct [6–10]. In addition, due to its good solubility in water and many organic solvents, $H_2O_2$ is a very operative oxidant in liquid phase reactions [9,10]. Furthermore, its safe storage, operation, and transportation of aqueous hydrogen peroxide, along with its commercially availability, justify its wide applications [11,12].

Selective oxidation of aliphatic or aromatic alcohols to the corresponding carbonyl compounds using hydrogen peroxide either in the absence or presence of catalysts in different solvents under neutral, alkaline, or acid conditions have been extensively studied [13–18]. Our research group was also interested in oxidation [19–22], using hydrogen peroxide as an oxidant of choice [23,24] and especially, selective oxidation of alcohols to the corresponding carbonyl compounds [25–27]. We recently reviewed the applications of pyridinium chlorochromate (PCC) as an important selective oxidant for oxidation of primarily and secondary alcohols to their corresponding carbonyl compounds [28]. As a continuation of these interests, herein, we try to highlight the importance and efficacy of hydrogen peroxide as an effective and green oxidant in the catalyzed selective oxidation of primary and secondary benzylic alcohols and primary heterocyclic alcohols to their corresponding carbonyl compounds. This review is also intended to cover recent literature, focusing on new methods of selective catalytic oxidation of benzylic and heterocyclic alcohols using hydrogen peroxide in aqueous phase, a solvent-free system, in various solvents, and dual-phase systems. In addition, we present an overview on the reasonable reaction mechanism and promising catalytic effects of the selective oxidation of alcohols which actually can be considered as dehydrogenation of alcohols.

## 2. Selective Catalytic Oxidation of Benzylic and Heterocyclic Alcohols

### 2.1. Oxidation of Benzylic and Heterocyclic Alcohols in Aqueous Media

Epichlorohydrin (ECH)-modified $Fe_3O_4$ microspheres (**1**) for the selective oxidation of benzyl alcohol to benzaldehyde with $H_2O_2$ has been reported [29,30]. The ECH-derived hydroxyl groups have been shown to be milder and beneficial for the structural stability of $Fe_3O_4$ as compare with corrosive organic acid and iron-chelating additives. With ECH modification, a few surface active sites have been occupied, so the activation energy for $H_2O_2$ on $Fe_3O_4$ microspheres decomposition increased from 50.1 to 116.3 kJ mol$^{-1}$. Therefore, the promotional effect of short-chain saturated alcohols as additives for $Fe_3O_4$-catalyzed decomposition in $H_2O_2$ was established for the first time, and exploited to improve the catalytic performance of $Fe_3O_4$ microspheres for the selective oxidation of benzyl alcohol to corresponding benzaldehyde with $H_2O_2$ in water (Table 1) [31]. The results showed that applications of only $Fe_3O_4$ microspheres converted only 5.7% of $PhCH_2OH$ (Table 1, Entry 1). Although, the catalytic activity of $Fe_3O_4$ microspheres were suggestively improved by the addition of small amounts of short-chain saturated alcohols (0.4 mol% with regards to $H_2O_2$) (Table 1, Entries 2–7), the selectivity for benzaldehyde reduced to 83 to 87%, due to over-oxidation to benzoic acid as a byproduct. The better promotional effect (same dosage, 0.4 mol%) of secondary and tertiary alcohols (41.8% to 48.9% in Table 1, Entries 2–4) on the conversion of benzyl alcohol (13.5% to 22.8% in Table 1, Entries 5–7) are observed as compared with primary alcohols. As shown in Table 1, for each alcohol, the optimal dosage could be different additives to accomplish the highest conversion of benzyl alcohol. Although the dosage of all alcohol additives was fixed (0.4 mol% with regard to hydrogen peroxide), the conversion of $PhCH_2OH$ may not be the optimal value for each alcohol additive.

**Table 1.** Catalytic oxidation of benzyl alcohol to benzaldehyde with $H_2O_2$ in water on $Fe_3O_4$ microspheres **1** with 0.4 mol% of alcohol additives [a].

| Entry | Additive | Conv. (%) | Yield (%) | Sel. (%) |
|-------|----------|-----------|-----------|----------|
| 1 | No additive | 5.7 | 5.6 | 98.8 |
| 2 | Glycerol | 48.9 | 42.6 | 87.2 |
| 3 | Tert-butanol | 45.3 | 38.9 | 86 |
| 4 | Iso-propanol | 41.8 | 34.9 | 83.5 |
| 5 | Methanol | 22.8 | 19.5 | 85.5 |
| 6 | Ethanol | 13.5 | 11.5 | 85.4 |
| 7 | MEG | 17.4 | 14.5 | 83.2 |

[a] Reaction conditions: Substrate (40 mmol and 4.1 mL), short-chain saturated alcohols (0.4 mol% with regards to $H_2O_2$), $Fe_3O_4$ microspheres **1** (0.2 g), $H_2O_2$ (30 wt%, 8 mL), $H_2O$ (8 mL), reflux at 100 °C, 1.5 h.

Recyclable and reusable nanocatalyst ($MgAl_2O_4@SiO_2$–PTA) (**2**) was applied for the selective oxidation of primary and secondary alcohols (Table 2) to the corresponding aldehydes (for example, benzyl alcohol Entry 1) and ketones (for example, 1-phenyl ethanol Entry 7) with hydrogen peroxide as oxidant and water [32]. The oxidation reaction of 1-phenyl ethanol to acetophenone was more efficient than other reactions (98%). The catalyst was successfully used five times without any loss of its high catalytic activity. In benzylic alcohol oxidation, since the replacement was different from electron donating to electron withdrawing (Table 2, Entries 2–8), the yield of products decreased from 94% to 81% [33].

**Table 2.** Synthesis of diversified benzylic alcohol in the presence of $MgAl_2O_4@SiO_2$–PTA **2** [a].

| Entry | Substrate | Product | Yield (%) [b] |
|-------|-----------|---------|---------------|
| 1 | $C_6H_5CH_2OH$ | $C_6H_5CHO$ | 96 |
| 2 | $4\text{-}ClC_6H_4CH_2OH$ | $4\text{-}ClC_6H_4CHO$ | 91 |
| 3 | $4\text{-}NO_2C_6H_4CH_2OH$ | $4\text{-}NO_2C_6H_4CHO$ | 85 |
| 4 | $3\text{-}NO_2C_6H_4CH_2OH$ | $3\text{-}NO_2C_6H_4CHO$ | 81 |
| 5 | $4\text{-}OMeC_6H_4CH_2OH$ | $4\text{-}OMeC_6H_4CHO$ | 93 |
| 6 | $4\text{-}MeC_6H_4CH_2OH$ | $4\text{-}MeC_6H_4CHO$ | 94 |
| 7 | $C_6H_5CH(OH)CH_3$ | $C_6H_5COCH_3$ | 98 |
| 8 | $4\text{-}MeC_6H_4CH(OH)CH_3$ | $4\text{-}MeC_6H_4COCH_3$ | 97 |
| 9 | $4\text{-}ClC_6H_4CH(OH)CH_3$ | $4\text{-}ClC_6H_4COCH_3$ | 95 |
| 10 | $3\text{-}BrC_6H_4CH(OH)CH_3$ | $3\text{-}BrC_6H_4COCH_3$ | 92 |
| 11 [c] | $C_6H_5CH_2OH$ | - | - |

[a] Reaction condition: Substrate (2 mmol), $MgAl_2O_4@SiO_2$–PTA **2** (0.05 g), $H_2O_2$ (30%, 1 mmol), $H_2O$ (3 mL), 90 °C, 1.5 h. [b] Isolated yields. [c] Substrate (2 mmol), without $MgAl_2O_4@SiO2$–PTA 2, $H_2O_2$ (1 mmol), $H_2O$ (3 mL), 90 °C, 4 h.

A heterogeneous organocatalyst "glycoluril" (**3**) was applied for the oxidation of alcohols to the corresponding carbonyls with excellent conversion and selectivity (more than 90%) in an aqueous medium using hydrogen peroxide (Table 3) [34]. The catalyst was successfully recycled for >10 times. Depending on the functional groups substituted on the aromatic rings of benzylic alcohols (Table 3, Entries 1–5), good conversions and selectivities were obtained with different reaction times. In addition, heteroaromatic substrates were successfully converted into the corresponding aldehydes in a short reaction time.

**Table 3.** Substrate scope for the oxidation reaction [a].

| Entry | Substrate | Product | t (h) | Conv. (%) | Sel. (%) |
|-------|-----------|---------|-------|-----------|----------|
| 1 | $C_6H_5CH_2OH$ | $C_6H_5CHO$ | 3 | 99 | 98 |
| 2 | $4\text{-}OHC_6H_4CH_2OH$ | $4\text{-}OHC_6H_4CHO$ | 5 | 93 | 97 |
| 3 | $4\text{-}FC_6H_4CH_2OH$ | $4\text{-}FC_6H_4CHO$ | 4 | 95 | 98 |
| 4 | $4\text{-}OMeC_6H_4CH_2OH$ | $4\text{-}OMeC_6H_4CHO$ | 4.5 | 92 | 98 |
| 5 | $4\text{-}NO_2C_6H_4CH_2OH$ | $4\text{-}NO_2C_6H_4CHO$ | 4.5 | 93 | 96 |
| 6 | 3-Pyridinemethanol | 3-Pyridinecarboxaldehyde | 2.5 | 100 | 59 |
| 7 | 2-Pyridinemethanol | Picolinaldehyde | 2.5 | 99 | 52 |
| 8 | $C_6H_5CH(OH)CH_3$ | $C_6H_5COCH_3$ | 3.5 | 99 | 98 |
| 9 | $4\text{-}MeC_6H_4CH(OH)CH_3$ | $4\text{-}MeC_6H_4COCH_3$ | 4 | 94 | 97 |
| 10 | $4\text{-}NO_2C_6H_4CH(OH)CH_3$ | $4\text{-}NO_2C_6H_4COCH_3$ | 6.5 | 72 | 98 |
| 11 | $(C_6H_5)_2CHOH$ | $(C_6H_5)_2CO$ | 3 | 99 | 99 |
| 12 | $C_6H_5CH(OH)(CH)_2C_6H_5$ | $C_6H_5CO(CH)_2C_6H_5$ | 3 | 99 | 91 |

[a] Reaction conditions: Substrate (1 mmol), glycoluril **3** (5 mol%), $H_2O_2$ (50%, 1.2 equiv.), $H_2O$ (2 mL), 60 °C.

Green, chemoselective, facile, and efficient oxidation of primary and secondary benzylic alcohols to corresponding aldehydes or ketones in aqueous $H_2O_2$ was reported by Kamal Amani and co-workers [35] in the presence of $GO/Fe_3O_4/HPW$ (**4**) nanocomposite catalyst with approximately 75% to 99% conversions, without any overoxidation to acid (100% selectivity), as shown in Table 4.

In water, exchanging the protons of HPW with PEG-bridged di-imidazolium cations produced double catalytic sites in a single molecule of Keggin-type phosphotungstic acid ($H_3PW_{12}O_{40}$, HPW)-based di-imidazolium ionic liquid (IL) hybrid (**5**) catalyst which enhanced the reaction rate and resulted in higher selectivity of benzyl alcohol oxidation with $H_2O_2$ (30 wt%) in water with excellent catalytic efficiency, convenient recovery, and steady reusability. Higher conversions (71%) and selectivity (82%) were obtained by the di-cation IL-based PIPA-0 due to the di-cationic structure of imidazolium IL (Table 5, Entry 4). In addition, the built-in phase transfer capability of PIPA-n that arose from PEG modifier provided the higher activity of PIPA-n (n = 4, 8, and 13) [36].

Incipient wetness impregnation method was applied for the preparation of ceria ($CeO_2$) supported tungstophosphoric acid ($H_3PW_{12}O_{40}$; HPW) catalysts. As illustrated in Table 6, $H_3PW_{12}O_{40}/CeO_2$ (**6**) catalyst among the various catalysts showed the best results for selective oxidation of benzyl alcohol (BzOH) with hydrogen peroxide (20 wt% $H_2O_2$) [37]. In addition, response surface methodology (RSM) based on the Box–Behnken design model showed 95.2% conversion of benzyl alcohol and a 94.2% yield of benzaldehyde with 98.9% selectivity which was in good agreement with the experimental results [37]. Reusability of the catalyst were further successfully tested for six consecutive runs.

**Table 4.** Catalytic performance of the GO/Fe$_3$O$_4$/HPW **4** nanocatalyst for the selective oxidation of alcohols with H$_2$O$_2$ under optimal reaction conditions [a].

| Entry | Substrate | Product | t/h | Yield (%) [b] | Sel. (%) [c] |
|---|---|---|---|---|---|
| 1 | C$_6$H$_5$CH$_2$OH | C$_6$H$_5$CHO | 3 | 99 | 100 |
| 2 | 4-OMeC$_6$H$_4$CH$_2$OH | 4-OMeC$_6$H$_4$CHO | 1.5 | 100 | 100 |
| 3 | 4-ClC$_6$H$_4$CH$_2$OH | 4-ClC$_6$H$_4$CHO | 2 | 99 | 100 |
| 4 | 2,4-ClC$_6$H$_3$CH$_2$OH | 2,4-ClC$_6$H$_3$CHO | 4 | 94 | 100 |
| 5 | 2-NO$_2$C$_6$H$_4$CH$_2$OH | 2-NO$_2$C$_6$H$_4$CHO | 9 | 70 | 100 |
| 6 | 4-NO$_2$C$_6$H$_4$CH$_2$OH | 4-NO$_2$C$_6$H$_4$CHO | 7 | 90 | 100 |
| 7 | 2-OHC$_6$H$_4$CH$_2$OH | 2-OHC$_6$H$_4$CHO | 6 | 85 | 100 |
| 8 | 4-CH$_2$OHC$_6$H$_5$CH$_2$OH | 4-CHOC$_6$H$_5$CHO | 7 | 70 | 100 |

[a] Reaction conditions: Substrate (1 mmol), GO/Fe$_3$O$_4$/HPW **4** (20 mg), H$_2$O$_2$ (10%, 5 mmol), 70 °C, 1.5-9 h. [b] Yields are referring to isolated yields. [c] Selectivity was based on the corresponding aldehydes or ketones. [d] Byproduct is phenylacetic acid. [e] Byproduct is 3-phenylpropanoic acid.

**Table 5.** Results of the selective oxidation of alcohols to aldehydes with H$_2$O$_2$ in water over various PW-based catalysts [a].

| Entry | Substrate | Product | Catalyst | Conv. (%) [b] | Sel. (%) [c] |
|---|---|---|---|---|---|
| 1 | | | HPW | 48 | 67 |
| 2 | | | [BMIm]Cl | Trace | - |
| 3 | C$_6$H$_5$CH$_2$OH | C$_6$H$_5$CHO | IL-PW | 59 | 80 |
| 4 | | | PIPA-0 | 71 | 82 |
| 5 | | | PIPA-4 | 88 | 79 |
| 6 | | | PIPA-8 | 91 | 84 |
| 7 | | | PIPA-13 | 96 | 86 |
| 8 | C$_6$H$_5$CH(OH)CH$_3$ | C$_6$H$_5$COCH$_3$ | HPW | 48 | >99 |
| 9 | | | IL-PW | 70 | >99 |
| 10 | | | PIPA-13 | 95 | >99 |
| 11 | (C$_6$H$_5$)$_2$CHOH | (C$_6$H$_5$)$_2$CO | HPW | 37 | >99 |
| 12 | | | IL-PW | 56 | >99 |
| 13 | | | PIPA-13 | 74 | >99 |

[a] Reaction conditions: Substrates (10 mmol), PIPA-13 **5** (1.3 mol% substrate, based on PW loading,), H$_2$O$_2$ (30 wt%, 15 mmol), H$_2$O (1.5 mL), 95 °C, 6 h. [b] Conversion as based on alcohols. [c] Selectivity was based on the corresponding aldehydes or ketones.

Unver [38] reported the synthesis of a water-soluble dinuclear Cu(II) complex, [Cu$_2$(OOCC$_6$H$_4$Br) (OCH$_3$)(C$_{10}$H$_8$N$_2$)$_2$(ClO$_4$)$_2$] (**7**) (4-bromobenzoic acid = HOOCC$_6$H$_4$Br; 2-2′-bipyridyl = C$_{10}$H$_8$N$_2$) which was successfully used without any additives under mild conditions in the oxidation of primary and secondary alcohols in water (Table 7). The catalyst showed high TON values (up to 100), good to moderate yields, and high selectivity with no traces of carboxylic acid producing during or after the reactions. In addition, the competition between the pure substrates as compared with the mixtures of alcohols, under the same conditions, resulted in higher product yields (Table 8) [38].

**Table 6.** Comparison of catalytic performances over various catalyst during oxidation of benzyl alcohol [a].

$H_3PW_{12}O_{40}/CeO_2$ **6** (0.8 g)

$BzOH/H_2O_2$ (1:2 mol mol$^{-1}$)

$H_2O$ (20 mL), 110 °C, 4 h

| Entry | Catalyst | BzOH Conv. (%) | BzH Sel. (%) | BzH yield (%) |
|-------|----------|----------------|--------------|---------------|
| 1 | $CeO_2$ | Nil | Nil | Nil |
| 2 | 15HPW/$CeO_2$ | 77.9 | 97.6 | 76 |
| 3 | 20HPW/$CeO_2$ | 94 | 98.2 | 92.3 |
| 4 | 25HPW/$CeO_2$ | 93.2 | 94.7 | 88.4 |
| 5 | 20HPW/$TiO_2$ | 84 | 96.6 | 81.2 |
| 6 | 20HPW/$ZrO_2$ | 90.8 | 91.7 | 83.3 |
| 7 | 20HPW/CeTiO [b] | 89.3 | 93.2 | 83.2 |

[a] Reaction conditions: $BzOH/H_2O_2$ (1:2 (mol mol$^{-1}$)), catalyst (0.8 g), $H_2O$ (20 mL), 110 °C, 4 h. [b] CeTiO was prepared by sol-gel method.

**Table 7.** Oxidation of various alcohols with copper (II) complex **7** in water [a].

$[Cu_2(OOCC_6H_4Br)(OCH_3)(C_{10}H_8N_2)_2(ClO_4)_2]$ **7**

$(5.7 \times 10^{-3}$ mmol)

$H_2O_2$ (19.5 mmol )

$H_2O$ (10 mL), 70 °C, 6 h

| Entry | Substrate | Product | Yield (%)[b] | TON | TOF (h$^{-1}$) |
|-------|-----------|---------|--------------|-----|----------------|
| 1 | $C_6H_5CH_2OH$ | $C_6H_5CHO$ | 100 | 100 | 16 |
| 2 | $C_6H_5CH(OH)CH_3$ | $C_6H_5COCH_3$ | 100 | 100 | 17 |

[a] Reaction conditions: Substrate (0.57 mmol), $[Cu_2(OOCC_6H_4Br)(OCH_3)(C_{10}H_8N_2)_2(ClO_4)_2]$ **7** $(5.7 \times 10^{-3}$ mmol), $H_2O_2$ (19.5 mmol), $H_2O$ (10 mL), 70 °C, 6 h. Blank experiment was conducted without catalyst for each substrate and negligible conversion was obtained (<3%). [b] Determined with GC.

**Table 8.** Selective oxidation of selected alcohols with copper (II) complex **7** in water [a].

$[Cu_2(OOCC_6H_4Br)(OCH_3)(C_{10}H_8N_2)_2(ClO_4)_2]$ **7**

$(5.7 \times 10^{-6}$ mmol)

$H_2O_2$ (19.5 mmol), $H_2O$ (10 mL)

70 °C, 6 h

**complex 7**

| Entry | Substrate | Product(s) | Conv. (%) [b] |
|-------|-----------|------------|---------------|
| 1 | $C_6H_5CH_2OH$ + $C_6H_5CH(OH)CH_3$ | $C_6H_5CHO$ + $C_6H_5COCH_3$ | 16.4 + 62.4 |

[a] Reaction conditions: Substrate $(5.7 \times 10^{-4}$ mmol), $[Cu_2(OOCC_6H_4Br)(OCH_3)(C_{10}H_8N_2)_2(ClO_4)_2]$ **7** $(5.7 \times 10^{-6}$ mmol), $H_2O_2$ (19.5 mmol), $H_2O$ (10 mL), 70 °C, 6 h. 1 mmol each of alcohols was tested with mmol complex **7**, mmol $H_2O_2$ in $H_2O$ at 70 °C. [b] Determined with GC.

Cobalt zeolitic imidazolate framework (ZIF-9@Zeolite) (**8**) was prepared and under a simple and clean protocol, aldehydes, as oxidative products of alcohols, were efficiently obtained in high yields (Table 9), generating water as the only byproduct and with no decrease in catalyst activity even after four catalytic cycles. Benzyl alcohols with electron-withdrawing substituent also yielded the corresponding aldehydes high percent (Table 9, Entries 4, 7, 11, and 12) [39].

**Table 9.** Oxidation of alcohols to aldehydes using ZIF-9@Zeolite **8** [a].

| Entry | Substrate | Product | t/min | Yield (%) | B.P/M.P (°C) | |
|---|---|---|---|---|---|---|
| | | | | | Found | Lit [b] |
| 1 | $C_6H_5CH_2OH$ | $C_6H_5CHO$ | 28 | 98 | 177 | 178 |
| 2 | $4\text{-}OMeC_6H_4CH_2OH$ | $4\text{-}OMeC_6H_4CHO$ | 27 | 99 | 246 | 248 |
| 3 | $4\text{-}ClC_6H_4CH_2OH$ | $4\text{-}ClC_6H_4CHO$ | 28 | 93 | 47 | 48 |
| 4 | $4\text{-}NO_2C_6H_4CH_2OH$ | $4\text{-}NO_2C_6H_4CHO$ | 29 | 84 | 104–106 | 106 |
| 5 | $4\text{-}MeC_6H_4CH_2OH$ | $4\text{-}MeC_6H_4CHO$ | 26 | 98 | 206 | 204–205 |
| 6 | $4\text{-}BrC_6H_4CH_2OH$ | $4\text{-}BrC_6H_4CHO$ | 30 | 85 | 58 | 55–58 |
| 7 | $4\text{-}FC_6H_4CH_2OH$ | $4\text{-}FC_6H_4CHO$ | 31 | 76 | 179–180 | 180 |
| 8 | $4\text{-}PhC_6H_4CH_2OH$ | $4\text{-}PhC_6H_4CHO$ | 28 | 86 | 58–59 | 57–59 |
| 9 | $3\text{-}ClC_6H_4CH_2OH$ | $3\text{-}ClC_6H_4CHO$ | 33 | 88 | 213–214 | 213–314 |
| 10 | $4\text{-}CO_2MeC_6H_4CH_2OH$ | $4\text{-}CO_2MeC_6H_4CHO$ | 30 | 83 | 60–62 | 59–63 |
| 11 | $3\text{-}NO_2C_6H_4CH_2OH$ | $3\text{-}NO_2C_6H_4CHO$ | 31 | 93 | 58–60 | 58.5 |
| 12 | $2\text{-}NO_2C_6H_4CH_2OH$ | $2\text{-}NO_2C_6H_4CHO$ | 29 | 95 | 42–44 | 43 |
| 13 | $3\text{-}MeC_6H_4CH_2OH$ | $3\text{-}MeC_6H_4CHO$ | 33 | 98 | 199 | 198–200 |
| 14 | $3\text{-}OMeC_6H_4CH_2OH$ | $3\text{-}OMeC_6H_4CHO$ | 27 | 99 | 142–144 | 143 |
| 15 | $3,4,5\text{-}OMeC_6H_2CH_2OH$ | $3,4,5\text{-}OMeC_6H_2CHO$ | 26 | 99 | 72–74 | 73 |

[a] Reaction conditions: Substrate (1 mmol), ZIF-9@Zeolite **8** (0.5 mmol), $H_2O_2$ (30%, 2 mmol), $H_2O$ (5 mL), 80 °C, 26 to 33 min. [b] Sigma-Aldrich.

The influence of polyetheramine (Jeffamine®) as a di-block copolymer with ethylene oxide and propylene oxide moieties along with terminal amine on phosphotungstic acid (PTA) (**9**) which is a polyoxometalate catalyst was determined, in detail, in $H_2O_2$-mediated oxidation of BzOH in water. Recyclable PTA-Jffamine® catalyst not only enhanced the conversion of BzOH as compared with pristine PTA, but also facilitated the easy separation of catalyst and benzaldehyde (BzH) from the reaction mixture. In addition, with and without pure-PTA as the catalyst, the BzOH oxidation reaction was investigated (Table 10). Very low conversions of BzOH were obtained for reactions with and without pure-PTA (Entries 1 and 2). In pH = 3.5, the conversion was only 25% after 1.5 h at low Jeffamine® concentration, but dramatically increased to 100% at higher Jeffamine® contents at (pH = 4.5, 6.5, 7.5, and 8.5) which had not been reported earlier (Table 10) [40].

**Table 10.** Summary of GC-MS results obtained from reactions performed at various conditions [a].

| Entry | Catalyst | pH | Conv. (%) | Sel. (%) |
|---|---|---|---|---|
| 1 | No catalyst | - | 2.9 | 5 |
| 2 | PTA (0.5 g) | 2.5 | 9 | 100 |
| 3 | Potassium salt of PTA (0.5 g) | 7.5 | 5 | 100 |
| 4 | PTA-Jffamine® (0.5 g) | 7.5 | 100 | 74 |
| 5 | PTA-Jffamine® (0.5 g) | 3.5 | 25 | 73 |
| 6 | PTA-Jffamine® (0.5 g) | 4.5 | 100 | 63 |

| Entry | Catalyst | pH | Conv. (%) | Sel. (%) |
|---|---|---|---|---|
| 7 | PTA-Jffamine® (0.5 g) | 6.5 | 100 | 59 |
| 8 | PTA-Jffamine® (0.5 g) | 8.5 | 100 | 73 |

[a] Amount of catalyst (PTA) **9** taken is mentioned in brackets, Medium-water, pH = 2.5–8.5, 90 °C, 1.5 h.

Water-soluble heteropolyacid-based ionic liquids were prepared by modifying tungstophosphoric acid ($H_3PW_{12}O_{40}$) and propyl sulfonic acid-functionalized ionic complex. Among various organic TPA salts, the [DMBPSH]$H_2PW_{12}O_{40}$ (**10**) catalyst, due to strong acidity and excellent surface activity played as an effective and reusable catalyst, exhibited the best oxidative activity with a desirable BzH selectivity of 97.0% and an excellent BzOH conversion of 98.5% under optimum conditions (Table 11) [41]. Additionally, after six consecutive experimental cycles, the catalyst showed no decreasing in conversion and selectivity.

**Table 11.** Catalytic performance of various catalysts during oxidation of benzyl alcohol with $H_2O_2$ [a].

| Entry | Catalyst | Conv. (%) [b] | Yield (%) | Sel. (%) [b] |
|---|---|---|---|---|
| 1 | $H_3PW_{12}O_{40}$ | 62.3 | 60.4 | 93.5 |
| 2 | DMBPS | 5.4 | 5.3 | 99.7 |
| 3 | [TEAPSH]$H_2PW_{12}O_{40}$ [c] | 68.1 | 65.1 | 95.6 |
| 4 | [DMPPSH]$H_2PW_{12}O_{40}$ [d] | 89.5 | 85.3 | 95.3 |
| 5 | [DMBPSH]$H_2PW_{12}O_{40}$ | 94.4 | 91.6 | 96.4 |
| 6 | [DMBPSH]$_2$HPW$_{12}$O$_{40}$ | 92.5 | 88.4 | 95.6 |
| 7 | [DMBPSH]$_3$PW$_{12}$O$_{40}$ | 90.6 | 86.7 | 95.7 |

[a] Reaction conditions: BzOH/$H_2O_2$ molar ratio (1:2 (mol/mol)), catalyst (5 wt%), $H_2O$ (20 mL), 120 °C, 3 h. [b] Analyzed by GC. [c] TEAPS, triethylammonium propyl sulfobetaine. [d] DMPPS, *N,N*-dimethyl(phenyl)ammonium propyl sulfobetaine.

A series of diverse amino acids such as phenylalanine, alanine, and glycine functionalized tungstophosphoric acid (TPA;$H_3PW_{12}O_{40}$) composite and were efficiently applied as recyclable thermally stable, eco-friendly, and cost-effective heterogeneous catalysts in the selective oxidation of benzyl alcohol (Table 12) [42]. Although [GlyH]$H_2PW_{12}O_{40}$ (**11**) had more Brønsted acidity than [PheH]$H_2PW_{12}O_{40}$ (**12**), [PheH]$H_2PW_{12}O_{40}$ (**12**) exhibited the best catalytic activity including conversion 97.9%, selectivity 97.4%, and yield of 95.4% [42,43]. This phenomenon indicated that catalytic oxidation required only a modest acidity.

**Table 12.** Catalytic performances of various catalysts during oxidation of benzyl alcohol with hydrogen peroxide [a].

| Entry | Catalyst | Conv. (%) [b] | Yield (%) [b] | Sel. (%) [b] |
|---|---|---|---|---|
| 1 | Phe | Nil | Nil | Nil |
| 2 | $H_3PW_{12}O_{40}$ | 96.7 | 87 | 90 |
| 3 | [PheH]$H_2PW_{12}O_{40}$ | 97.9 | 95.4 | 97.4 |
| 4 | [PheH]$_2$HPW$_{12}$O$_{40}$ | 97.5 | 89.7 | 92 |
| 5 | [PheH]$_3$PW$_{12}$O$_{40}$ | 96.3 | 88.4 | 91.8 |
| 6 | [AlaH]$H_2PW_{12}O_{40}$ | 97.8 | 92.7 | 94.7 |
| 7 | [GlyH]$H_2PW_{12}O_{40}$ | 97.7 | 90.8 | 92.9 |

[a] Reaction conditions: BzOH/$H_2O_2$ (1:2 (mol/mol)), catalyst (6 wt%), $H_2O$ (30 mL), 110 °C, 4 h. [b] Analyzed by GC.

Oxidation of primary alcohols to aldehydes catalyzed by $H_2O_2$ as an oxidant and a reusable and water-soluble iron (III) catalyst in water (Table 13) [44]. This novel and reusable $FeCl_3$ complex (**13**) in situ formed with quaternary ammonium salt-functionalized 8-aminoquinoline. The reaction showed not only unique chemoselectivity similar to the oxidation a benzylic primary alcohol even in the presence of an aliphatic one, but also exhibited broad functional-group tolerance.

**Table 13.** The oxidation of benzylic primary alcohols to aldehydes [a].

| Entry | Substrate | Product | t/h | Yield (%) [b] |
|---|---|---|---|---|
| 1 | 2-OMeC$_6$H$_4$CH$_2$OH | 2-OMeC$_6$H$_4$CHO | 2 | 87 |
| 2 | 4-PhC$_6$H$_4$CH$_2$OH | 4-PhC$_6$H$_4$CHO | 2 | 91 |
| 3 | 3-ClC$_6$H$_4$CH$_2$OH | 3-ClC$_6$H$_4$CHO | 3 | 89 |
| 4 | 3,4-OCH$_2$OC$_6$H$_3$CH$_2$OH | 3,4-OCH$_2$OC$_6$H$_3$CHO | 3 | 94 |
| 5 | 4-NO$_2$C$_6$H$_4$CH$_2$OH | 4-NO$_2$C$_6$H$_4$CHO | 3 | 82 |
| 6 | 2-FC$_6$H$_4$CH$_2$OH | 2-FC$_6$H$_4$CHO | 3 | 73 |
| 7 | 4-N(CH$_3$)$_2$C$_6$H$_4$CH$_2$OH | 4-N(CH$_3$)$_2$C$_6$H$_4$CHO | 3 | 85 |
| 8 | 4-COOCH$_3$C$_6$H$_4$CH$_2$OH | 4-COOCH$_3$C$_6$H$_5$CHO | 3 | 86 |
| 9 | 1-Naphthylmethanol | 1-Naphthaldehyde | 3 | 83 |
| 10 | Indole-3-carbinol | Indole-3-carboxaldehyde | 3 | 80 |
| 11 | 4-Pyridinemethanol | Isonicotinaldehyde | 2 | 79 |
| 12 | (5-Bromothiophen-2-yl)methanol | 5-Bromothiophene-2-carbaldehyde | 3 | 77 |
| 13 | [5-(4-Bromophenyl)-2-furyl]methanol | 5-(4-Bromophenyl)furan-2-carbaldehyde | 3 | 88 |
| 14 | C$_6$H$_5$CH(OH)Et | C$_6$H$_5$COEt | 5 | Trace [c] 91 recovery |
| 15 | 4-OMeC$_6$H$_4$CH$_2$OH (a) + C$_6$H$_5$CH(OH)Et (b) | 4-OMeC$_6$H$_4$CHO | 2 | 85 [d] |

[a] Reaction conditions: Substrate (1 mmol), the in situ formed FeCl$_3$/**I** complex **13** (2 mol%), H$_2$O$_2$ (30 wt%, 2 mmol), H$_2$O (1 mL), room temperature, 2 to 5 h. [b] Isolated yield. [c] Apart from at room temperature, heating at 80 °C was also performed. [d] A mixture of **a** and **b** in a molar ratio of 1:1 was used.

Various alcohols were oxidized with hydrogen peroxide over ammonium tungstate promoted by GO (**14**) as heterogeneous acid catalyst (Table 14) [45]. The aromatic primary alcohols and secondary alcohols were converted to the corresponding aldehydes and ketones in excellent to satisfactory yields. In addition, the catalytic system could be efficiently reused in at least seven cycles.

**Table 14.** Substrate scope of the alcohol oxidation reaction [a].

| Entry | Substrate | Product | Yield (%) [b] |
|---|---|---|---|
| 1 | C$_6$H$_5$CH(OH)CH$_3$ | C$_6$H$_5$COCH$_3$ | 96 |
| 2 | 3-MeC$_6$H$_4$CH(OH)CH$_3$ | 3-MeC$_6$H$_4$COCH$_3$ | 94 |
| 3 | 3-ClC$_6$H$_4$CH(OH)CH$_3$ | 3-ClC$_6$H$_4$COCH$_3$ | 93 |
| 4 | (C$_6$H$_5$)$_2$CHOH | (C$_6$H$_5$)$_2$CO | 98 |
| 5 | C$_6$H$_5$CH(OH)CH$_2$C$_6$H$_5$ | C$_6$H$_5$COCH$_2$C$_6$H$_5$ | 96 |
| 6 | C$_6$H$_5$CH$_2$OH | C$_6$H$_5$CHO | 91 |

[a] Reaction conditions: Substrate (2 mmol), GO **14** (0.01 g), (NH$_4$)$_3$H$_5$[H$_2$(W$O_4$)$_6$] (0.03 mmol, 0.05 g), H$_2$O$_2$ (30%, 8 mmol, 0.92 g), H$_2$O (3 mL), 70 °C, 11 h. [b] Isolated yield.

Cu(II) nanoparticles immobilized on nanocage-like mesoporous KIT-5 as a support and a 3-aminopropyltriethoxysilane (APTES) group as a coordinating agent for Cu(II) provided an active catalyst in the selective oxidation of primary and secondary alcohols in water. The computational investigation also confirmed the catalytic role of APTES-KIT-5 silica-supported copper(II) nanocatalyst (**15**). The results in Table 15 showed that the oxidation of various benzylic alcohols were considerably dependent on substituents (–Cl, –OCH$_3$, –OH, –NO$_2$, and –NH$_2$) and on their positions [46]. In addition, aldehydes obtained with excellent selectivity without overoxidation into carboxylic acids from the oxidation of primary alcohols and secondary benzylic alcohols oxidation provided the corresponding ketone in satisfactory yield (Entry 12). The mentioned heterogeneous nanocatalyst could be recovered and reused six times.

**Table 15.** Cu(OAc)$_2$ supported on AK **15** as catalyst in oxidation of alcohols to corresponding aldehydes [a].

| Entry | Substrate | Product | t/min [c] | Yield (%) [d,e] |
|-------|-----------|---------|-----------|------------------|
| 1 | C$_6$H$_5$CH$_2$OH | C$_6$H$_5$CHO | 21 | 90 |
| 2 | 2-ClC$_6$H$_4$CH$_2$OH | 2-ClC$_6$H$_4$CHO | 23 | 75 |
| 3 [b] | 3-OHC$_6$H$_4$CH$_2$OH | 3-OHC$_6$H$_4$CHO | 50 | 85 |
| 4 | 2-OHC$_6$H$_4$CH$_2$OH | 2-OHC$_6$H$_4$CHO | 5 | 97 |
| 5 | 4-OHC$_6$H$_4$CH$_2$OH | 4-OHC$_6$H$_4$CHO | 5 | 98 |
| 6 | 2-NH$_2$C$_6$H$_5$CH$_2$OH | 2-NH$_2$C$_6$H$_5$CHO | 7 | 97 |
| 7 | 3-OMeC$_6$H$_5$CH$_2$OH | 3-OMeC$_6$H$_5$CHO | 22 | 83 |
| 8 | 2-NO$_2$C$_6$H$_4$CH$_2$OH | 2-NO$_2$C$_6$H$_4$CHO | 23 | 48 |
| 9 | 4-NO$_2$C$_6$H$_4$CH$_2$OH | 4-NO$_2$C$_6$H$_4$CHO | 35 | 61 |
| 10 | 3-NO$_2$C$_6$H$_4$CH$_2$OH | 3-NO$_2$C$_6$H$_4$CHO | 22 | 56 |
| 11 | Hydroquinone | Benzoquinone | 5 | 96 |
| 12 | (C$_6$H$_5$)$_2$CHOH | (C$_6$H$_5$)$_2$CO | 60 | 70 |

[a] Reaction conditions: Substrate (1 mmol), Cu(OAc)$_2$/APTES-KIT-5 **15** (0.05 g), H$_2$O$_2$ (3 mmol), H$_2$O (2 mL), reflux, all reactions carried out in reflux condition, except Entries 5, 9,11, which were performed at room temperature, 5 to 60 min. [b] Acetone was used as a solvent. [c] Time of maximum conversion (determined by TLC/or GC). [d] Isolated yield. [e] All reactions give corresponding aldehyde in 100% selectivity.

Tungstate salt with imidazolium (((1,3,5-triazine-2,4,6-triyl)tris(1-octyl-1H-imidazol-3-ium))$_2$(WO$_4$$^=$)$_3$ (**16**) framework provided a catalytic system, under neutral aqueous reaction conditions, for the highly selective oxidation of primary benzylic alcohols (bearing both electron-releasing and electron-withdrawing groups) using H$_2$O$_2$ as a green oxidant (Table 16) [47]. The catalyst could be reused for at least seven subsequent reaction cycles.

**Table 16.** Oxidation of substituted benzylic alcohols to aldehydes using imidazolium WO$_4$$^=$ salt **16** as catalyst [a].

| Entry | Substrate | Product | t/min | Yield (%) | B.P/M.P (°C) | |
|-------|-----------|---------|-------|-----------|--------------|---|
| | | | | | Found | Lit. |
| 1 | C$_6$H$_5$CH$_2$OH | C$_6$H$_5$CHO | 5 | 99 | 177–178 | 178 |

**Table 16.** *Cont.*

| Entry | Substrate | Product | t/min | Yield (%) | B.P/M.P (°C) Found | B.P/M.P (°C) Lit. |
|---|---|---|---|---|---|---|
| 2 | 4-OMeC$_6$H$_4$CH$_2$OH | 4-OMeC$_6$H$_4$CHO | 7 | 93 | 246–248 | 248 |
| 3 | 4-ClC$_6$H$_4$CH$_2$OH | 4-ClC$_6$H$_4$CHO | 7 | 95 | 47–48 | 48 |
| 4 | 4-NO$_2$C$_6$H$_4$CH$_2$OH | 4-NO$_2$C$_6$H$_4$CHO | 10 | 94 | 104–105 | 106 |
| 5 | 4-MeC$_6$H$_4$CH$_2$OH | 4-MeC$_6$H$_4$CHO | 11 | 95 | 204–206 | 205 |
| 6 | 4-BrC$_6$H$_4$CH$_2$OH | 4-BrC$_6$H$_4$CHO | 13 | 93 | 56–58 | 55–58 |
| 7 | 4-PhC$_6$H$_4$CH$_2$OH | 4-PhC$_6$H$_4$CHO | 8 | 97 | 57–59 | 56–58 |
| 8 | 4-CO$_2$MeC$_6$H$_4$CH$_2$OH | 4-CO$_2$MeC$_6$H$_4$CHO | 3 | 87 | 59–61 | 61–62 |
| 9 | 4-FC$_6$H$_4$CH$_2$OH | 4-FC$_6$H$_4$CHO | 13 | 86 | 178–180 | 180 |
| 10 | 3-ClC$_6$H$_4$CH$_2$OH | 3-ClC$_6$H$_4$CHO | 16 | 90 | 213–215 | 211–213 |
| 11 | 4-(*tert*-Butyl)-C$_6$H$_4$CH$_2$OH | 4-(*tert*-Butyl)-C$_6$H$_4$CHO | 11 | 94 | 128–130 | 130 |
| 12 | 4-N(CH$_3$)$_2$C$_6$H$_4$CH$_2$OH | 4-N(CH$_3$)$_2$C$_6$H$_4$CHO | 5 | 96 | 73–75 | 74 |
| 13 | 3-NO$_2$C$_6$H$_4$CH$_2$OH | 3-NO$_2$C$_6$H$_4$CHO | 14 | 87 | 58-60 | 57–59 |
| 14 | 3-OPhC$_6$H$_4$CH$_2$OH | 3-OPhC$_6$H$_4$CHO | 14 | 92 | 168–169 | 169 |
| 15 | 2-NO$_2$C$_6$H$_4$CH$_2$OH | 2-NO$_2$C$_6$H$_4$CHO | 12 | 92 | 43–45 | 43 |
| 16 | 2-ClC$_6$H$_4$CH$_2$OH | 2-ClC$_6$H$_4$CHO | 14 | 95 | 210–215 | 212–213 |
| 17 | 2-OHC$_6$H$_4$CH2OH | 2-OHC$_6$H$_4$CHO | 15 | 85 | 196–198 | 197 |
| 18 | 3,4-ClC$_6$H$_3$CH$_2$OH | 3,4-ClC$_6$H$_3$CHO | 10 | 96 | 40–42 | 43 |
| 19 | 2,4-ClC$_6$H$_3$CH$_2$OH | 2,4-ClC$_6$H$_3$CHO | 15 | 86 | 71–73 | 69–70 |
| 20 | 3,4-OMeC$_6$H$_3$CH$_2$OH | 3,4-OMeC$_6$H$_3$CHO | 10 | 88 | 42–44 | 42–43 |

[a] Reaction conditions: Substrate (0.5 mmol), ((1,3,5-triazine-2,4,6-triyl)tris(1-octyl-1*H*-imidazol-3-ium))$_2$(WO$_4$$^=$)$_3$ **16** (0.21 mmol), H$_2$O$_2$ (30%, 1 mmol), H$_2$O (2 mL), room temperature, 3 to 16 min.

Via simple method, Na$_7$PW$_{11}$O$_{39}$ (PW$_{11}$) was immobilized on quarternary ammonium functionalized chloromethylated polystyrene (DMA16/CMPS) (PW$_{11}$-DMA16/CMPS (**17**)) and used as high active, stable, recoverable, and recyclable catalyst in the oxidation of aliphatic and aromatic alcohol with H$_2$O$_2$ (Table 17) [48]. Although, benzhydrol oxidation is difficult because of the deficiency of interaction between the alcohol moiety and the catalyst [49], in this catalytic system, benzhydrol attained 100% conversion and 97.5% selectivity (Entry 1).

**Table 17.** Oxidation of various alcohols catalyzed by PW$_{11}$-DMA16/CMPS **17** with 30% H$_2$O$_2$ [a].

| Entry | Substrate | Product | T (°C) | X$_{ANOL}$ (%) | S$_{ANOL}$ (%) |
|---|---|---|---|---|---|
| 1 | (C$_6$H$_5$)$_2$CHOH | (C$_6$H$_5$)$_2$CO | 80 | 100 | 97.5 |
| 2 | C$_6$H$_5$CH(OH)CH$_3$ | C$_6$H$_5$COCH$_3$ | 90 | 92 | 98.7 |
| 3 | C$_6$H$_5$CH$_2$OH | C$_6$H$_5$CHO | 80 | 91.7 | 84 |

[a] Reaction conditions: PW$_{11}$-DMA16/CMPS **17** (0.3 g), H$_2$O$_2$:alcohols (2:1 mol/mol), H$_2$O (10 mL), 80 to 90 °C, 6 h.

As reported in Table 18, Ramazani et al. [50] introduced nanomagnetic MgFe$_2$O$_4$ (**18**) as an active, and reusable (seven runs) catalyst for the oxidation of various primary and secondary alcohols in good yields in water as a solvent, and either oxone (at room temperature) or H$_2$O$_2$ (at 60 °C) as an oxidant. Overoxidation of aldehydes to the corresponding carboxylic acids was not observed which emphasized that the aldehyde selectivity, in most cases, was quite high (>99%).

**Table 18.** Oxidation of various alcohols using MgFe$_2$O$_4$ MNPs **18** as catalyst in the present of H$_2$O$_2$ in water [a].

| Entry | Substrate | Product | t/min | Yield (%) [b] |
|-------|-----------|---------|-------|-----------|
| 1 | C$_6$H$_5$CH$_2$OH | C$_6$H$_5$CHO | 55 | 88 |
| 2 | 4-ClC$_6$H$_4$CH$_2$OH | 4-ClC$_6$H$_4$CHO | 60 | 86 |
| 3 | 3-ClC$_6$H$_4$CH$_2$OH | 3-ClC$_6$H$_4$CHO | 65 | 87 |
| 4 | 2-ClC$_6$H$_4$CH$_2$OH | 2-ClC$_6$H$_4$CHO | 60 | 85 |
| 5 | 2,5-ClC$_6$H$_3$CH$_2$OH | 2,5-ClC$_6$H$_3$CHO | 80 | 85 |
| 6 | 2-BrC$_6$H$_4$CH$_2$OH | 2-BrC$_6$H$_4$CHO | 65 | 87 |
| 7 | 3-BrC$_6$H$_4$CH$_2$OH | 3-BrC$_6$H$_4$CHO | 65 | 86 |
| 8 | 4-BrC$_6$H$_4$CH$_2$OH | 4-BrC$_6$H$_4$CHO | 60 | 84 |
| 9 | 4-FC$_6$H$_4$CH$_2$OH | 4-FC$_6$H$_4$CHO | 70 | 88 |
| 10 | 3-FC$_6$H$_4$CH$_2$OH | 3-FC$_6$H$_4$CHO | 70 | 88 |
| 11 | 4-NO$_2$C$_6$H$_4$CH$_2$OH | 4-NO$_2$C$_6$H$_4$CHO | 120 | 81 |
| 12 | 3-NO$_2$C$_6$H$_4$CH$_2$OH | 3-NO$_2$C$_6$H$_4$CHO | 120 | 85 |
| 13 | 2-NO$_2$C$_6$H$_4$CH$_2$OH | 2-NO$_2$C$_6$H$_4$CHO | 120 | 86 |
| 14 | 4-OMeC$_6$H$_4$CH$_2$OH | 4-OMeC$_6$H$_4$CHO | 50 | 92 |
| 15 | (C$_6$H$_5$)$_2$CHOH | (C$_6$H$_5$)$_2$CO | 70 | 87 |
| 16 | C$_6$H$_5$CH(OH)COC$_6$H$_5$ | (C$_6$H$_5$)$_2$(CO)$_2$ | 100 | 72 |

[a] Reaction conditions: Substrate (1 mmol), MgFe$_2$O$_4$ MNPs **18** (5 mol%, 10 mg), H$_2$O$_2$ (1.3 mmol), H$_2$O (2 mL), 60 °C, 50 to 120 h. [b] Isolated.

A solution based chemical reduction method was applied for the production of gold nanoparticles supported on magnesium oxide nanorods (Au-MgO) (**19**) which were found to be an efficient heterogeneous catalyst with hydrogen peroxide for the base free oxidation of alcohols in aqueous medium at room temperature, Table 19 [51]. It is worthwhile to mention that the catalyst was reused for five cycles.

**Table 19.** Oxidation of alcohols using Au-MgO nanorodes **19** [a].

| Entry | Substrate | Product | t/h | Yield (%) [b] | Sel. (%) [d] |
|-------|-----------|---------|-----|-----------|-----------|
| 1 | C$_6$H$_5$CH(OH)CH$_3$ | C$_6$H$_5$COCH$_3$ | 5 | 68 | 100 |
| 2 | 4-ClC$_6$H$_4$CH(OH)CH$_3$ | 4-ClC$_6$H$_4$COCH$_3$ | 24 | 21 | 100 |
| 3 | 4-BrC$_6$H$_4$CH(OH)CH$_3$ | 4-BrC$_6$H$_4$COCH$_3$ | 24 | 4 | 100 |
| 4 | 1-(naphthalene-2-yl)ethanol | 2-Acetonaphthanone | 5 | 98 | 100 |
| 5 | (C$_6$H$_5$)$_2$CHOH | (C$_6$H$_5$)$_2$CO | 5 | 41 [c] | 100 |
| 6 | 1-Indanol | 1-Indanone | 6 | 90 | 100 |
| 7 | C$_6$H$_5$CH$_2$OH | C$_6$H$_5$CHO | 4 | 50 | 99 |
| 8 | 4-OMeC$_6$H$_4$CH$_2$OH | 4-OMeC$_6$H$_4$CHO | 24 | 15 [c] | 100 |
| 9 | 4-NO$_2$C$_6$H$_4$CH$_2$OH | 4-NO$_2$C$_6$H$_4$CHO | 3 | 99 | 99 |

**Table 19.** *Cont.*

| Entry | Substrate | Product | t/h | Yield (%) [b] | Sel. (%) [d] |
|-------|-----------|---------|-----|-----------|-----------|
| 10 | 4-CH$_2$ClC$_6$H$_4$CH$_2$OH | 4-CH$_2$ClC$_6$H$_4$CHO | 4 | 50 | 67 |

[a] Reaction conditions: Substrate (1 mM), Au-MgO nanorodes **19** (30 mg), H$_2$O$_2$ (15 equiv.), H$_2$O (20 mL), 27 °C, 3 to 24 h. [b] Determined from GC. [c] Determined from GC-MS. [d] Selectivity = [(peak area of substrate + peak area of product/total peak area)] × 100.

Perez[52] reported the application of copper(II) complexes, copper, and copper oxide nanoparticles supported on SBA-15 applied in the benzyl alcohol oxidation in aqueous phase as catalyst and in the presence of H$_2$O$_2$ (Table 20). Immobilization of ionic liquid containing copper followed by chemical reduction method provided the catalyst which showed the highest benzyl alcohol oxidation activity with a 73% conversion and 54% selectivity with 30 min reaction time. Different copper species of the catalyst explored different conversion and selectivity. The highest activity, with 73% conversion and 54% selectivity for benzaldehyde, was obtained in metallic copper nanoparticle catalyst such as Cu/IMILeSBA-15-G1 (**20**) (Entry 9), in 30 min. The **20** had also the higher activity as compare with similar previously reported catalyst such as copper nanoparticle-polyacrylamide/SBA-15 and polymer-supported copper(II)-*L*-valine complexes [53,54].

**Table 20.** Benzyl alcohol oxidation catalyzed by the different copper supported catalysts [a].

| Entry | Catalyst | Conv. (%) | Sel. (%) [b] | TOF (h$^{-1}$) [c] |
|-------|----------|-----------|-----------|------------|
| 1 | SBA-15 | 0.9 | 100 | - |
| 2 | Cu(II)-PADO-HMDS-SBA-15 | 16 | 78 | 23 |
| 3 | Cu(II)-IMIL-SBA-15-G1 | 32 | 69 | 39 |
| 4 | Cu(II)-IMIL-SBA-15-G2 | 65 | 51 | 32 |
| 5 | Cu(II)-IMIL-SBA-15-G2-R [d] | 14 | 71 | 7 |
| 6 | CuO/SBA-15-C1 | 19 | 83 | 63 |
| 7 | 3.7%CuO/SBA-15-C2 | 11 | 82 | 11 |
| 8 | 5%CuO/SBA-15-C2 | 20 | 73 | 15 |
| 9 | Cu/IMIL-SBA-15-G1 | 73 | 54 | 66 |
| 10 | Cu/IMIL-SBA-15-G2 | 66 | 33 | 27 |
| 11 | Cu/IMIL-SB-15-G2-R [d] | 16 | 66 | 7 |
| 12 | 2%Cu/PEG/SBA-15 | 27 | 68 | 52 |
| 13 | 2%Cu/PEG/SBA-15-R [d] | 25 | 67 | 48 |

[a] Reaction conditions: Benzyl alcohol (3 mmol), catalyst (0.1 g), H$_2$O$_2$ (9 mmol), H$_2$O (5 mL), 80 °C, 30 min. [b] Selectivity towards benzaldehyde determined by GC. [c] TOF = moles of benzyl alcohol converted per mol of copper per hour. [d] Reused material.

The efficient and selective oxidation of primary and secondary alcohols were reported by Shi[55] who immobilized ionic liquids/peroxotungstates/SiO$_2$ (**21**) catalyzed reactions in the presence of H$_2$O$_2$ as an oxidant in neat water (Table 21). Hydrophilic imidazoliums and hydrophobic hydrocarbon chains in the ILs caused to diffusion of both hydrophobic alcohols and hydrophilic H$_2$O$_2$ oxidant into the micro reactor and provided carbonyls catalyzed by the peroxotungstates. Substituted-benzylic alcohols were oxidized to selective carbonyl products in satisfied yields but electronic effect of substituted groups on the activity was significantly observed. In addition, aromatic secondary alcohols were also selectively oxidized to ketones in good yields. Moreover, the catalyst was easily recovered by filtration and reused at least for six times.

**Table 21.** The selective oxidation of different alcohols catalyzed by $SiO_2$-BisILs$[W_2O_3(O_2)_4]$ **21** with $H_2O_2$ [a].

| Entry | Substrate | Product | t/h | Yield (%) [b] | Sel. (%) [c] |
|-------|-----------|---------|-----|---------------|--------------|
| 1 | $C_6H_5CH_2OH$ | $C_6H_5CHO$ | 18 | 96 | 98 |
| 2 | $4\text{-}OMeC_6H_4CH_2OH$ | $4\text{-}OMeC_6H_4CHO$ | 18 | 94 | 95 |
| 3 | $2\text{-}MeC_6H_4CH_2OH$ | $2\text{-}MeC_6H_4CHO$ | 18 | 88 | 96 |
| 4 | $4\text{-}ClC_6H_4CH_2OH$ | $4\text{-}ClC_6H_4CHO$ | 22 | 81 | 87 |
| 5 | $4\text{-}BrC_6H_4CH_2OH$ | $4\text{-}BrC_6H_4CHO$ | 22 | 80 | 85 |
| 6 | $2\text{-}BrC_6H_4CH_2OH$ | $2\text{-}BrC_6H_4CHO$ | 22 | 73 | 88 |
| 7 | $4\text{-}NO_2C_6H_4CH_2OH$ | $4\text{-}NO_2C_6H_4CHO$ | 30 | 75 | 82 |
| 8 | $3,4\text{-}OMeC_6H_3CH_2OH$ | $3,4\text{-}OMe\ C_6H_3CHO$ | 26 | 67 | 97 |
| 9 | $(C_6H_5)_2CHOH$ | $(C_6H_5)_2CO$ | 24 | 58 | 100 |
| 10 | $C_6H_5CH(OH)CH_3$ | $C_6H_5COCH_3$ | 18 | 88 | 99 |
| 11 | $C_6H_5CH_2OH$ | $C_6H_5CHO$ | 18 | 89 | 99 |

[a] Reaction conditions: Substrate (1 mmol), $SiO_2$-BisILs$[W_2O_3(O_2)_4]$ **21** (1 mol%) based on content of tungstate, $H_2O_2$ (30%, 1.4 mmol), $H_2O$ (2 mL), 90 °C, 18 to 30 h. [b] GC yields. [c] The selectivity of product was calculated based on GC.

Bis-imidazolium tungstate ionic liquid produced magnetically recoverable catalyst, at least 5 times, which were extremely dispersible in water and could selectively oxidized a wide variety of alcohols and sulfides using $H_2O_2$ oxidant. The results in Table 22 demonstrated the key role of MNP@IL/W (**22**) as both oxidant and phase transfer catalyst in various alcohol oxidations [56]. Good to excellent conversion and selectivity was also obtained in oxidation of secondary alcohols.

**Table 22.** Oxidation of alcohols to aldehydes catalyzed by MNP@IL/W **22** [a].

| Entry | Substrate | Product | t/h | Conv. (%) [b] | Yield (%) [c] |
|-------|-----------|---------|-----|---------------|---------------|
| 1 | $C_6H_5CH_2OH$ | $C_6H_5CHO$ | 3 | 99 | 95 |
| 2 | $4\text{-}OMeC_6H_4CH_2OH$ | $4\text{-}OMeC_6H_4CHO$ | 4 | 89 | 85 |
| 3 | $4\text{-}MeC_6H_4CH_2OH$ | $4\text{-}MeC_6H_4CHO$ | 5 | 96 | 95 |
| 4 | $4\text{-}ClC_6H_4CH_2OH$ | $4\text{-}ClC_6H_4CHO$ | 3 | 99 | 94 |
| 5 | $4\text{-}BrC_6H_4CH_2OH$ | $4\text{-}BrC_6H_4CHO$ | 3 | 99 | 97 |
| 6 | $2\text{-}ClC_6H_4CH_2OH$ | $2\text{-}ClC_6H_4CHO$ | 8 | 85 | 85 [d] |
| 7 | $2\text{-}NO_2C_6H_4CH_2OH$ | $2\text{-}NO_2C_6H_4CHO$ | 10 | 68 | 65 [d] |
| 8 | $C_6H_5CH(OH)Et$ | $C_6H_5COEt$ | 4 | 99 | 99 [d] |

[a] Reaction conditions: Substrate (1 mmol), MNP@IL/W **22** (3 mol%), $H_2O_2$ (30%, 3 mmol), $H_2O$ (2 mL), 90 °C, 3 to 10 h. [b] Conversion calculated based on initial mmol of substrates. [c] Yields of aldehydes were determined by GC. [d] Solvent, 1$CH_3CN$:2$H_2O$ (2 mL).

Using the anion-exchange method produced some hybrid materials ($[C4mim]_{3+x}PMo_{12-x}V_xO_{40}$, x = 0, 1, 2) based on V-substituted phosphomolybdic acid $H_{3+x}PMo_{12-x}V_xO_{40}$ (x = 0, 1, 2) and ionic liquid 1-butyl-3-methyl imidazolium bromide ([C4mim]Br) and applied in the benzyl alcohol oxidation reaction Table 23 [57]. The hybrids $[C4mim]_{3+x}PMo_{12-x}V_xO_{40}$ demonstrated much higher catalytic

activities than both corresponding moieties. Particularly, $[C4mim]_4PMo_{11}VO_{40}$ (**23**) provided 34% benzyl alcohol conversion and 99% selectivity for benzaldehyde under the optimized conditions (Entry 7) and was reused for five runs without much decrease in selectivity and conversion. In this case, the reaction mixture turned out to be *L-L-S* triphase system because of the insolubility of the solid hybrid and immiscibility between water and benzyl alcohol [57]. Although high selectivities for benzaldehyde were obtained, in all cases, conversions of the benzyl alcohol were below 50% which justified more efforts to improve the catalytic performances by modifiying morphologies and hydrophilic and hydrophobic properties of the hybrid catalysts in upcoming work.

**Table 23.** Catalytic performances of various POM catalyst for oxidation of benzyl alcohol with $H_2O_2$ [a].

| Entry | Catalyst | Reaction System [b] | Conv. (%) | Sel. (%) [c] | $H_2O_2$-Efficiency (%) |
|-------|----------|---------------------|-----------|--------------|--------------------------|
| 1 | None | L-L | 3 | 99 | 10 |
| 2 | $[C_4mim]Br$ | L-L | 3 | 99 | 10 |
| 3 | $H_3PMo_{12}O_{40}$ | L-L | 22 | 99 | 48 |
| 4 | $H_4PMo_{11}VO_{40}$ | L-L | 20 | 99 | 45 |
| 5 | $H_5PMo_{10}V_2O_{40}$ | L-L | 12 | 99 | 33 |
| 6 | $[C_4mim]_3PMo_{12}O_{40}$ | L-L | 47 | 88 | 66 |
| 7 | $[C_4mim]_4PMo_{11}VO_{40}$ | L-L-S | 34 | 99 | 59 |
| 8 | $[C_4mim]_5PMo_{10}V_2O_{40}$ | L-L | 16 | 99 | 40 |

[a] Reaction conditions: Benzyl alcohol (30 mmol), catalyst (100 mg), $H_2O_2$ (30 wt%, 36 mmol), $H_2O$ (6 mL), 80 °C, 6 h.
[b] L (liquid, water phase), L (liquid, organic phase), and S (solid, catalyst). [c] Selectivity for benzaldehyde.

Catalytic activities of $Fe_3O_4@C$ materials in neat water were examined in the selective oxidation of alcohols using $H_2O_2$ under base-free conditions. Both aryl and alkyl alcohols as a comprehensive substrate scope were oxidized with high activity and selectivity over the B-600 materials (**24**) (Table 24) [58]. In addition, the magnetic catalyst could be easily removed using an external magnetic field and reused at least four times. Different primary and secondary benzylic alcohols with electron-donating or -withdrawing functional groups converted their corresponding substituted aldehyde or ketones into good to excellent yields (Entries 1–14) but difference in activities were seen which were attributed to their compositions and structures. Secondary benzylic alcohols such as 1-phenylethanol and derivatives were converted to the corresponding ketones in relatively lower yields as compared with primary benzylic alcohols due to the steric effect of the α-CH position in secondary alcohols (Entries 11–15).

**Table 24.** Oxidation of various alcohols by B-600 **24** in water [a].

| Entry | Substrate | Product | Conv. (%) [b] | Sel. (%) [b] |
|-------|-----------|---------|---------------|--------------|
| 1 | $4\text{-}OMeC_6H_4CH_2OH$ | $4\text{-}OMeC_6H_4CHO$ | 86 | 99 |
| 2 | $2\text{-}MeC_6H_4CH_2OH$ | $2\text{-}MeC_6H_4CHO$ | 93 | 99 |
| 3 | $4\text{-}MeC_6H_4CH_2OH$ | $4\text{-}MeC_6H_4CHO$ | 91 | 99 |
| 4 | $2\text{-}NH_2C_6H_4CH_2OH$ | $2\text{-}NH_2C_6H_4CHO$ | >99 | 99 |
| 5 | $4\text{-}COOMeC_6H_4CH_2OH$ | $4\text{-}COOMeC_6H_4CHO$ | 88 | 99 |
| 6 | $4\text{-}NO_2C_6H_4CH_2OH$ | $4\text{-}NO_2C_6H_4CHO$ | 84 | 99 |
| 7 | $4\text{-}FC_6H_4CH_2OH$ | $4\text{-}FC_6H_4CHO$ | 96 | 99 |

**Table 24.** *Cont.*

| Entry | Substrate | Product | Conv. (%) [b] | Sel. (%) [b] |
|-------|-----------|---------|---------------|--------------|
| 8 | 4-ClC$_6$H$_4$CH$_2$OH | 4-ClC$_6$H$_4$CHO | 94 | 99 |
| 9 | 3-ClC$_6$H$_4$CH$_2$OH | 3-ClC$_6$H$_4$CHO | 83 | 99 |
| 10 | 4-BrC$_6$H$_4$CH$_2$OH | 4-BrC$_6$H$_4$CHO | >99 | 99 |
| 11 | C$_6$H$_5$CH(OH)CH$_3$ | C$_6$H$_5$COCH$_3$ | 81 | 99 |
| 12 | 4-MeC$_6$H$_4$CH(OH)CH$_3$ | 4-MeC$_6$H$_4$COCH$_3$ | 82 | 99 |
| 13 | 4-ClC$_6$H$_4$CH(OH)CH$_3$ | 4-ClC$_6$H$_4$COCH$_3$ | 89 | 99 |
| 14 | 4-BrC$_6$H$_4$CH(OH)CH$_3$ | 4-BrC$_6$H$_4$COCH$_3$ | 91 | 99 |
| 15 | (C$_6$H$_5$)$_2$CHOH | (C$_6$H$_5$)$_2$CO | 81 | 99 |
| 16 | 4,4-(CH$_2$OH)$_2$C$_6$H$_4$ | 4,4-(CHO)$_2$C$_6$H$_4$ | 91 | 90 |
| 17 [c] | 2-Pyridinemethanol | Picolinaldehyde | 98 | 96 |
| 18 [c] | Furfuryl alcohol | Furfural | 90 | 99 |

[a] Reaction conditions: Substrate (0.5 mmol), B-600 **24** (10 mol% Fe), H$_2$O$_2$ (30 wt%, 1.5 mmol), H$_2$O (3 mL), 110 °C, 48 h. [b] Conversion and selectivity were determined by GC-MS with an external standard. [c] 60 h.

Recyclable heterogeneous catalyst β-CD grafted on lignin cross linked by epichlorohydrin (EPI) as crosslinking agent (L-β-CD) (**25**) were prepared and under mild reaction were used in the oxidation of BzOH to BzH conditions. The catalyst provided the solution selective oxidation of BzOH in high selectivity (>99%) and catalytic activity was not significantly decreased after five cycles. As illustrated in Table 25, (**25**) was applied for different benzyl alcohol oxidations to the corresponding BzHs. The results not only emphasized the catalytic power of the **25**/H$_2$O$_2$/NaHCO$_3$ system for the oxidation of various substituted benzyl alcohols in good yields (79% to 99%) and high selectivity (>99%), but also obviously showed the electronic and steric hindrance effect of substituent groups on the catalytic oxidation of substrates [59].

**Table 25.** Scope oxidation of alcohols catalyzed by L-β-CD **25** in water [a].

| Entry | Substrate | Product | Yield (%) [b] |
|-------|-----------|---------|---------------|
| 1 | 4-OMeC$_6$H$_4$CH$_2$OH | 4-OMeC$_6$H$_4$CHO | 99 |
| 2 | 3-OMeC$_6$H$_4$CH$_2$OH | 3-OMeC$_6$H$_4$CHO | 98 |
| 3 | 2-OMeC$_6$H$_4$CH$_2$OH | 2-OMeC$_6$H$_4$CHO | 97 |
| 4 | 2-MeC$_6$H$_4$CH$_2$OH | 2-MeC$_6$H$_4$CHO | 99 |
| 5 | 3-MeC$_6$H$_4$CH$_2$OH | 3-MeC$_6$H$_4$CHO | 97 |
| 6 | 4-MeC$_6$H$_4$CH$_2$OH | 4-MeC$_6$H$_4$CHO | 96 |
| 7 | 4-NO$_2$C$_6$H$_4$CH$_2$OH | 4-NO$_2$C$_6$H$_4$CHO | 83 |
| 8 | 3-NO$_2$C$_6$H$_4$CH$_2$OH | 3-NO$_2$C$_6$H$_4$CHO | 81 |
| 9 | 2-NO$_2$C$_6$H$_4$CH$_2$OH | 2-NO$_2$C$_6$H$_4$CHO | 79 |
| 10 | (C$_6$H$_5$)$_2$CHOH | (C$_6$H$_5$)$_2$CO | Trace |

[a] Reaction condition: Substrate (1 mmol), L-β-CD **25** (0.8 g), NaHCO$_3$ (1.5 mmol), H$_2$O$_2$ (30%, 2.5 mL), H$_2$O (25 mL), 60 °C, 240 min. [b] GC yield.

Yadollahi [60] reported the application of sodium and potassium salts of a sandwich-type tetracobalt tungstophosphate, [Co$_4$(H$_2$O)$_2$(PW$_9$O$_{34}$)$_2$]$^{10-}$ catalysts for the selective oxidation of alcohols with H$_2$O$_2$ in water. In general, the Na$_{10}$[Co$_4$(H$_2$O)$_2$(PW$_9$O$_{34}$)$_2$]·27H$_2$O complex (**26**) presented better activity and was recycled for five times. Results in Table 26 showed that alcohols with electron-donating substituent converted to their corresponding aldehydes even faster than benzyl alcohol [60].

**Table 26.** Selective oxidation of different alcohols to the corresponding aldehyde using Na-Co-POM **26** [a].

| Entry | Substrate | Product | t/h | Yield (%) [b] |
|-------|-----------|---------|-----|---------------|
| 1 | $C_6H_5CH_2OH$ | $C_6H_5CHO$ | 3 | 97 |
| 2 | $4\text{-}NO_2C_6H_4CH_2OH$ | $4\text{-}NO_2C_6H_4CHO$ | 4 | 95 |
| 3 | $3\text{-}NO_2C_6H_4CH_2OH$ | $3\text{-}NO_2C_6H_4CHO$ | 4.15 | 95 |
| 4 | $2\text{-}NO_2C_6H_4CH_2OH$ | $2\text{-}NO_2C_6H_4CHO$ | 4.30 | 95 |
| 5 | $4\text{-}FC_6H_4CH_2OH$ | $4\text{-}FC_6H_4CHO$ | 6 | 98 |
| 6 | $4\text{-}ClC6H4CH_2OH$ | $4\text{-}ClC_6H_4CHO$ | 6.15 | 97 |
| 7 | $4\text{-}BrC6H4CH2OH$ | $4\text{-}BrC6H4CHO$ | 6.45 | 98 |
| 8 | $2\text{-}ClC_6H_4CH2OH$ | $2\text{-}ClC_6H_4CHO$ | 6.15 | 92 |
| 9 | $4\text{-}MeC_6H_4CH_2OH$ | $4\text{-}MeC_6H_4CHO$ | 2 | 97 |
| 10 | $2\text{-}MeC_6H_4CH_2OH$ | $2\text{-}MeC_6H_4CHO$ | 2.10 | 97 |
| 11 | $4\text{-}OMeC_6H_4CH_2OH$ | $4\text{-}OMeC_6H_4CHO$ | 2 | 99 |
| 12 | $2\text{-}OMeC_6H_4CH_2OH$ | $2\text{-}OMeC_6H_4CHO$ | 2.05 | 99 |
| 13 | $3\text{-}OMeC_6H_4CH_2OH$ | $3\text{-}OMeC_6H_4CHO$ | 2.30 | 99 |
| 14 | $C_6H_5CH(OH)Et$ | $C_6H_5COEt$ | 3 | 96 |

[a] Reaction conditions: Substrate (1 mmol), Na-Co-POM **26** (1 μmol), $H_2O_2$ (30%, 9.8 mmol), $H_2O$ (3 mL), reflux, 2 to 6.15 h. [b] Yield refers to GC yield.

Direct solvothermal synthesis or post-synthetic modification were applied for the preparation of ECH-modified $Fe_3O_4$ microspheres in ethylene glycol (MEG). As summarized in Table 27, the catalyst was successfully performed for the selective oxidation of BzOH to BzH in water with $H_2O_2$ [29]. The reaction was sensitive to both reaction temperature and the molar ratio of $H_2O_2$ to BzOH. Magnetic $Fe_3O_4$ microspheres (**27**) were recoverable and reusable at least five times without loss of catalytic selectivity and activity after $NaBH_4$ reduction and ECH modification. In light of the above, surface modification with organic groups increased the catalytic performance of $Fe_3O_4$ and exhibited suitable interactions with $H_2O_2$.

**Table 27.** Catalytic oxidation of benzyl alcohol to benzaldehyde with $H_2O_2$ [a].

| Entry | Catalyst | BzOH Conv. (%) | BzH yield (%) | Sel. (%) [b] |
|-------|----------|----------------|---------------|--------------|
| 1 | None | 0 | 0 | 0 |
| 2 | ECH (1 g) | 2 | 2 | 100 |
| 3 | $Fe_3O_4$-blank | 8.3 | 7.9 | 94.8 |
| 4 | $Fe_3O_4$-ECH-P-1g | 25.5 | 25.2 | 98.9 |
| 5 | $Fe_3O_4$-ECH-P-3g | 39.2 | 33.2 | 84.8 |
| 6 | $Fe_3O_4$-ECH-P-4g | 34.7 | 32.6 | 93.9 |
| 7 | $Fe_3O_4$-ECH-P-5g | 37.7 | 32.5 | 86.3 |
| 8 | $Fe_3O_4$-ECH-D | 36.6 | 34.2 | 93.5 |
| 9 [c] | $Fe_3O_4$-ECH-D | 23.1 | 22.4 | 96.9 |
| 10 [d] | $Fe_3O_4$-ECH-D | 1.7 | 1.7 | 100 |
| 11 [e] | $Fe_3O_4$-ECH-D | 23.4 | 21.7 | 92.9 |

[a] Reaction conditions: Benzyl alcohol (40 mmol), catalyst (0.2 g), $H_2O_2$ (80 mmol), $H_2O$ (8 mL), 100 °C, 1.5 h. [b] Selectivity based on the conversion of BzOH. [c] Reaction temperature, 90 °C. [d] Reaction temperature, 80 °C. [e] $H_2O_2$, 40 mmol added.

Benzyl alcohol oxidation catalyzed by gold nanoparticles supported on gamma alumina ($\gamma$-Al$_2$O$_3$) were prepared using a deposition-precipitation method. The Nano Au/c-Al$_2$O$_3$ (**28**) selectivity provided benzaldehyde (over 98%) under environment-friendly conditions and could be recovered after three times. More investigation showed that changes in support material, even the same type of alumina ($\gamma$-Al$_2$O$_3$), had a significant effect on both catalytic activity and physical–chemical of the catalysts. Gold particles were responsible for catalytic efficiency since in the tests conducted in like conditions using no catalyst or $\gamma$-alumina catalyst caused a very low conversion of benzyl alcohol and yield of benzaldehyde (<1%) ()(Table 28) [61]. For all the supports, both the conversion of benzyl alcohol and the selectivity of benzaldehyde also increased. As a result, for the 2% Au/$\gamma$-Al$_2$O$_3$ catalysts, higher yields of benzaldehyde were achieved8)(Table 29) [61]. While the benzaldehyde selectivities were high in the S723-supported catalysts, the conversion of the benzyl alcohol was comparatively low.

**Table 28.** Effect of Au catalyst concentration on the oxidation of benzyl alcohol [a].

| Entry | Catalyst | Conv. (%) | Yield (%) | Sel. (%) |
|-------|----------|-----------|-----------|----------|
| 1 | No catalyst | 1.8 | - | - |
| 2 | S823 | 3.4 | 0.8 | 25 |
| 3 | 1%Au/S823 | 24.7 | 12.4 | 50.2 |
| 4 | 2%Au/S823 | 29.6 | 18.4 | 61.9 |
| 5 | 4%Au/S823 | 33.3 | 21.3 | 63.9 |

[a] Reaction conditions: Benzyl alcohol (5.3 g), catalyst (0.3 g), molar ratio BzOH/H$_2$O$_2$ (1:1.3), 80 °C, 2 h.

**Table 29.** Effect of nanogold on the oxidation of benzyl alcohol over various support materials [a].

| Entry | Type of Al$_2$O$_3$ Support | Conv. (%) | | Yield (%) | | Sel. (%) | |
|-------|-----------------------------|-----------|------|-----------|------|----------|------|
| | | 1%Au | 2%Au | 1%Au | 2%Au | 1%Au | 2%Au |
| 1 | M | 37.1 | 33..8 | 11.5 | 16.7 | 31 | 49.4 |
| 2 | S823 | 24.7 | 29.6 | 12.4 | 18.4 | 50.2 | 61.9 |
| 3 | S723 | 20.4 | 18.1 | 10.9 | 15.6 | 53.7 | 86.3 |

[a] Reaction conditions: Benzyl alcohol (5.3 g), catalyst (0.3 g), molar ratio BzOH/H$_2$O$_2$ (1:1.3), 80 °C, 2 h.

Platinum nanoparticles supported on Ca(Mg)-ZSM-5 (**29**) were a highly stable and selective catalyst for the room-temperature oxidation of alcohols in water. In situ EPR measurement and the radical trapping technique demonstrated that •OH radicals generated by O–O cleavage bond of H$_2$O$_2$ intermediate as the rate determining step, contributed to the H abstraction of the $\alpha$-C–H bond of alcohols to provide aldehydes/ketones. PtNPs on different supports applied for the oxidation reactions of alcohols in aqueous solution at room temperature and the results are summarized in Table 30 [62]. From the results, the basic zeolites, such as Mg-ZSM-5 and Ca-ZSM-5, exhibited the optimal performance during benzyl alcohol oxidation, with 95% conversion and 99% of selectivity at a carbon balance of 98%, (Entries 1 and 2). The PtNPs catalytic power extremely depended on the size of Pt which was sized similar to the PtNPs/Ca-ZSM-5 catalyst with a Pt mean size of ~5.8 nm, and when prepared by the impregnation method, could only convert 89% of the benzyl alcohol in 20 h of reaction (Entry 2). In contrast, high basicity supports, such as CaO and MgO, significantly suppressed the oxidation process (Entries 6 and 7). The PtNPs on inert supports, such as silica gel, exhibited some activity but did not exceed the conversion upper of 70%, even with a prolonged reaction time (Entry 5). Moreover, the application of AuNPs or PdNPs, instead of PtNPs, yielded no benzaldehyde product

under identical experimental conditions which could be attributed to the unique catalytic property of PtNPs (Entries 8 and 9). In addition, by replacing water with other solvents, only a slight conversion of benzyl alcohol was observed (Entries 13–15) which indicated the important role of water in the PtNP catalyzed alcohol oxidation procedure. The catalyst could be steady and reused even after four reaction cycles with no Pt leaching and less than 1% Ca leaching.

**Table 30.** The catalytic property of PtNPs on different supports [a].

| Entry | Catalyst | Substrate [b] | Solvent | Conv. (%) | Sel. (%) |
|-------|----------|---------------|---------|-----------|----------|
| 1 | Pt/Ca-ZSM-5 | BA | $H_2O$ | 98.8 | 99 |
| 2 | Pt/Mg-ZSM-5 | BA | $H_2O$ | 95.5 | 98 |
| 3 | Pt/Ca-ZSM-5 [c] | BA | $H_2O$ | 89 | 96 |
| 4 | Pt/HZSM-5 | BA | $H_2O$ | 65 | 87 |
| 5 | Pt/SiO$_2$ | BA | $H_2O$ | 64 | 87 |
| 6 | Pt/CaO | BA | $H_2O$ | 0 | - |
| 7 | Pt/MgO | BA | $H_2O$ | 0 | - |
| 8 | Au/Ca-ZSM-5 | BA | $H_2O$ | 0 | - |
| 9 | Pd/Ca-ZSM-5 | BA | $H_2O$ | 0 | - |
| 10 | Pt/Mg-ZSM-5 | CHA | $H_2O$ | 20.7 | 85.5 |
| 11 | Pt/Ca-ZSM-5 | CHA | $H_2O$ | 21.8 | 82 |
| 12 | Pt/SiO$_2$ | CHA | $H_2O$ | 5.1 | 90 |
| 13 | Pt/Ca-ZSM-5 | BA | $CH_3CN$ | <1 | - |
| 14 | Pt/Ca-ZSM-5 | BA | EtOH | <1 | - |
| 15 | Pt/Ca-ZSM-5 | BA | $CCl_4$ | <1 | - |

[a] Reaction conditions: A mixture of 50 mg catalyst with a unified 1 wt% Pt and 20 mM alcohol in various solvents was stirred at room temperature (25 °C) in open air for 20 h. [b] BA, benzyl alcohol; CHA, cyclohexanol. [c] Impregnation method was used to prepare the catalyst.

An acid-base reaction using a Keggin-type phosphotungstic acid and TEA was applied for the synthesis of several triethylamine (TEA) salts of phosphotungstic acid ((TEAH)nH$_{3-n}$PW$_{12}$O$_{40}$ (n = 1, 2, 3)). These catalysts used for the alcohol oxidation reactions, and their catalytic activity, selectivity, and recovery rate are listed in Table 31 [63]. The (TEAH)H$_2$PW$_{12}$O$_{40}$ (**30**) catalyst showed the best results, with 99.6% conversion of benzyl alcohol and 100% of selectivity to benzaldehyde under optimized reaction conditions. The activity and selectivity were also essentially unchanged even in the third and the fifth cycles.

**Table 31.** Activity and selectivity in the oxidation of benzyl alcohol with the catalysts [a].

| Entry | Catalyst | Conv. of BA (%) [c] | Sel. for BzH (%) [c] | Yield of Benzoic Acid | Recovery Rate of Catalyst (%) |
|-------|----------|---------------------|----------------------|-----------------------|-------------------------------|
| 1 | Blank | 17.9 | 100 | 0 | - |
| 2 | $H_3PW_{12}O_{40}$ | 100 | 92.4 | 7.6 | 0 |
| 3 | $[TEAH]_3PW_{12}O_{40}$ | 95 | 100 | 0 | 77 |
| 4 | $[TEAH]3P W_{12}O_{40}+H_2SO_4$ [b] | 84.4 | 100 | 0 | 62 |
| 5 | $[TEAH]_2HPW_{12}O_{40}$ | 97.5 | 100 | 0 | 58 |
| 6 | $[TEAH]H_2PW_{12}O_{40}$ | 99.6 | 100 | 0 | 53 |
| 7 | $[TEAH]H_2PW_{12}O_{40} + H_2SO_4$ [b] | 86.7 | 100 | 0 | 42 |
| 8 | $[TEAH]H_2PW_{12}O_{40}$ (cycle 3) | 99.8 | 100 | 0 | - |

**Table 31.** *Cont.*

| Entry | Catalyst | Conv. of BA (%) [c] | Sel. for BzH (%) [c] | Yield of Benzoic Acid | Recovery Rate of Catalyst (%) |
|---|---|---|---|---|---|
| 9 | [TEAH]H$_2$PW$_{12}$O$_{40}$ (cycle 5) | 99.5 | 100 | 0 | - |

[a] Reaction conditions: Benzyl alcohol (1 mL, 9.6 mmol), catalyst (0.04 mmol), H$_2$O$_2$ (30%, 1.2 mL, 11.8 mmol), H$_2$O (10 mL), 100 °C, 3 h. [b] H$_2$SO$_4$ (1 mol/L) was added to adjust the acidity from pH ≈ 5 to pH = 1 before heating the system. [c] BA, benzyl alcohol; BzH, benzlaldehyde.

An iron(III) complex formulated as L(14,28-[1,3-diiminoisoindolinato]phthalocyaninato)Fe(III) (**31**) in which L was a labile axial ligand was synthesized. The efficiency of the prepared catalyst was investigated by the oxidations of primary and secondary benzylic alcohols with both hydrogen peroxide and *tert*-butyl hydroperoxide (TBHP). Benzyl alcohol, 4-chlorobenzyl alcohol, 1-phenylethanol, and diphenylmethanol (benzhydrol), as illustrated in Table 32, were effectively oxidized without needing highly problematic oxidants or adding organic solvent. In primary alcohol oxidation (benzyl alcohol and 4-chlorobenzyl alcohol) no significant carboxylic acid as overoxidation products were observed. Moreover, secondary alcohols produced ketones with excellent selectivity [64].

**Table 32.** Summary of results from catalysis and control experiments [a].

| Entry | Substrate | Product | t/min | Yield (%) [e] | TON [f] | TOF (h$^{-1}$) [g] |
|---|---|---|---|---|---|---|
| 1 | C$_6$H$_5$CH$_2$OH | C$_6$H$_5$CHO | 15<br>15 [b] | 1.6<br>0.03 | 84<br>- | 340<br>- |
| 2 | 4-ClC$_6$H$_4$CH$_2$OH | 4-ClC$_6$H$_4$CHO | 15 [c]<br>15 [b] | 40<br>0.41 | 940<br>- | 3800<br>- |
| 3 | C$_6$H$_5$CH(OH)CH$_3$ | C$_6$H$_5$COCH$_3$ | 6<br>6 [b] | 4.9<br>0.085 | 230<br>- | 2300<br>- |
| 4 | (C$_6$H$_5$)$_2$CHOH | (C$_6$H$_5$)$_2$CO | 10 [d]<br>10 [b] | 3.7<br>0.77 | 210<br>- | 1200<br>- |

[a] All reactions were run in magnetically stirred round-bottomed flasks open to the air at room temperature. Catalyst **31** (1.0 to 2.8 mg, 1.2 to 3.5 μmol) was dissolved in the alcohol substrate before adding the oxidant as an aqueous solution. Caution: solutions containing significant concentrations of hydrogen peroxide and metal-containing compounds are potentially dangerous, and reactions should be carried out behind adequate protection. Optimal reaction times for hydrogen peroxide oxidations were determined by quenching the reaction at various points with sodium thiosulfate. [b] No catalyst. [c] 72 °C. [d] 70 °C. [e] Oxidized product yields on the basis of total oxidant added. [f] TON, moles product/moles catalyst. Products were identified and quantified by HPLC vs. external standard. The catalyst **31** used in this study is prepared in crystalline form, where the composition of the crystals can vary slightly. In the crystallographically determined structure, ligand L is observed to be a mixture of methanol and water, with

two co-crystallized methanol molecules, also present for every complex bearing a methanol ligand, whereas three co-crystallized methanol molecules are present for every complex bearing a water ligand. In a desire to report catalyst performance conservatively, in light of this issue, we calculated TON values based on an assumption that the catalyst had 100% methanol as the L ligand, which along with the two co-crystallized methanol molecules gave a formula weight of 806.65, less than that for the case where L = water and three co-crystallized methanol molecules were present, in which case the formula weight was 824.66. This results in slightly under-reported turnover numbers (TON), assuming a mixture of the two species is actually present, because the number of moles of catalyst actually present in the reaction mixtures is slightly less than the number used in TON calculations. [g] TOF, turnovers per hour, average over entire reaction period.

A straightforward method was suggested for the oxidation of alcohols to their corresponding carbonyl compounds using nanomagnetic $Fe_3O_4$ (**32**) catalyst in water by $H_2O_2$ and the results are summarized in Table 33 [65]. High yields and the best selectivity (>99%) were obtained by producing aldehyde without observing overoxidation of aldehydes to parallel carboxylic acids [66].

**Table 33.** Oxidation of various alcohols by using $H_2O_2$ in the presence of nanomagnetic $Fe_3O_4$ **32** catalyst in water at 50 °C [a].

| Entry | Substrate | Product | t/min | Yield (%) [b] |
|-------|-----------|---------|-------|---------------|
| 1 | $C_6H_5CH_2OH$ | $C_6H_5CHO$ | 20 | 91 |
| 2 | $4\text{-}ClC_6H_4CH_2OH$ | $4\text{-}ClC_6H_4CHO$ | 30 | 89 |
| 3 | $3\text{-}ClC_6H_4CH_2OH$ | $3\text{-}ClC_6H_4CHO$ | 35 | 87 |
| 4 | $2\text{-}ClC_6H_4CH_2OH$ | $2\text{-}ClC_6H_4CHO$ | 40 | 90 |
| 5 | $2,5\text{-}ClC_6H_3CH_2OH$ | $2,5\text{-}ClC_6H_3CHO$ | 45 | 87 |
| 6 | $2\text{-}BrC_6H_4CH_2OH$ | $2\text{-}BrC_6H_4CHO$ | 30 | 85 |
| 7 | $3\text{-}BrC_6H_4CH_2OH$ | $3\text{-}BrC_6H_4CHO$ | 35 | 88 |
| 8 | $4\text{-}BrC_6H_4CH_2OH$ | $4\text{-}BrC_6H_4CHO$ | 35 | 85 |
| 9 | $4\text{-}FC_6H_4CH_2OH$ | $4\text{-}FC_6H_4CHO$ | 40 | 88 |
| 10 | $3\text{-}FC_6H_4CH_2OH$ | $3\text{-}FC_6H_4CHO$ | 40 | 85 |
| 11 | $4\text{-}NO_2C_6H_4CH_2OH$ | $4\text{-}NO_2C_6H_4CHO$ | 120 | 85 |
| 12 | $3\text{-}NO_2C_6H_4CH_2OH$ | $3\text{-}NO_2C_6H_4CHO$ | 120 | 81 |
| 13 | $2\text{-}NO_2C_6H_4CH_2OH$ | $2\text{-}NO_2C_6H_4CHO$ | 120 | 82 |
| 14 | $4\text{-}OMeC_6H_4CH_2OH$ | $4\text{-}OMeC_6H_4CHO$ | 15 | 90 |
| 15 | $(C_6H_5)_2CHOH$ | $(C_6H_5)_2CO$ | 30 | 88 |
| 16 | $C_6H_5CH(OH)COC_6H_5$ | $(C_6H_5)_2(CO)_2$ | 120 | 85 |

[a] Reaction conditions: Substrate (1 mmol), nanomagnetic $Fe_3O_4$ **32** (10 mol%, 23 mg), $H_2O_2$ (1.2 mmol), $H_2O$ (1 mL), 50 °C, 15 to 120 min. [b] Yield refers to isolated products. The products were characterized from their spectra data (IR) and compared with authentic samples.

Grafting the Cu(II) Schiff base complex onto the channels of mesoporous silica material SBA-15 (Cu(II)-Shiff base-SBA-15 (**33**)) was conducted by Ma et al. [67], and provided an effective catalyst for the selective oxidation of alcohols in water phase with hydrogen peroxide. Benzyl alcohol converted 98.5% with 100% of the selectivity to benzyl aldehyde. Other substituted benzyl alcohol was also successfully oxidized under the optimal conditions in which the substituted groups impacted the catalytic activity (Table 34).

**Table 34.** Selective oxidation of other primary alcohols [a].

| Entry | Substrate | Product | t/h | Conv. (%) | Sel. (%) |
|-------|-----------|---------|-----|-----------|----------|
| 1 | 4-MeC$_6$H$_4$CH$_2$OH | 4-MeC$_6$H$_4$CHO | 3 | 43.9 | >99 |
| 2 | 4-OMeC$_6$H$_4$CH$_2$OH | 4-OMeC$_6$H$_4$CHO | 3 | 30.5 | >99 |
| 3 | 4-ClC$_6$H$_4$CH$_2$OH | 4-ClC$_6$H$_4$CHO | 2 | 72.7 | >99 |
| 4 | 4-NO$_2$C$_6$H$_4$CH$_2$OH | 4-NO$_2$C$_6$H$_4$CHO | 2 | 48.6 | >99 |

[a] Reaction conditions: Substrate (2 mmol, 0.2 mL), Cu(II)-Schiff base-SBA-15 **33** (0.02 g), H$_2$O$_2$ (30%, 2 mL), H$_2$O (7.5 to 15 mL), reflux, 2 to 3 h.

Malakooti et al. [68] reported the catalytic activity of the Fe$_2$O$_3$/SBA-15 (**34**) catalyst in oxidation of alcohols. As compared with different heterogeneously catalyzed procedures [7,69] the catalyst showed good activity, and therefore the catalyst was successfully examined for the oxidation of a variety of primary and secondary alcohols using H$_2$O$_2$ (Table 35). Benzyl alcohols were converted to their corresponding aldehydes with excellent yields and also alcohols containing heteroatom such as furfuryl alcohol, hindered substituted alcohols, and secondary benzylic alcohols were oxidized. No further oxidation of aldehyde products to carboxylic acid were seen and the selectivity of the catalyst was also confirmed. The recovered catalyst was reused for at least six successive runs with only a few decreasing in conversion.

**Table 35.** Oxidation of alcohols to aldehyde using Fe$_2$O$_3$/SBA-15 **34** catalyst in water in the presence of H$_2$O$_2$ [a].

| Entry | Substrate | Product | t/h | Conv. (%) |
|-------|-----------|---------|-----|-----------|
| 1 | C$_6$H$_5$CH$_2$OH | C$_6$H$_5$CHO | 3.5 | 95 |
| 2 | C$_6$H$_5$CH(OH)CH$_3$ | C$_6$H$_5$COCH$_3$ | 4 | 80 |
| 3 | Furfuryl alcohol | Furfural | 2 | 100 |
| 4 | 2-OHC$_6$H$_4$CH$_2$OH | 2-OHC$_6$H$_4$CHO | 2.5 | 100 |
| 5 | 4-OMeC$_6$H$_4$CH$_2$OH | 4-OMeC$_6$H$_4$CHO | 2 | 100 |
| 6 | 4-NO$_2$C$_6$H$_4$CH$_2$OH | 4-NO$_2$C$_6$H$_4$CHO | 3 | 100 |
| 7 | 4-ClC$_6$H$_4$CH$_2$OH | 4-ClC$_6$H$_4$CHO | 4.5 | 90 |
| 8 | 2-ClC$_6$H$_4$CH$_2$OH | 2-ClC$_6$H$_4$CHO | 4.5 | 80 |
| 9 | 2,4-ClC$_6$H$_3$CH$_2$OH | 2,4-ClC$_6$H$_3$CHO | 2 | 95 |
| 10 | C$_6$H$_5$CH(OH)Et | C$_6$H$_5$COEt | 4 | 40 |
| 11 | (C$_6$H$_5$)$_2$CHOH | (C$_6$H$_5$)$_2$CO | 6 | 45 |
| 12 | C$_6$H$_5$CH(OH)COC$_6$H$_5$ | (C$_6$H$_5$)$_2$(CO)$_2$ | 6 | 85 |

[a] Reaction conditions: Substrate (1 mmol), Fe$_2$O$_3$/SBA-15 **34** (0.02 g, 1.64 mol% Fe), H$_2$O$_2$ (2 mmol, 0.2 mL), H$_2$O (2 mL), 80 °C, 2 to 6 h.

Simple and inexpensive iron(III)-benzenetricarboxylate (Fe-BTC) (**35**) metal-organic gel catalyst, which was previously synthesized [70], was applied in a catalytic system for the oxidation of alcohols with H$_2$O$_2$ oxidant in water with no proof of other side products (Table 36) [71].

**Table 36.** Fe-BTC gel **35** catalyzed oxidation of various alcohols [a].

| Entry | Substrate | Product | t/min | Conv. (%) | Sel. (%) | TON [b] |
|-------|-----------|---------|-------|-----------|----------|---------|
| 1 | $C_6H_5CH_2OH$ | $C_6H_5CHO$ | 180 | 98 | >99 | 504 |
| 2 | $C_6H_5CH(OH)CH_3$ | $C_6H_5COCH_3$ | 30 | 56 | >99 | 288 |
| | | | 180 | 100 | >99 | 514 |

[a] Reaction conditions: Substrate (9.26 mmol), dried Fe-BTC gel **35** (0.01 g, 2 mg Fe, 36 μmol Fe, 18 μmol $[Fe_2(C_9H_3O_6)(NO_3)_3(H_2O)_3$ catalyst], molar ratio alcohol/Fe = 9.26/0.036 = 257, $H_2O_2$ (20 mL of 10% $H_2O_2$ (c = 3.07 mmol/mL) at an addition rate of 0.12 mL min$^{-1}$, $H_2O$ (5 mL), molar ratio $H_2O_2$/alcohol (6.6), 30 to 180 min. [b] TON = [mol product/mol catalyst], the turnover frequency TOF [mol product/(mol catalyst x time)] is obtained by dividing TON through time.

Two cobalt (II) and cobalt (III) complexes of a terpyridine based ligand, (40-(2-thienyl)-2,2′,6′,2″-terpyridine (L)), were prepared in which each complex had two units of the tridentate ligand. The cobaltous complex and the cobaltic complex showed the formula $[Co(L)_2](NO_3)_2.2CH_3OHH_2O$ (**36**) and $[Co(L)_2](NO_3)_3.2CH_3OH$ (**37**), respectively. The aromatic alcohol oxidation reactions in the presence of catalysts was performed and the results are summarized in Table 37 [72]. As it was clear, in oxidation by hydrogen peroxide, the alcohols with more solubility in an aqueous media provided more reactivity towards and needed less reaction times. It is worthwhile mentioning that the cobaltous species was more effective than the cobalt (III) catalyst for alcohol oxidation reactions.

**Table 37.** Oxidation of a wide range of alcohols using the Co(II) and Co(III) complexes (**36**, **37**) as catalyst in water at room temperature [a,b].

| Entry | Substrate | Product | Co(II) catalyst | | Co(III) catalyst | |
|-------|-----------|---------|-----------------|---|------------------|---|
| | | | Conv. (%) | Carbonyl Compound: Carboxylic Acid/Hydroperoxide | Conv. (%) | Carbonyl Compound: Carboxylic Acid/Hydroperoxide |
| 1 | $C_6H_5CH_2OH$ | $C_6H_5CHO$ | 94 | 13:1 | 34 | 6:1 |
| 2 | $4\text{-}OMeC_6H_4CH_2OH$ | $4\text{-}OMeC_6H_4CHO$ | 95 | 15:1 | 37 | 12:1 |
| 3 | $4\text{-}ClC_6H_4CH_2OH$ | $4\text{-}ClC_6H_4CHO$ | 73 | 9:1 | 24 | 8:1 |
| 4 | 1-Indanol | 1-Indanone | 91 | 31:1 | 30 | 22:1 |
| 5 | Furfuryl alcohol | Furfural | 89 [c] | 5:1 | 42 | 4:1 |
| 6 | 2-Thiophenemethanol | 2-Thiophenecarboxaldehyde | 87 [c] | 8:1 | 49 | 5:1 |
| 7 | $(C_6H_5)_2CHOH$ | $(C_6H_5)_2CO$ | 93 [d] | 19:1 | 36 | 17:1 |
| 8 | $4,4\text{-}(OMe)_2(C_6H_4)_2CHOH$ | $4,4\text{-}(OMe)_2(C_6H_4)_2CO$ | 96 [d] | 22:1 | 39 | 19:1 |
| 9 | $2\text{-}FC_6H_4CH(OH)C_6H_5$ | $2\text{-}FC_6H_4COC_6H_5$ | 94 [e] | 19:1 | 37 | 18:1 |

[a] Reaction conditions: Substrate/catalyst/oxidant (20:1:60), $H_2O$ (2 mL), room temperature, 8 h. [b] The conversion was calculated based on the starting substrate, by using GC and with a comparison of authentic samples. [c] The reaction time was 2 h. [d] The reaction time was 12 h. [e] The reaction was 6 h.

Efficient usage of Lacunary Keggin-tungstoborate of $K_8[BW_{11}O_{39}H]\cdot13H_2O$ (**38**) catalyst was reported for the first time in an aqueous/oil system for the oxidation of alcohols [73]. Benzyl alcohol oxidation provided benzaldehyde in high conversion and selectivity and secondary alcohols delivered high yields of ketones. The molar ratio of $H_2O_2$/benzyl alcohol was optimized (1:1) because a greater

content of oxidant decreased the activity and selectivity of the oxidation product which benzoic acid produced (Tables 38 and 39). The catalyst could be reused after four consecutive cycles of the reaction.

**Table 38.** Oxidation of benzyl alcohol catalyzed by $K_8[BW_{11}O_{39}H].13H_2O$ **38** with $H_2O_2$ [a].

| Entry | $H_2O_2$/BzOH (mol/mol) | Sel. of BzH (mol%) | BzH Yield (mol%) |
|---|---|---|---|
| 1 [b] | 1/2 | 98 | 98 |
| 2 | 1/1 | 90 | 86 |
| 3 | 2/1 | 83 | 81 |
| 4 | 3/1 | 80 | 79 |
| 5 | 4/1 | 78 | 77 |
| 6 | 5/1 | 61 | 61 |

[a] Reaction conditions: Benzyl alcohol (1 mmol), $K_8[BW_{11}O_{39}H]·13H_2O$ **38** (0.015 mmol), $H_2O_2$ (30%), $H_2O$ (3 mL), 90 °C, 6 h, the conversion was based on benzyl alcohol. [b] $H_2O_2$ (1 mmol), benzyl alcohol (2 mmol), the conversion was based on $H_2O_2$.

**Table 39.** Oxidation of various alcohols catalyzed by $K_8[BW_{11}O_{39}H].13H_2O$ **38** with $H_2O_2$ [a].

| Entry | Substrate | Product | t/h | Conv. (mol%) | Sel. (mol%) | $H_2O_2$ Efficiency (%) |
|---|---|---|---|---|---|---|
| 1 | $C_6H_5CH(OH)CH_3$ | $C_6H_5COCH_3$ | 3 | 99 | 99 | 67 |
| 2 | $C_6H_5CH_2OH$ | $C_6H_5CHO$ $C_6H_5COOH$ | 6 6 | 98 98 | 83 16 | 67 |

[a] Reaction conditions: Substrate (1 mmol), $K_8[BW_{11}O_{39}H].13H_2O$ **38** (0.015 mmol), $H_2O_2$ (30% aq), $H_2O$ (3 mL), 90 °C, 3 to 6 h, the conversion was based on alcohols, the selectivity was based on ketone, aldehyde, or acid. $H_2O_2$ efficiency (%) = products (mol)/consumed $H_2O_2$ (mol) × 100.

Using easily prepared water-soluble POMs, $K_8[\gamma\text{-SiW}_{10}O_{36}]·13H_2O$ (**39**) precatalyst for the selective oxidation of alcohols were reported for the first time. Benzyl alcohol, 1-phenylethanol, and benzhydrol, etc. [74] as activated benzylic alcohols with 30% $H_2O_2$ at 90 °C were selectively oxidized to the corresponding ketone in high yields (Table 40) and the catalyst was recycled five times. With the molar ratio of $H_2O_2$ to benzyl alcohol 5:1, benzyl alcohol was absolutely converted to benzoic acid after 7 h at 90 °C (Entry 1). By decreasing the temperature of the reaction from 90 to 70 °C and the molar ratio of $H_2O_2$ to benzyl alcohol 1:1, benzaldehyde was obtained as the only oxidation product (Entry 2).

To gain more understanding of the catalytic system, the reactions were performed at 20 °C and the results are shown in Table 41. Dropping the reaction temperature from 90 to 20 °C was needed to achieve good yield, longer time (even some days), and more amount of catalyst (Entries 1 and 2). These data indicated that the high reaction temperature and the good water-solubility of alcohols provided high yields of products.

Keggin-type heteropolyacids (**40**) and hydrogen peroxide as a multiphase system were applied for the selective oxidation of alcohols to ketones or aldehydes (Table 42) with no appreciable detection of higher oxidation state by-product in most cases (Table 42). In the case of benzyl alcohols and 4-chloro

benzyl alcohol, the formation of small amounts of benzoic and 4-chlorobenzoic acid due to the addition of hydrogen peroxide twice was decomposed rapidly under homogeneous conditions (Table 42, Entries 6 to 10) [75].

**Table 40.** Selective oxidation of alcohol catalyzed by $K_8[\gamma\text{-SiW}_{10}O_{36}]\cdot 13H_2O$ **39** with $H_2O_2$ at 90 °C [a].

$$K_8[\gamma\text{-SiW}_{10}O_{36}].13H_2O \text{ **39** (6.7 μmol)}$$

$$H_2O_2 \text{ (30%, 5 mmol ), } H_2O \text{ (2 mL)}$$

90 °C, 7-8 h

| Entry | Substrate | Product | t/h | Conv. (mol%) | Yield (mol%) |
|---|---|---|---|---|---|
| 1 | $C_6H_5CH_2OH$ | $C_6H_5CO_2H$ | 8 | 100 | 100 |
| 2 [b] | $C_6H_5CH_2OH$ | $C_6H_5CHO$ | 7 | 84 | 84 |
| 3 | $C_6H_5CH(OH)CH_3$ | $C_6H_5COCH_3$ | 7 | 100 | 100 |
| 4 | $(C_6H_5)_2CHOH$ | $(C_6H_5)_2CO$ | 7 | 100 | 100 |

[a] Reaction conditions: Substrate (1 mmol), $K_8[\gamma\text{-SiW}_{10}O_{36}].13H_2O$ **39** (6.7 μmol), $H_2O_2$ (30%, 5 mmol), $H_2O$ (2 mL), 90 °C (oil bath temperature), 7 to 8 h. Conversion% = consumed alcohol (mol)/alcohol added (mol) × 100. [b] Substrate (1 mmol), $K_8[\gamma\text{-SiW}_{10}O_{36}].13H_2O$ (6.7 μmol), $H_2O_2$ (1 mmol), 70 °C (oil bath temperature).

**Table 41.** Selective oxidation of alcohol catalyzed by $K_8[\gamma\text{-SiW}_{10}O_{36}].13H_2O$ **39** with $H_2O_2$ at 20 °C [a].

$$K_8[\gamma\text{-SiW}_{10}O_{36}].13H_2O \text{ **39** (67 μmol)}$$

$$H_2O_2 \text{ (30%, 5 mmol ), } H_2O \text{ (2 mL)}$$

20 °C, 72 h

| Entry | Substrate | Product | Conv. (mol%) | Yield (mol%) |
|---|---|---|---|---|
| 1 | $C_6H_5CH_2OH$ | $C_6H_5CHO$ | 64 | 64 |
| 2 | $C_6H_5CH(OH)CH_3$ | $C_6H_5COCH_3$ | 78 | 78 |

[a] Reaction conditions: Substrate (1 mmol), $K_8[\gamma\text{-SiW}_{10}O_{36}]\cdot 13H_2O$ **39** (67 μmol), $H_2O_2$ (30%, 5 mmol), $H_2O$ (2 mL), 20 °C, 72 h. Conversion (%) = consumed alcohol (mol)/alcohol added (mol) × 100.

**Table 42.** Oxidation of alcohols with hydrogen peroxide catalyzed by different heteropolyacids at 70 °C in multiphase conditions [a].

HPA **40** (3% in mmol)

Aliquat 336 (tricaprylmethylammonium chloride) (0.59 mmol)

$$H_2O_2 \text{ (35%, 1 mL), } H_2O \text{ (4 mL)}$$

70 °C, 2-9 h

| Entry | Substrate | Product | Catalyst | t/h | $C_6H_5COCH_3$ (%) | Other Products (%) |
|---|---|---|---|---|---|---|
| 1 | $C_6H_5CH(OH)CH_3$ | $C_6H_5COCH_3$ | $H_3PMo_{12}O_{40}$ | 9 | 95 (5) [b] | None |
| 2 | $C_6H_5CH(OH)CH_3$ | $C_6H_5COCH_3$ | $Py_3PMo_{12}O_{40}$ | 5 | 100 (6) [b] | None |
| 3 | $C_6H_5CH(OH)CH_3$ | $C_6H_5COCH_3$ | $H_6PMo_{11}AlO_{40}$ | 7 | 99 | None |
| 4 | $(C_6H_5)_2CHOH$ | $(C_6H_5)_2CO$ | $H_3PMo_{12}O_{40}$ | 5 | 99 (8) [b] | None |
| 5 | $(C_6H_5)_2CHOH$ | $(C_6H_5)_2CO$ | $Py_3PMo_{12}O_{40}$ | 4 | 99 | None |
| 6 | $C_6H_5CH_2OH$ | $C_6H_5CHO$ | $H_3PMo_{12}O_{40}$ | 5 | 84 (12) [b] | (Benzoic acid) (4) |
| 7 | $C_6H_5CH_2OH$ | $C_6H_5CHO$ | $Py_3PMo_{12}O_{40}$ | 3 | 89 (5) [b] | (Benzoic acid) (2) |
| 8 | $C_6H_5CH_2OH$ | $C_6H_5CHO$ | $H_6PMo_{11}AlO_{40}$ | 4 | 85 | (Benzoic acid) (2) |

**Table 42.** *Cont.*

| Entry | Substrate | Product | Catalyst | t/h | $C_6H_5COCH_3$ (%) | Other Products (%) |
|---|---|---|---|---|---|---|
| 9 | $4\text{-}ClC_6H_4CH_2OH$ | $4\text{-}ClC_6H_4CHO$ | $H_3PMo_{12}O_{40}$ | 53 | 93 (10) [b] | (4-Chlorobenzoic acid) (1) |
| 10 | $4\text{-}ClC_6H_4CH_2OH$ | $4\text{-}ClC_6H_4CHO$ | $H_6PMo_{11}AlO_{40}$ | | 95 | (4-Chlorobenzoic acid) (2) |
| 11 | $4\text{-}MeC_6H_4CH_2OH$ | $4\text{-}MeC_6H_4CHO$ | $H_3PMo_{12}O_{40}$ | 43 | 96 (7) [b] | (4-Methylbenzoic acid) (3) |
| 12 | $4\text{-}MeC_6H_4CH_2OH$ | $4\text{-}MeC_6H_4CHO$ | $Py_3PMo_{12}O_{40}$ | | 96 | (4-Methylbenzoic acid) (3) |
| 13 | $4\text{-}OMeC_6H_4CH_2OH$ | $4\text{-}OMeC_6H_4CHO$ | $H_3PMo_{12}O_{40}$ | 3 | 90 | (4-Methoxybenzoic acid) (6) |
| 14 | $4\text{-}OMeC_6H_4CH_2OH$ | $4\text{-}OMeC_6H_4CHO$ | $Py_3PMo_{12}O_{40}$ | 2 | 91 | (4-Methoxybenzoic acid) (5) |

[a] Reaction conditions: Substrate (0.7 mmol), HPA **40** (3% in mmol), $H_2O_2$ (35 *w/v*%, 1 mL), Aliquat 336 (tricaprylmethylammonium chloride, 0.232 g, 0.59 mmol), $H_2O$ (4 mL), *n*-decane as internal standard (0.056 g, 0.39 mmol), 70 °C, 2 to 9 h, 700 rpm. [b] In homogeneous system corresponding to the numbers in the parentheses, substrate (0.7 mmol), HPA (3% in mmol), $H_2O_2$ (35 *w/v*%, 1 mL), CH3CN (5 mL), *n*-decane as internal standard (0.056 g, 0.39 mmol), 700 rpm.

Easy product isolation, good turnovers, and high selectivities were achieved using the Zn substituted polyoxoanion $(NH_4)_7Zn_{0.5}[\alpha\text{-}ZnO_4W_{11}O_{30}ZnO_5(OH_2)]\cdot nH_2O$ (**41**) as the catalyst for the oxidation of organic functionalities. As illustrated in Table 43, only minor (≤5%) amounts of benzoic acids as overoxidation byproduct was provided and the catalyst could be reused three times with little loss in its efficacy [76]. In addition, in contrast to a previously reported system [77,78] using $Na_2WO_4$, a phase transfer catalyst and $H_2O_2$ in water, in this system, no phase transfer co-catalyst was required.

**Table 43.** Oxidation of benzyl alcohol by $(NH_4)_7Zn_{0.5}[\alpha\text{-}ZnO_4W_{11}O_{30}ZnO_5(OH_2)]\cdot nH_2O$ **41** in water with aqueous hydrogen peroxide as an oxidant [a].

| Entry | Substrate | Product | Conv. (%) [b] |
|---|---|---|---|
| 1 | $C_6H_5CH_2OH$ | $C_6H_5CHO$ | 80 |
| 2 | $4\text{-}MeC_6H_4CH_2OH$ | $4\text{-}MeC_6H_4CHO$ | 82 |

[a] Reaction conditions: Substrate (1 mmol), $(NH_4)_7Zn_{0.5}[\alpha\text{-}ZnO_4W_{11}O_{30}ZnO_5(OH_2)]\cdot nH_2O$ **41** (n ≈ 18, 4 μmol), $H_2O_2$ (30 *w/v*%, 4 mmol), $H_2O$ (5 mL), 80 °C, 10 h, with vigorous stirring. [b] Conversion was determined by GC or proton NMR.

Alkali-treated ZSM-5 zeolite (**42**) was obtained by the alkali-treatment modification of the commercially available ZSM-5 zeolite with high concentration NaOH solution at low temperature. The oxidation reaction in the presence of the catalyst (Table 44) showed 53% conversion of BzOH and 86% selectivity to benzaldehyde (BzH). Additionally, the catalyst could be reused for more than six times and was very stable [79]. The results also showed that with an increase of the $SiO_2/Al_2O_3$ ratio in as-received zeolites, the conversion of BzOH decreased seriously which could be attributed to decreasing the concentration of Lewis acids on the surface of treated zeolite.

Among different W- and Mo-based heteropolyoxometalate catalysts which were used in the oxidation of aromatic alcohols in the presence of hydrogen peroxide in water, dodecatungstophosphoric acid, $H_3PW_{12}O_{40}$ (**43**), had the most efficiency. This catalytic system provided a highly selective, efficient, fast, environmentally friendly, and inexpensive approach for the conversion of alcoholic functions to carbonyl groups with $H_2O_2$ in water, and even after fifteen runs, the efficiency of the oxygenation system was ~10% decreased (Table 45) [80].

Table 46 [81] summarizes the application of cheap, effective, organic-solvent-free and phase-transfer $WO_4^{2-}$ catalyst ($Na_2WO_4\cdot 2H_2O$ (**44**)) and aqueous hydrogen peroxide for six primary or secondary alcohol oxidations which are liquids or melt below 90 °C [77,82]. Primary alcohols, such as benzyl alcohol, were easily oxidized to aldehydes with no overoxidation.

**Table 44.** Selective oxidation of benzyl alcohol to benzaldehyde by hydrogen peroxide over different catalyst [a].

catalyst (0.1-2 g)

$H_2O_2$/BzOH (1:3)
reflux, 4 h

| Entry | Catalyst | Amount of Catalyst (g) | Conv. (%) | Sel. (%) | | |
|---|---|---|---|---|---|---|
| | | | | BzH | BzA | Benzyl Benzoate (%) |
| 1 | None | - | 2 | 73 | 27 | 0 |
| 2 | 25ZSM(AT-0) | 1 | 44 | 80 | 19 | 1 |
| 3 | 25HZSM(AT-0.5) [b] | 1 | 52 | 85 | 14 | 1 |
| 4 | 25ZSM(AT-0.5) | 0.1 | 37 | 61 | 31 | 8 |
| 5 | 25ZSM(AT-0.5) | 0.2 | 45 | 68 | 27 | 5 |
| 6 | 25ZSM(AT-0.5) | 0.5 | 52 | 80 | 18 | 2 |
| 7 | 25ZSM(AT-0.5) | 1 | 53 | 86 | 13 | 1 |
| 8 | 25ZSM(AT-0.5) | 1.5 | 49 | 87 | 13 | 0 |
| 9 | 25ZSM(AT-0.5) | 2 | 47 | 88 | 11 | 1 |
| 10 | 25ZSM(AT-1.0) | 1 | 50 | 85 | 14 | 1 |
| 11 | 25ZSM(AT-1.5) | 1 | 48 | 86 | 13 | 1 |
| 12 | 25ZSM(AT-3.0) | 1 | 37 | 89 | 10 | 1 |
| 13 | 38ZSM(AT-0.5) | 1 | 41 | 86 | 14 | 0 |
| 14 | 50ZSM(AT-0.5) | 1 | 33 | 84 | 15 | 1 |

[a] Reaction condition: Catalyst (0.1 to 2 g), $H_2O_2$/BzOH (1:3), reaction at reflux temperature, 4 h. [b] Acid exchange with HCl after alkali-treatment for 0.5 h.

**Table 45.** Oxidation of some aromatic alcohols with 34% $H_2O_2$ in water catalyzed by $H_2PW_{12}O_{40}$ **43** [a].

$H_2PW_{12}O_{40}$ **43** (0.018 mmol)

$H_2O_2$ (34%, 5 mmol), $H_2O$ (5 mL)
rt, 120-150 min

| Entry | Substrate | Product | t/min | Conv. (%) | Sel. (%) |
|---|---|---|---|---|---|
| 1 | 4-OMeC$_6$H$_4$CH$_2$OH | 4-OMeC$_6$H$_4$CHO | 120 | 73 | 100 |
| 2 | C$_6$H$_5$CH$_2$OH | C$_6$H$_5$CHO | 150 | 78 | 100 |
| 3 | C$_6$H$_5$CH(OH)CH$_3$ | C$_6$H$_5$COCH$_3$ | 150 | 85 | 100 |
| 4 | 3-NO$_2$C$_6$H$_4$CH$_2$OH | 3-NO$_2$C$_6$H$_4$CHO | 150 | 90 | 100 |

[a] Reaction conditions: Substrate (0.94 mmol), $H_2PW_{12}O_{40}$ **43** (0.018 mmol), $H_2O_2$ (34%, 5 mmol), $H_2O$ (5 mL), room temperature, 120 to 150 min. Progress of the reactions was followed by aliquots withdrawn directly and periodically from the reaction mixture and analyzed by gas chromatography. Products were isolated as described in the experimental section.

## 2.2. Oxidation of Benzylic and Heterocyclic Alcohols in a Solvent-Free System

Green oxidation of primary and secondary alcohols in the presence of nano-MoO$_3$/copper Schiff base complex (**45**), using $H_2O_2$, was investigated under solvent-free conditions with high conversion and excellent selectivity. The benefits of the reaction also included the ease of isolating products from green media, and the reusability of the catalyst for six times without loss of activity and selectivity (Table 47) [83]. Further oxidation such as the preparation of acid or ester were not investigating (Table 47, Entries 1–4 and 6–12) and, additionally, benzyl phenyl sulfide oxidation provided no sulfoxide or sulfone byproducts (Entry 5).

**Table 46.** Phase-transfer catalyzed oxidation of alcohols [a].

| Entry | Substrate | Product | Work-Up | Yield (%) | Purity (%) [b] |
|-------|-----------|---------|---------|-----------|----------------|
| 1 | $C_6H_5CH(OH)CH_3$ | $C_6H_5COCH_3$ | Extractive | 88<br>91<br>94 | >98<br>96 [c]<br>94 |
| 2 | $C_6H_5CH(OH)Et$ | $C_6H_5COEt$ | Extractive | 85<br>94 | >98<br>>98 |
| 3 | $(C_6H_5)_2CHOH$ | $(C_6H_5)_2CO$ | Extractive | 93<br>98 | >98<br>>98 |
| 4 | $4\text{-}MeC_6H_4CH(OH)C_6H_5$ | $4\text{-}MeC_6H_4COC_6H_5$ | Vacuum filtration | 99<br>65 [d] | >98<br>>98 |
| 5 | $C_6H_5CH_2OH$ | $C_6H_5CHO$ | Extractive | 91<br>81<br>88 [e] | 98<br>94<br>97 |

[a] Reaction conditions: Substrate (1 mol equiv.), $Na_2WO_4 \cdot 2H_2O$ **44** (1 mol%), $[CH_3(C_8H_{17})_3N]HSO_4$ (1 mol%), $H_2O_2$ (30%, 1.1 mol equiv.), $H_2O$, 90 °C, 1 to 3 h. [b] Determined by GC analysis. The remainder was the starting alcohol. [c] Determined by $^1H$ NMR spectroscopy by integration of product and substrate methyl resonances. [d] Yield after recrystallization from methanol. [e] $[CH_3(C_8H_{17})_3N]HSO_4$ prepared in situ from $[CH_3(C_8H_{17})_3N]Cl$ and $NaHSO_4 \cdot H_2O$.

**Table 47.** The oxidation of different alcohols catalyzed by NMCS bio-nanocomposite **45** using $H_2O_2$ within 3 h at 80 °C under ultrasonic irradiation [a].

| Entry | Substrate | Product | Yield (%) |
|-------|-----------|---------|-----------|
| 1 | $4\text{-}(tert\text{-}Butyl)C_6H_4CH(OH)CH_3$ | $4\text{-}(tert\text{-}Butyl)C_6H_4COCH_3$ | 100 |
| 2 | $C_6H_5CH(OH)Et$ | $C_6H_5COEt$ | 90 |
| 3 | $(C_6H_5)_2CHOH$ | $(C_6H_5)_2CO$ | 60 |
| 4 | $C_6H_5CH(OH)CH_3$ | $C_6H_5COCH_3$ | 100 |
| 5 | $2\text{-}SHC_6H_4CH_2OH$ | $2\text{-}SHC_6H_4CHO$ | 100 |
| 6 | $4\text{-}MeC_6H_4CH_2OH$ | $4\text{-}MeC_6H_4CHO$ | 100 |
| 7 | Furfuryl alcohol | Furfural | 100 |
| 8 | $2\text{-}OHC_6H_4CH_2OH$ | $2\text{-}OHC_6H_4CHO$ | 90 |
| 9 | $C_6H_5CH_2OH$ | $C_6H_5CHO$ | 100 |
| 10 | $2,4\text{-}ClC_6H_3CH_2OH$ | $2,4\text{-}ClC_6H_3CHO$ | 90 |
| 11 | $4\text{-}NO_2C_6H_4CH_2OH$ | $4\text{-}NO_2C_6H_4CHO$ | 85 |
| 12 | $4\text{-}ClC_6H_4CH_2OH$ | $4\text{-}ClC_6H_4CHO$ | 80 |

[a] Reaction conditions: Substrate (0.1 mmol), NMCS bio-nanocomposite **45** (0.007 g), $H_2O_2$ (30%, 0.2 mmol, 20 µL), 80 °C, solvent free, air, 3 h.

Dioxo-molybdenum(VI) complex supported functionalized Merrifield resin (MR-SB-Mo) (**46**). The catalyst efficiently and selectively oxidized a wide variety of alcohols to aldehydes or ketones using $H_2O_2$ as an oxidant with reasonably good TOF (660 h$^{-1}$ in case of benzyl alcohol) under solvent-free reaction conditions and did not lead to overoxidized products under optimized conditions (Table 48) [84]. The catalyst afforded regeneration and could be reused for at least five reaction cycles without loss of efficiency and product selectivity. In the case of substituted benzyl alcohols, different types of substituents such as –F, –Cl, –Br, –OMe, –OH, and –NO$_2$ were well-tolerated during the oxidation process (Table 48, Entries 2–7), some of which could be utilized for further derivation. One of the notable aspects of the developed catalytic system was its ability to oxidize benzyl alcohol to benzaldehyde at a relatively higher scale (10 g scale) without losing the catalytic efficiency and product selectivity (Table 48, Entry 1$^d$), which provides its potential application in commercial processes.

**Table 48.** Oxidation of alcohols to aldehydes or ketones catalyzed by MR-SB-Mo **46** using 30% aqueous $H_2O_2$ as oxidant [a].

| Entry | Substrate | Product | t (min) | Yield (%) [b] | TOF (h$^{-1}$) [c] |
|---|---|---|---|---|---|
| 1 | $C_6H_5CH_2OH$ | $C_6H_5CHO$ | 90 / 90 / 90 | 99 / 98 [d] / 97 [e] | 660 / 653 / 646 |
| 2 [f] | $4\text{-}FC_6H_4CH_2OH$ | $4\text{-}FC_6H_4CHO$ | 100 | 97 | 582 |
| 3 [f] | $4\text{-}ClC_6H_4CH_2OH$ | $4\text{-}ClC_6H_4CHO$ | 100 | 96 | 576 |
| 4 [f] | $4\text{-}BrC_6H_4CH_2OH$ | $4\text{-}BrC_6H_4CHO$ | 100 | 97 | 582 |
| 5 [f] | $4\text{-}NO_2C_6H_4CH_2OH$ | $4\text{-}NO_2C_6H_4CHO$ | 105 | 97 | 554 |
| 6 [f] | $4\text{-}OMeC_6H_4CH_2OH$ | $4\text{-}OMeC_6H_4CHO$ | 100 | 99 | 594 |
| 7 | $4\text{-}OHC_6H_4CH_2OH$ | $4\text{-}OHC_6H_4CHO$ | 100 | 98 | 588 |
| 8 | $(C_6H_5)_2CHOH$ | $(C_6H_5)_2CO$ | 120 | 98 | 490 |
| 9 | $C_6H_5CH_2OH$ | $C_6H_5CHO$ | 105 | 97 | 554 |

[a] Reaction conditions: Unless otherwise stated, all reactions were performed, substrate (2.5 mmol), MR-SB-Mo **46** (5.6 mg, contain 0.0025 mmol of Mo), $H_2O_2$ (30%, 2.75 mmol), solvent free, 65 °C, 90 to 120 min. [b] Isolated yield. [c] TOF = (mmol of product)/[(mmol of catalyst) × (time)]. [d] Yield at 5th reaction cycle. [e] Yield at 10 g scale reaction. [f] Reaction conducted with 2 mL acetonitrile.

Active Au-Pd catalysts on carbon and titanium oxide (Au-Pd/C (**47**) and Au-Pd/TiO$_2$ (**48**)) were applied for the benzyl alcohol oxidation without adding solvents. The results showed that carbon-supported bimetallic catalysts provided a higher conversion as compared with Au-Pd/TiO$_2$ **48** catalyst, due to the vast superficial area of carbon supported catalyst as compared with the titanium supported catalyst (Figure 1) [85,86]. These results demonstrated the critical importance of correct selection of support for the metal nanoparticles immobilization in the catalyst preparation approaches. Additionally, some dissimilarities in the selectivity and catalytic activities of catalysts could be the result of both the interaction of catalyst support and the influence of the shape of metal particles.

Fe$_3$O$_4$/GrOSi(CH$_2$)$_3$–NH$_2$/HPMo nanocomposite (**49**) as a magnetically recyclable (at least four times) heterogeneous separable oxidation catalyst was applied for selective oxidation of alcohols to corresponding aldehydes (Table 49, Entries 1–10) and ketones (Table 49, Entries 11–13) with very high selectivity (≥99%) in moderate to excellent yields (60% to 96%) with $H_2O_2$ under solvent-free conditions (Table 49) [87].

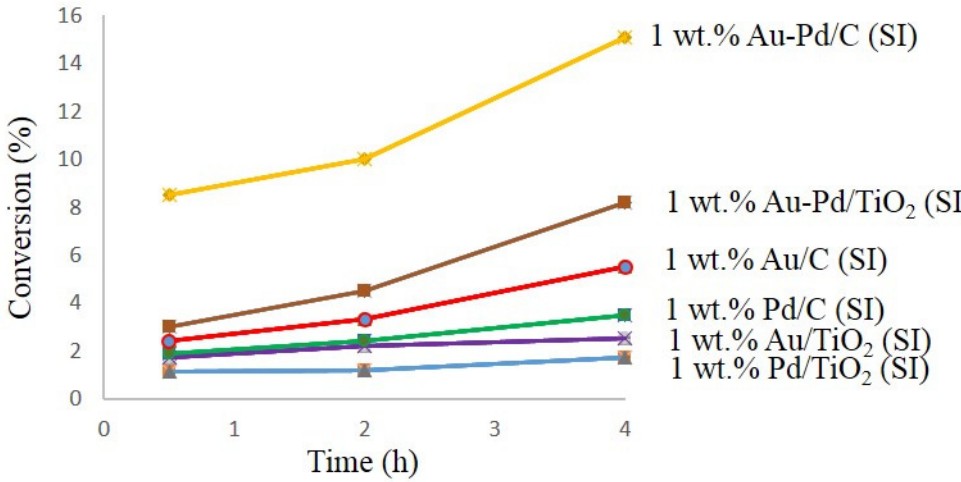

**Figure 1.** Curves conversion versus time for carbon-supported catalysts and titanium supported catalysts.

**Table 49.** Results of various alcohol oxidations with $H_2O_2$ catalyzed by the $Fe_3O_4/GrOSi(CH_2)_3$-$NH_2/HPMo$ **49** catalyst [a].

| Entry | Substrate | Product | Yield (%)[b] |
|-------|-----------|---------|-----------|
| 1 | $C_6H_5CH_2OH$ | $C_6H_5CHO$ | 90 |
| 2 | $4\text{-}iPrC_6H_4CH_2OH$ | $4\text{-}iPrC_6H_4CHO$ | 96 |
| 3 | $4\text{-}OMeC_6H_4CH_2OH$ | $4\text{-}OMeC_6H_4CHO$ | 94 |
| 4 | $4\text{-}NO_2C_6H_4CH_2OH$ | $4\text{-}NO_2C_6H_4CHO$ | 86 |
| 5 | $2\text{-}NO_2C_6H_4CH_2OH$ | $2\text{-}NO_2C_6H_4CHO$ | 60 |
| 6 | $3\text{-}NO_2C_6H_4CH_2OH$ | $3\text{-}NO_2C_6H_4CHO$ | 82 |
| 7 | $4\text{-}BrC_6H_4CH_2OH$ | $4\text{-}BrC_6H_4CHO$ | 92 |
| 8 | $4\text{-}ClC_6H_4CH_2OH$ | $4\text{-}ClC_6H_4CHO$ | 94 |
| 9 | $2\text{-}ClC_6H_4CH_2OH$ | $2\text{-}ClC_6H_4CHO$ | 82 |
| 10 | $2,4\text{-}ClC_6H_3CH_2OH$ | $2,4\text{-}ClC_6H_3CHO$ | 90 |
| 11 | $(C_6H_5)_2CHOH$ | $(C_6H_5)_2CO$ | 90 |
| 12 | $C_6H_5CH(OH)CH_3$ | $C_6H_5COCH_3$ | 90 |
| 13 | $C_6H_5CH(OH)COC_6H_5$ | $(C_6H_5)_2(CO)_2$ | 90 |

[a] Reaction conditions: Substrate (10 mmol), $Fe_3O_4/GrOSi(CH_2)_3$-$NH_2/HPMo$ **49** (0.2 g), $H_2O_2$ (30 wt%, 15 mmol), solvent free, 100 °C, 4 h. [b] Yields were determined by GC analysis with *n*-decane as internal standard.

Homotrinuclear copper catalyst $[Cu_3(slmh)(\mu\text{-}Cl)_2(CH_3OH)_3]\cdot0.5CH_3OH$ (**50**), where $H_4slmh$ stands for disalicylaldehyde malonoyldihydrazone was prepared by Lal et al., in 2017 [88]. This heterogeneous catalyst oxidized several aromatic ring substituted benzylic alcohols by aqueous hydrogen peroxide and the results are summarized in Table 50 [88]. No benzoic acid production was observed in benzylic alcohol oxidations. Moreover, recyclability of the catalyst showed the five consecutive runs ability with no loss of activity.

**Table 50.** Oxidation of alcohols catalyzed by $[Cu_3(slmh)(\mu-Cl)_2(CH_3OH)_3]\cdot0.5CH_3OH$ **50** under solvent-free conditions in the absence of base and co-catalyst [a].

| Entry | Substrate | Product | Yield (%) [b] |
|-------|-----------|---------|---------------|
| 1 | $C_6H_5CH_2OH$ | $C_6H_5CHO$ | 78 |
| 2 | $2\text{-}MeC_6H_4CH_2OH$ | $2\text{-}MeC_6H_4CHO$ | 80 |
| 3 | $4\text{-}OMeC_6H_4CH_2OH$ | $4\text{-}OMeC_6H_4CHO$ | 88 |
| 4 | $3\text{-}ClC_6H_4CH_2OH$ | $3\text{-}ClC_6H_4CHO$ | 94 |
| 5 | $4\text{-}MeC_6H_4CH_2OH$ | $4\text{-}MeC_6H_4CHO$ | 81 |
| 6 | 2-pyridinemethanol | picolinaldehyde | 93 |
| 7 | $3,4,5\text{-}OMeC_6H_2CH_2OH$ | $3,4,5\text{-}OMeC_6H_2CHO$ | 88 |
| 8 | Furfuryl alcohol | Furfural | 92 |
| 9 | 2-Thiophenemethanol | 2-Thiophenecarboxaldehyde | 90 |
| 10 | $4\text{-}NO_2C_6H_4CH_2OH$ | $4\text{-}NO_2C_6H_4CHO$ | 97 |
| 11 | $C_6H_5CH(OH)COC_6H_5$ | $(C_6H_5)_2(CO)_2$ | 85 |
| 12 | $C_6H_5CH(OH)CH_3$ | $C_6H_5COCH_3$ | 1 |
| 13 | $4\text{-}OMeC_6H_4CH_2OH{:}C_6H_5CH(OH)CH_3$ | $4\text{-}OMeC_6H_4CHO{:}C_6H_5COCH_3$ | 77:1 |

[a] Reaction conditions: Substrate (4.62 mmol), $[Cu_3(slmh)(\mu-Cl)_2(CH_3OH)_3]\cdot0.5CH_3OH$ **50** (0.04 mmol), $H_2O_2$ (15%, 8.46 mmol), solvent free, 70 °C, 15 h. [b] Isolated yield.

The selective oxidation of alcohols using green oxidants, $H_2O_2$, in the presence of $Mo_3O_{10}(C_5H_6N)_2\cdot H_2O$ (**51**) showed the efficiency of the catalyst for the oxidations of alcohols to the corresponding ketones or aldehydes under solvent-free conditions over reaction-controlled phase-transfer catalysis (Table 51) [89]. The method was so selective with no overoxidation product preparation even with longer reaction time. In addition, $MoO_x$-pyridine could efficiently oxidize secondary benzylic alcohols to their corresponding ketones with excellent conversions and high selectivity (Table 51, Entries 8–11) [89]. It is worthwhile to mention that the catalyst could be reused in $H_2O_2$ systems for at least four runs.

**Table 51.** Oxidation of alcohols to corresponding aldehydes or ketones using $Mo_3O_{10}(C_5H_6N)_2\cdot H_2O$ wires (MoOx-pyridine) **51** catalyst [a].

| Entry | Substrate | Product | t/min | Conv. (%) [b] |
|-------|-----------|---------|-------|---------------|
| 1 | $C_6H_5CH_2OH$ | $C_6H_5CHO$ | 45 | >95 |
| 2 | $4\text{-}ClC_6H_4CH_2OH$ | $4\text{-}ClC_6H_4CHO$ | 90 | 95 |
| 3 | $2\text{-}ClC_6H_4CH_2OH$ | $2\text{-}ClC_6H_4CHO$ | 60 | >95 |
| 4 | $2,4\text{-}ClC_6H_3CH_2OH$ | $2,4\text{-}ClC_6H_3CHO$ | 120 | 90 |
| 5 | $4\text{-}NO_2C_6H_4CH_2OH$ | $4\text{-}NO_2C_6H_4CHO$ | 10 | 50 |
| 6 | $2\text{-}OHC_6H_4CH_2OH$ | $2\text{-}OHC_6H_4CHO$ | 70 | 95 |

**Table 51.** *Cont.*

| Entry | Substrate | Product | t/min | Conv. (%) [b] |
|-------|-----------|---------|-------|---------------|
| 7 | 4-OMeC$_6$H$_4$CH$_2$OH | 4-OMeC$_6$H$_4$CHO | 30 | 90 |
| 8 | C$_6$H$_5$CH(OH)Et | C$_6$H$_5$COEt | 75 | 95 |
| 9 | C$_6$H$_5$CH(OH)CH$_3$ | C$_6$H$_5$COCH$_3$ | 90 | 95 |
| 10 | (C$_6$H$_5$)$_2$CHOH | (C$_6$H$_5$)$_2$CO | 60 | 95 |
| 11 | C$_6$H$_5$CH(OH)COC$_6$H$_5$ | (C$_6$H$_5$)$_2$(CO)$_2$ | 50 | 95 |
| 12 | Furfuryl alcohol | Furfural | 40 | 80 |

[a] Reaction conditions: Substrate (1 mmol), Mo$_3$O$_{10}$(C$_5$H$_6$N)$_2$·H$_2$O wires **51** (1.6 mol%), H$_2$O$_2$ (1 mmol), solvent free, 80 °C, 10 to 120 min. [b] Conversions were obtained by TLC and GC.

Water-soluble heterobimetallic complex [CuZn(bz)$_3$(bpy)$_2$]BF$_4$ (**52**) was applied for selective oxidations of primary alcohols to the corresponding aldehydes with no base or co-catalyst. In addition, the catalyst could be recyclable and keep its activity intact over five cycles (Table 52) [90].

**Table 52.** Oxidation of alcohols catalyzed by [CuZn(bz)$_3$(bpy)$_2$]BF$_4$ **52** under solvent-free condition in the absent of base [a].

| Entry | Substrate | Product | t/h | Yield (%) [b] Isolated Product (GC) [c] |
|-------|-----------|---------|-----|-----------------------------------------|
| 1 | C$_6$H$_5$CH$_2$OH | C$_6$H$_5$CHO | 7 | 84 (86)[c] |
| 2 | 4-OMeC$_6$H$_4$CH$_2$OH | 4-OMeC$_6$H$_4$CHO | 3 | 98 (100)[c] |
| 3 | 2-MeC$_6$H$_4$CH$_2$OH | 2-MeC$_6$H$_4$CHO | 9 | 97 |
| 4 | 3-MeC$_6$H$_4$CH$_2$OH | 3-MeC$_6$H$_4$CHO | 10 | 93 |
| 5 | 4-MeC$_6$H$_4$CH$_2$OH | 4-MeC$_6$H$_4$CHO | 4 | 98 |
| 6 | 3-ClC$_6$H$_4$CH$_2$OH | 3-ClC$_6$H$_4$CHO | 9 | 97 |
| 7 | 3-NO$_2$C$_6$H$_4$CH$_2$OH | 3-NO$_2$C$_6$H$_4$CHO | 3 | 85 (97) [c] |
| 8 | Furfuryl alcohol | Furfural | 3 | 97 (100) [c] |
| 9 | 2-Pyridinemethanol | Picolinaldehyde | 3 | 96 |
| 10 | 2-Thiophenemethanol | 2-Thiophenecarboxaldehyde | 3 | 98 |
| 11 | C$_6$H$_5$CH(OH)COC$_6$H$_5$ | (C$_6$H$_5$)$_2$(CO)$_2$ | 2 | 98 (100) [c] |
| 12 | 3,4,5-OMeC$_6$H$_2$CH$_2$OH | 3,4,5-OMeC$_6$H$_2$CHO | 6 | 98 |
| 13 | C$_6$H$_5$CH(OH)CH$_3$ | C$_6$H$_5$COCH$_3$ | 16 | 1 [c] |

[a] Reaction conditions: Substrate (4.84 mmol), [CuZn(bz)$_3$(bpy)$_2$]BF$_4$ **52** (0.023 mmol), H$_2$O$_2$ (30%, 8.83 mmol), solvent free, 70 °C, 2 to 16 h. [b] Isolated yield via column chromatography. [c] GC conversion.

Table 53 presents the application of Cu(II) Schiff base complex [Cu(HL)(H$_2$O)NO$_3$] (**53**) using 2-[(2-hydroxy-1,1-dimethyl-ethylimino)methylphenol (H$_2$L) in the oxidation of different alcohols [91]. Primary benzylic alcohols either with electron donating or electron withdrawing groups underwent selective and efficient oxidation to the corresponding aldehydes with no overoxidation (Table 53, Entries 1–8). The oxidation of secondary alcohols provided ketones in good to high yield (Table 53, Entries 10–12). The catalyst was also reused five times without losing of its activity.

**Table 53.** Oxidation of alcohols using $H_2O_2$ catalyzed by $[Cu(HL)(H_2O)NO_3]$ **53** [a].

| Entry | Substrate | Product [b] | Yield (%) [c] |
|-------|-----------|-------------|---------------|
| 1 | $C_6H_5CH_2OH$ | $C_6H_5CHO$ | 94 |
| 2 | $2\text{-}OHC_6H_4CH_2OH$ | $2\text{-}OHC_6H_4CHO$ | 62 |
| 3 | $4\text{-}OHC_6H_4CH_2OH$ | $4\text{-}OHC_6H_4CHO$ | 69 |
| 4 | $2\text{-}ClC_6H_4CH_2OH$ | $2\text{-}ClC_6H_4CHO$ | 65 |
| 5 | $4\text{-}ClC_6H_4CH_2OH$ | $4\text{-}ClC_6H_4CHO$ | 84 [d] |
| 6 | $2,4\text{-}ClC_6H_3CH_2OH$ | $2,4\text{-}ClC_6H_3CHO$ | 90 |
| 7 | $4\text{-}MeC_6H_4CH_2OH$ | $4\text{-}MeC_6H_4CHO$ | 81 [e] |
| 8 | $4\text{-}OMeC_6H_4CH_2OH$ | $4\text{-}OMeC_6H_4CHO$ | 82 |
| 9 | Furfuryl alcohol | Furfural | 68 |
| 10 | $4\text{-}OMeC_6H_5CH(OH)CH_3$ | $4\text{-}OMeC_6H_5COCH_3$ | 86 |
| 11 | $C_6H_5CH(OH)CH_3$ | $C_6H_5COCH_3$ | 87 |
| 12 | 1-Tetralol | 1-Tetralone | 70 |

[a] Reaction conditions: Substrate (1 mmol), $[Cu(HL)(H_2O)NO_3]$ **53** (0.02 mmol), $H_2O_2$ (30%, 3 mmol), solvent free, 70 °C, 3 h. [b] All products are identified by comparison of their physical data with those of authentic samples. [c] Isolated yield. Yields are determined by TLC and GC based on the starting alcohol. [d] $^1$H NMR (500 MHz, CDCl$_3$) δ: 7.26 to 7.82 (m, 4H, ArH) and 9.96 (s, 1H, aldehyde H). [e] $^1$H NMR (500 MHz, CDCl$_3$) δ: 2.31 (s, 3H, CH$_3$), 7.20 (d, $J$ = 15 Hz, 2H, ArH), 7.66 (d, $J$ = 15 Hz, 2H, ArH), and 9.95 (s, 1H, aldehyde H).

The treatment of mordenite acid using hydrochloric acid (MOR-HC) (**54**), nitric acid (MOR-HN) (**55**), and oxalic acid (MOR-O$_x$) improved its textural properties such as increasing the BET surface area, total pore volume, mesopore volume, and external surface area (Table 54) [92]. These improvements influenced the catalytic activity of the mordenite in oxidation of benzyl alcohol not only in benzyl alcohol conversion both also in benzaldehyde selectivities. In terms of catalytic properties, the highest improvement was observed in the case of the nitric acid treated sample as follows: benzyl alcohol conversion >99% and benzaldehyde selectivities >99%. In addition, the fittingness of the MOR-HN for various acid catalyzed bulky molecular transformations was observed. Decreasing acid site density of mordenite by dealumination and the increasing porosity, resulted in controlling overoxidation of benzaldehyde to undesired benzoic acid in the MOR-HN catalyst [92].

**Table 54.** Performance of various catalysts in benzyl alcohol oxidation [a].

| Entry | Catalyst | Conv. of BzOH (wt%) | Product Yield (wt%) BzH | Product Yield (wt%) Benzoic Acid |
|-------|----------|---------------------|-------------------------|----------------------------------|
| 1 | MOR | 42.95 | 41.15 | 1.80 |
| 2 | MOR-HN | 99.94 | 99.84 | 0.1 |
| 3 | MOR-HC | 54.49 | 54.35 | 0.14 |
| 4 | MOR-OX | 60.37 | 60.25 | 0.12 |

[a] Reaction conditions: Benzyl alcohol (14 mL), catalyst (0.5 g), $H_2O_2$ (30 wt%, 13 mL), deionized $H_2O$ (26 mL), pressure (atmospheric), 90 °C, 4 h.

Tin-containing 1,3-din-butylimidazolium bromide ([BBIM]Br)-SnCl$_2$ (**56**) was prepared by Xing et al. [93] and successfully applied for the simple and efficient oxidation of benzyl alcohol using hydrogen peroxide as the oxidant (Table 55). In this catalyst, coordination of Sn species with the imidazole ring resulted in the easy recovery and reusability of the catalyst for six reaction runs without loss of catalytic activity.

**Table 55.** Catalytic properties of [BBIM]Br-SnCl$_2$ **56** in the oxidation of various alcohols with H$_2$O$_2$ [a].

| Entry | Substrate | Product | t/min | Conv. (%) [b] | Sel. (%) [b] |
|---|---|---|---|---|---|
| 1 | C$_6$H$_5$CH$_2$OH | C$_6$H$_5$CHO | 15 | 100 | 95.1 |
| 2 | 4-NO$_2$C$_6$H$_4$CH$_2$OH | 4-NO$_2$C$_6$H$_4$CHO | 60 | 71 | 99 |
| 3 | 4-CH$_2$OHC$_6$H$_4$CH$_2$OH | 4-CH$_2$OHC$_6$H$_4$CHO | 15 [c] | 99 | 90 |
| 4 | 4-OMeC$_6$H$_4$CH$_2$OH | 4-OMeC$_6$H$_4$CHO | 160 | 87 | 91 |
| 5 | 4-iPrC$_6$H$_4$CH$_2$OH | 4-iPrC$_6$H$_4$CHO | 15 | 99 | 50 |
| 6 | 4-OHC$_6$H$_4$CH$_2$OH | 4-OHC$_6$H$_4$CHO | 15 | 99 | 98 |
| 7 | (C$_6$H$_5$)$_2$CHOH | (C$_6$H$_5$)$_2$CO | 15 | 99 | >99 |

[a] Reaction conditions: Substrate (1.5 mmol), [BBIM]Br–SnCl$_2$ **56** (0.11 g), H$_2$O$_2$ (30%, 2 mmol), solvent free, 65 °C, 15 to 160 min. [b] GC analysis. [c] Reaction was carried out using H$_2$O$_2$ (30%, 3.5 mmol).

The CoFe$_2$O$_4$ nanoparticles (NPs) (**57**) catalyzed the solvent-free oxidation of benzyl alcohol (BzOH) to benzaldehyde (BzH) with hydrogen peroxide (Table 56) [94]. Excellent results, >99% conversation of BzOH and 100% selectivity, were obtained with no obvious loss of catalyst activity after three consecutive runs.

**Table 56.** Selective oxidation of benzyl alcohol to benzaldehyde catalyzed by CoFe$_2$O$_4$ NPs **57** [a].

| Entry | Substrate | Product | Conv. (%) | Sel. (%) |
|---|---|---|---|---|
| 1 | C$_6$H$_5$CH$_2$OH | C$_6$H$_5$CHO | >99 | 100 |

[a] Reaction conditions: Substrate (3 mL, 29 mmol), CoFe$_2$O$_4$ NPs **57** (40 mg), H$_2$O$_2$ (35%, 145 mmol, 12.5 mL), solvent free, 110 °C, 5 h.

The Mn- and Co-substituted polyoxotungstates [MPW11O39]$^{5-}$ (M = Mn or Co) was immobilized on MCM-41 or into the interlayer of Mg$_3$Al-layered double hydroxide. The catalysts showed catalytic activity in the presence of H$_2$O$_2$ for the solvent-free oxidation of benzyl alcohol for at least four times with no loss of activity and selectivity. As illustrated in Table 57 [95], POM loaded-LDH (**58**) and POM loaded-LDH-adipate (**59**) catalysts were better catalysts than POM-free catalysts due to easier reduction of metal ion via interaction of POM anion with the clay surface [96]. In addition, the basic sites of LDH-supported catalysts resulted in higher selectivity for benzaldehyde as compared with the unsupported catalysts [97].

**Table 57.** Oxidation of benzyl alcohol with various catalyst [a].

| Entry | Catalyst | t/h | Conv. (%) | Sel. for BzH (%) |
|---|---|---|---|---|
| 1 | - | 5 | 2 | 70 |
| 2 | MnPOM/MCM | 5 | 67 | 75 |
| | | 8 | 76 | 69 |
| 3 | CoPOM/MCM | 5 | 70 | 71 |
| | | 8 | 78 | 67 |
| 4 | TBA-MnPOM [b] | 5 | 49 | 75 |
| 5 | TBA-CoPOM [b] | 5 | 52 | 70 |
| 6 | TBA-MnPOM [c] | 2.5 | 35 | Not determined |
| 7 | TBA-CoPOM [c] | 2.5 | 13 | Not determined |
| 8 | LDH | 5 | 3 | 91 |
| 9 | MnPOM/LDH | 5 | 45 | 97 |
| 10 | CoPOM/LDH | 5 | 37 | 96 |
| 11 | LDH-adi | 5 | 4 | 90 |
| 12 | MnPOM/LDH-adi | 5 | 66 | 100 |
| 13 | CoPOM/LDH-adi | 5 | 69 | 99 |

[a] Reaction conditions: Benzyl alcohol (10 mL), catalyst (5 wt%), $H_2O_2$/alcohol molar ratio (3:1), 90 °C, 2.5 to 8 h. Oxidation reaction: Substrate (benzyl alcohol), catalyst, and aqueous $H_2O_2$ (30%) were inserted into a 60 mL parr reactor. The reactor was heated to a desired temperature and the mixture was stirred. The catalyst was separated, and the products were extracted with diethyl ether, dried, and analyzed by GC (chromatograph, Varian CP-3800 GC; column, CP-Sil 30 m × 0.25 mm). [b] $H_2O_2$/alcohol molar ratio of 2:1. [c] $H_2O_2$/alcohol molar ratio of 5:1, in $H_2O$, 100 °C [98].

Two long chain multi-$SO_3H$ functionalized heteropolyanion-based ionic liquids S4SiIL (**60**) and S3PIL (**61**) as homogeneous catalysts provided aldehydes and ketones with 63% to 100% yields through the oxidation of alcohols with no phase transfer catalyst and could be reused five times. The corresponding benzoic acids of benzyl alcohols were obtained with 64% to 94% yields (Table 58) [99]. For **60**, benzaldehyde was obtained in 94% yield, whereas for **61**, the benzoic acid provided 83% yield.

**Table 58.** Results of selective oxidation of alcohols catalyzed by S4SiIL **60** and S3PIL **61** [a].

| Entry | Substrate | Product | Conv. (%) | Yield (%) |
|---|---|---|---|---|
| 1 | $C_6H_5CH_2OH$ | $C_6H_5CHO$ | 100 [b] | 94 [b] |
| | | $C_6H_5CO_2H$ | 99 [c] | 83 [c] |
| 2 | 4-$MeC_6H_4CH_2OH$ | 4-$MeC_6H_4CHO$ | 100 [b] | 100 [b] |
| | | 4-$MeC_6H_4CO_2H$ | 99 [c] | 87 [b] |
| 3 | 4-$OMeC_6H_4CH_2OH$ | 4-$OMeC_6H_4CHO$ | 100 [b] | 100 [c] |
| | | 4-$OMeC_6H_4CO_2H$ | 99 [c] | 94 [c] |
| 4 | 4-$ClC_6H_4CH_2OH$ | 4-$ClC_6H_4CHO$ | 99 [b] | 91 [c] |
| | | 4-$ClC_6H_4CO_2H$ | 97 [c] | 77 [c] |
| 5 | 4-$NO_2C_6H_4CH_2OH$ | 4-$NO_2C_6H_4CHO$ | 96 [b] | 81 [c] |
| | | 4-$NO_2C_6H_4CO_2H$ | 95 [c] | 64 [c] |
| 6 [d] | $C_6H_5CH(OH)CH_3$ | $C_6H_5COCH_3$ | 98 [b]/95[c] | 94 [b]/80 [c] |
| 7 [d] | $(C_6H_5)_2CHOH$ | $(C_6H_5)_2CO$ | 95 [b]/89[c] | 83 [b]/76 [c] |

[a] Reaction conditions: Substrate (30 mmol), S4SiIL **60** and S3PIL **61** (0.05 mmol), $H_2O_2$ (35%, 45 mmol), solvent free, 70 °C, 4 h. [b] Results of S4SiIL. [c] Results of S3PIL. [d] Substrate (30 mmol), S4SiIL and S3PIL (0.05 mmol), $H_2O_2$ (60 mmol), 70 °C, 4 h.

A polyethylene oxide-supported long-chain imidazolium polyoxometalate hybrid catalyst ([PEO-didodecylimidazolium]₃[PW₁₂O₄₀]₂) (**62**) was employed as the recyclable catalyst (could be reused six times) for the selective oxidation reactions of alcohols using $H_2O_2$ at room temperature, without significant loss of its activity, as shown in Table 59 [100]. Ketones or aldehydes were the only produced products. Primary benzylic alcohols had the best activities with no significant effects on the electronic properties, electron-withdrawing, or electron-donating substituents, or steric hindrance of the groups on the benzene ring [100].

**Table 59.** Oxidation of various alcohols [a].

| Entry | Substrate | Product | t/h | Conv. (%) | Yield (%) |
|---|---|---|---|---|---|
| 1 | $C_6H_5CH_2OH$ | $C_6H_5CHO$ | 6 | 100 | 98 |
| 2 | $2\text{-OMeC}_6H_4CH_2OH$ | $2\text{-OMeC}_6H_4CHO$ | 6 | 99 | 96 |
| 3 | $4\text{-OMeC}_6H_4CH_2OH$ | $4\text{-OMeC}_6H_4CHO$ | 5 | >99 | 98 |
| 4 | $3\text{-OMeC}_6H_4CH_2OH$ | $3\text{-OMeC}_6H_4CHO$ | 8 | 98 | 94 |
| 5 | $4\text{-MeC}_6H_4CH_2OH$ | $4\text{-MeC}_6H_4CHO$ | 5 | >99 | 99 |
| 6 | $4\text{-ClC}_6H_4CH_2OH$ | $4\text{-ClC}_6H_4CHO$ | 8 | 96 | 95 |
| 7 | $4\text{-NO}_2C_6H_4CH_2OH$ | $4\text{-NO}_2C_6H_4CHO$ | 8 | 95 | 93 |
| 8 | $(C_6H_5)_2CHOH$ | $(C_6H_5)_2CO$ | 16 | 81 | 80 |
| 9 | $C_6H_5CH(OH)CH_3$ | $C_6H_5COCH_3$ | 14 | 83 | 81 |
| 10 | 2-Pyridinemethanol | Picolinaldehyde | 16 | 86 | 84 |
| 11 | Furfuryl alcohol | Furfural | 12 | 90 | 82 |

[a] Reaction conditions: Substrate (10 mmol), [PEO-didodecylimidazolium]₃[PW₁₂O₄₀]₂ **62** (0.05 mol % of alcohol), $H_2O_2$ (30%, 2 equiv.), solvent free, room temperature, 5 to 16 h.

Schiff base complexes of transition metal ions supported on silica coated magnetic metallic cobalt nanoparticles (Co@SiO₂@[M(II)SBC], M = Co, Cu, Ni, Zn) (**63**, **64**, **65**, **66**) as heterogeneous catalysts were prepared and explored for the oxidations of the alcohols to corresponding aldehydes in high yields. The catalysts could be recyclable five times. The obtained results, as shown in Table 60 [101], indicated that alcohol conversions occurred during approximately half an hour with 100% selectivity. Although all of these catalysts selectively converted different alcohols to corresponding aldehyde, the Ni(II) catalysts were more active than three others and converted alcohols to aldehydes in shorter time.

**Table 60.** Investigation of catalytic activity of Co@SiO₂@[Co(II)SBC] **63** catalysts on the oxidation reaction [a].

| Entry | Substrate | Product | t/min | Yield (%) [b] | Sel. (%) |
|---|---|---|---|---|---|
| 1 | $4\text{-ClC}_6H_4CH_2OH$ | $4\text{-ClC}_6H_4CHO$ | 27–28 | 91 | 100 |
| 2 | $4\text{-MeC}_6H_4CH_2OH$ | $4\text{-MeC}_6H_4CHO$ | 24–26 | 93 | 100 |
| 3 | $4\text{-FC}_6H_4CH_2OH$ | $4\text{-FC}_6H_4CHO$ | 32–33 | 94 | 100 |
| 4 | $4\text{-OMeC}_6H_4CH_2OH$ | $4\text{-OMeC}_6H_4CHO$ | 22–25 | 90 | 100 |

**Table 60.** *Cont.*

| Entry | Substrate | Product | t/min | Yield (%) [b] | Sel. (%) |
|-------|-----------|---------|-------|-----------|----------|
| 5 | 4-NO$_2$C$_6$H$_4$CH$_2$OH | 4-NO$_2$C$_6$H$_4$CHO | 25–27 | 91 | 100 |
| 6 | C$_6$H$_5$CH$_2$OH | C$_6$H$_5$CHO | 24–26 | 92 | 100 |

[a] Reaction conditions: Substrate (0.5 mmol), Co@SiO$_2$@[Co(II)SBC] **63** (0.04 g), H$_2$O$_2$ (30%, 60 µL), solvent free, 50 °C, 24 to 33 min. [b] Isolated yield.

Bismuth tribromide (BiBr$_3$) (**67**) catalyzed oxidations of both primary and secondary benzylic alcohols with aqueous H$_2$O$_2$ to corresponding carbonyl compounds in high yields, as summarized in Table 61 [102]. Without overoxidation, primary benzylic alcohols resulted in exclusive formation of aldehydes. Generally, the oxidation of primary benzylic alcohols proceeded slower than primary secondary alcohols.

**Table 61.** Oxidation of alcohols with H$_2$O$_2$ and BiBr$_3$ **67** [a].

| Entry | Substrate | Product | t/min | Yield (%) [b] |
|-------|-----------|---------|-------|-----------|
| 1 | C$_6$H$_5$CH$_2$OH | C$_6$H$_5$CHO | 30 | 90 |
| 2 | 2-MeC$_6$H$_4$CH$_2$OH | 2-MeC$_6$H$_4$CHO | 30 | 93 |
| 3 | 4-MeC$_6$H$_4$CH$_2$OH | 4-MeC$_6$H$_4$CHO | 30 | 92 |
| 4 | 2-ClC$_6$H$_4$CH$_2$OH | 2-ClC$_6$H$_4$CHO | 30 | 85 |
| 5 | 4-ClC$_6$H$_4$CH$_2$OH | 4-ClC$_6$H$_4$CHO | 30 | 89 |
| 6 | 4-BrC$_6$H$_4$CH$_2$OH | 4-BrC$_6$H$_4$CHO | 30 | 85 |
| 7 | 4-OMeC$_6$H$_4$CH$_2$OH | 4-OMeC$_6$H$_4$CHO | 30 | 72 |
| 8 | 4-NO$_2$C$_6$H$_4$CH$_2$OH | 4-NO$_2$C$_6$H$_4$CHO | 30 | 80 |
| 9 | 1-Naphthalenmethanol | 1-Naphthaldehyde | 10 | 75 |
| 10 | C$_6$H$_5$CH(OH)CH$_3$ | C$_6$H$_5$COCH$_3$ | 10 | 80 |
| 11 | C$_6$H$_5$CH(OH)C$_2$H$_5$ | C$_6$H$_5$COC$_2$H$_5$ | 10 | 85 |
| 12 | 4-MeC$_6$H$_5$CH(OH)CH$_3$ | 4-MeC$_6$H$_5$COCH$_3$ | 20 | 90 |
| 13 | 4-ClC$_6$H$_5$CH(OH)CH$_3$ | 4-ClC$_6$H$_5$COCH$_3$ | 10 | 95 |
| 14 | 4-BrC$_6$H$_5$CH(OH)CH$_3$ | 4-BrC$_6$H$_5$COCH$_3$ | 10 | 95 |
| 15 | 4-FC$_6$H$_5$CH(OH)CH$_3$ | 4-FC$_6$H$_5$COCH$_3$ | 10 | 96 |
| 16 | 1-(Naphthalen-2-yl)ethanol | 2-Acetonaphthone | 10 | 80 |
| 17 | 1-Tetralol | 1-Tetralone | 10 | 85 |
| 18 | (C$_6$H$_5$)$_2$CHOH | (C$_6$H$_5$)$_2$CO | 40 | 85 |
| 19 | C$_6$H$_5$CH(OH)CO$_2$Et | C$_6$H$_5$COCO$_2$Et | 10 | 88 |

[a] Reaction conditions: Substrate (1 mmol), BiBr$_3$ **67** (10 mol%), H$_2$O$_2$ (30%, 5 mmol), solvent free, 70 °C, 10 to 40 min. [b] Isolated yield.

Three amphiphilic sulphonato-salen-chromium(III) complexes immobilized on MCM-41 (Cr(SO$_3$-salphen)-MCM-41) (**68**) were applied for the selective oxidation of BzOH to BzH with H$_2$O$_2$, and the results are listed in Table 62 [103]. The introduction of hydrophilic sulfonic acid groups to the catalyst structure enhanced accessibility of the catalyst with oxidant which increased the catalytic act in oxidation reactions as compare with their corresponding lipophilic complexes. The best results were also obtained, i.e., 60.3% benzyl alcohol conversion with 100% benzaldehyde selectivity. In addition, Cr(SO$_3$-salphen)-MCM-41 complex could be reused for five runs.

**Table 62.** Catalytic performance of samples in benzyl alcohol oxidation [a].

heterogenized catalysts (0.25 g)
homogeneous catalysts (1.25 mmol)
$H_2O_2$ (30%, 125 mmol), solvent free
50 °C, 4 h

| Entry | Catalyst | Conv. of BzOH (%) | Sel. (%) | | | $H_2O_2$ Efficiency (mol %) |
|---|---|---|---|---|---|---|
| | | | BzH | Benzoic Acid | Benzyl Benzoate | |
| 1 | Blank | 3 | 100 | 0 | 0 | 5.1 |
| 2 | Cr(salen) | 12.2 | 65.4 | 23.5 | 11.1 | 15.3 |
| 3 | Cr(salten) | 15 | 70.2 | 19.8 | 10 | 17.8 |
| 4 | Cr(salphen) | 20.5 | 75.6 | 18.2 | 6.2 | 23.6 |
| 5 | $Cr(SO_3$-salen) | 25 | 76.5 | 17.5 | 6 | 28 |
| 6 | $Cr(SO_3$-salten) | 31.4 | 82 | 15.3 | 2.7 | 35.5 |
| 7 | $Cr(SO_3$-salphen) | 34.7 | 80.5 | 15.4 | 4.1 | 37.9 |
| 8 | Cr(salen)-MCM-41 | 42.5 | 98.2 | 1.3 | 0.4 | 45.6 |
| 9 | Cr(salten)-MCM-41 | 43.2 | 100 | 0 | 0 | 48.1 |
| 10 | Cr(salphen)-MCM-41 | 45.2 | 100 | 0 | 0 | 52 |
| 11 | $Cr(SO_3$-salen)-MCM-41 | 53.1 | 100 | 0 | 0 | 62.5 |
| 12 | $Cr(SO_3$-salten)-MCM-41 | 57 | 100 | 0 | 0 | 68.3 |
| 13 | $Cr(SO_3$-salphen)-MCM-41 | 60.3 | 100 | 0 | 0 | 72.5 |

[a] Reaction conditions: Benzyl alcohol (50 mmol), $H_2O_2$ (30%, 125 mmol), heterogenized catalysts (0.25 g) or homogeneous catalysts (1.25 mmol), 50 °C, 4 h.

Cesium salt of transition metal (M = Co, Mn, Ni) substituted phosphomolybdates $PMo_{11}M$ as catalysts for the oxidation of alcohol were applied and the catalyst showed the activity in the order of activity $PMo_{11}Co$ (**69**) ≥ $PMo_{11}Ni$ (**70**) > $PMo_{11}Mn$ (**71**) as illustrated in Table 63 [104]. In addition, the catalysts could be reused up to 2 cycles.

**Table 63.** Oxidation of various alcohols catalyzed by $PMo_{11}M$ (M = Co, Mn, Ni), under optimized conditions [a].

$PMo_{11}M$ (M=Co, Ni, Mn) **69, 70, 71**
(20 mg)
alcohol/$H_2O_2$ (1:3)
90 °C, 24 h

| Entry | Substrate | Product | Conv. (%) | Sel. (%) |
|---|---|---|---|---|
| 1 | $C_6H_5CH_2OH$ | $C_6H_5CHO$ | 56.5 [b] | 90.9 |
| | | | 37.9 [c] | 84.9 |
| | | | 56.2 [d] | 81.3 |

[a] Reaction conditions: $PMo_{11}M$ (M = Co, Ni, Mn) **69, 70, 71** (20 mg), mole ratio of alcohol to $H_2O_2$ (1:3), 90 °C, 24 h.
[b] [M] = Co. [c] [M] = Mn. [d] [M] = Ni.

Using cationic surfactants with different carbon-chain lengths for the functionalization of the V-containing Keggin POM $H_4PMo_{11}VO_{40}$ provided a series of polyoxometalate (POM)-based amphiphilic catalysts. The catalysts were for selective benzyl alcohol oxidation by $H_2O_2$ (Table 64) [105] and the best catalytic efficiency was obtained by $(ODA)_4PMo_{11}VO_{40}$ (**72**) (ODA: octadecylmethylammonium) due to its amphiphilic property (60.6% of benzyl alcohol conversion with 99% of selectivity for benzaldehyde). Furthermore, the catalyst could be recycled over four run reactions without losing activity.

**Table 64.** Selective oxidation of benzyl alcohol to benzaldehyde with $H_2O_2$ in the presence of the catalysts investigated [a].

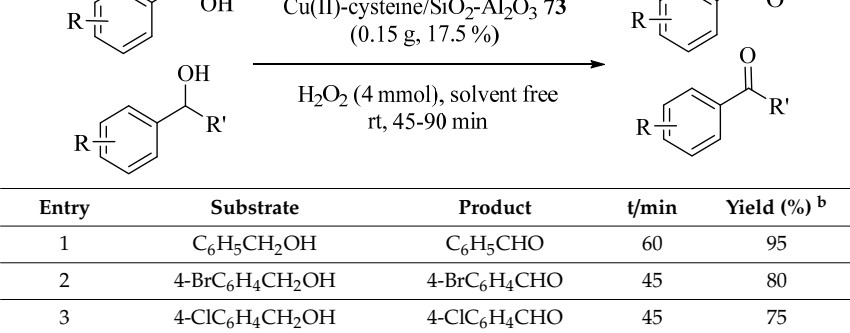

| Entry | Catalyst | Conv. (%) [b] | Sel. (%) [c] |
|---|---|---|---|
| 1 | $H_3PMo_{12}O_{40}$ | $19.1 \pm 1.9$ [d] | $88.6 \pm 1.6$ [d] |
| 2 | $H_4PMo_{11}VO_{40}$ | $28.2 \pm 1.8$ | $90 \pm 1.4$ |
| 3 | $(ODA)_4PMo_{11}VO_{40}$ | $60.6 \pm 1.5$ | $99 \pm 0.8$ |
| 4 | ODACl | 0 | 0 |
| 5 | $(ODA)_3PMo_{12}VO_{40}$ | $36.5 \pm 1.7$ | $94.5 \pm 1.1$ |
| 6 | $(ODA)_4PMo_{11}VO_{40}$-P | $58.9 \pm 1.6$ | $99 \pm 0.8$ |
| 7 | $(ODACl + H_4PMo_{11}VO_{40})$ [e] | $57.1 \pm 1.7$ | $98.8 \pm 0.9$ |
| 8 | $(DODA)_4PMo_{11}VO_{40}$ | $53.3 \pm 1.8$ | $99 \pm 0.9$ |
| 9 | $(DDA)_4PMo_{11}VO_{40}$ | $51.6 \pm 1.8$ | $98 \pm 1$ |
| 10 | $(HDA)PMo_{11}VO_{40}$ | $56.3 \pm 1.8$ | $99 \pm 0.9$ |
| 11 | $(TBA)_4PMo_{11}VO_{40}$ | $28.9 \pm 1.2$ | $91.1 \pm 1.4$ |

[a] Reaction conditions: Benzyl alcohol (60 mmol), catalyst (0.03 mmol), $H_2O_2$ (50 mmol), solvent free, 90 °C, 6 h. [b] Conversion of benzyl alcohol. [c] Selectivity for benzaldehyde. [d] Standard deviation. [e] ODACl (0.12 mmol) and $H_4PMo_{11}VO_{40}$ (0.03 mmol) were simultaneously added into the reaction system.

A simple adsorption method provided heterogeneous copper(II)-cysteine/$SiO_2$-$Al_2O_3$ (**73**) catalyst which represented good activity and selectivity in aromatic alcohol oxidation reactions, as shown in Table 65 [106]. All reactants were completely converted to selective carbonyl compounds. In addition, shifting the substituent from electron withdrawing (Entries 2–5) to electron donating (Entries 6–9) remarkably increased the yield of corresponding aldehydes from 45% to 98%. The catalyst could also be recycled over five times.

**Table 65.** Solvent-free oxidation of various aromatic alcohols over Cu(II)-cysteine/$SiO_2$-$Al_2O_3$ **73** [a].

| Entry | Substrate | Product | t/min | Yield (%) [b] |
|---|---|---|---|---|
| 1 | $C_6H_5CH_2OH$ | $C_6H_5CHO$ | 60 | 95 |
| 2 | $4$-$BrC_6H_4CH_2OH$ | $4$-$BrC_6H_4CHO$ | 45 | 80 |
| 3 | $4$-$ClC_6H_4CH_2OH$ | $4$-$ClC_6H_4CHO$ | 45 | 75 |
| 4 | $3$-$NO_2C_6H_4CH_2OH$ | $3$-$NO_2C_6H_4CHO$ | 90 | 45 |
| 5 | $4$-$NO_2C_6H_4CH_2OH$ | $4$-$NO_2C_6H_4CHO$ | 85 | 45 |
| 6 | $2$-$MeC_6H_4CH_2OH$ | $2$-$MeC_6H_4CHO$ | 50 | 90 |
| 7 | $3$-$OHC_6H_4CH_2OH$ | $3$-$OHC_6H_4CHO$ | 45 | 98 |
| 8 | $3$-$OMeC_6H_4CH_2OH$ | $3$-$OMeC_6H_4CHO$ | 45 | 98 |
| 9 | $4$-$ClC_6H_4CH(OH)C_6H_5$ | $4$-$ClC_6H_4COC_6H_5$ | 80 | 80 |
| 10 | $(C_6H_5)_2CHOH$ | $(C_6H_5)_2CO$ | 70 | 85 |

[a] Reaction conditions: Substrate (1 mmol), Cu(II)-cysteine/$SiO_2$-$Al_2O_3$ **73** (0.15 g, 17.5% Cu(II)-cysteine/$SiO_2$-$Al_2O_3$), $H_2O_2$ (4 mmol), solvent free, room temperature, 45 to 90 min. [b] Isolated yield.

Lal et al. [107] reported the solvent-free oxidation of primary (Entries 2–5, 9) and secondary (Entries 6–8) benzylic alcohols using a heterotrinuclear complex containing a dicopper(II)–monozinc(II)

centre, [ZnCu$_2$(slsch)(NO$_3$)$_2$(H$_2$O)$_8$]·2H$_2$O (**74**), with hydrogen peroxide, as summarized in Table 66, with good yield and excellent selectivity and four consecutive run reusability.

**Table 66.** Hydrogen peroxide mediated oxidation of alcohols to aldehydes and ketones [a].

| Entry | Substrate | Product | t/h | Yield (%) [b] |
|-------|-----------|---------|-----|---------------|
| 1 | C$_6$H$_5$CH$_2$OH | C$_6$H$_5$CHO | 10 | 85 |
| 2 | 4-OMeC$_6$H$_4$CH$_2$OH | 4-OMeC$_6$H$_4$CHO | 11 | 87 |
| 3 | 4-NMe$_2$C$_6$H$_4$CH$_2$OH | 4-NMe$_2$C$_6$H$_4$CHO | 11 | 88 |
| 4 | 4-NO$_2$C$_6$H$_4$CH$_2$OH | 4-NO$_2$C$_6$H$_4$CHO | 9 | 82 |
| 5 | 2-ClC$_6$H$_4$CH$_2$OH | 2-ClC$_6$H$_4$CHO | 10 | 78 |
| 6 | C$_6$H$_5$CH(OH)COC$_6$H$_5$ | (C$_6$H$_5$)$_2$(CO)$_2$ | 12 | 80 |
| 7 | C$_6$H$_5$CH(OH)CH$_3$ | C$_6$H$_5$COCH$_3$ | 12 | 80 |
| 8 | 4-ClC$_6$H$_4$CH(OH)CH$_3$ | 4-ClC$_6$H$_4$COCH$_3$ | 14 | 82 |
| 9 | 4-MeC$_6$H$_4$CH(OH)CH$_3$ | 4-MeC$_6$H$_4$COCH$_3$ | 12 | 78 |

[a] Reaction condition: Substrate (5 mmol), [ZnCu$_2$(slsch)(NO$_3$)$_2$(H$_2$O)$_8$]·2H$_2$O **74** (8 mol%), H$_2$O$_2$ (30%, 10 mmol), solvent free, 100 °C, 9 to 14 h. [b] Isolated yields.

Under mild conditions, primary and secondary alcohols were oxidized by applying rutheniumbis(benzimidazole)pyridinedicarboxylate complex [Ru(bbp)(pydic)] (**75**) as catalyst and H$_2$O$_2$ as oxidant. Aldehydes and ketones were prepared with good yield and excellent selectivity (Table 67) [108].

**Table 67.** Oxidation of various alcohols with H$_2$O$_2$ catalyzed by Ru(bbp)(pydic) **75** [a].

| Entry | Substrate | Product | t/h | Conv. (%) [b] | Yield (%) [b] |
|-------|-----------|---------|-----|---------------|---------------|
| 1 | C$_6$H$_5$CH$_2$OH | C$_6$H$_5$CHO | 1 | 97 | 96 |
| 2 | 4-MeC$_6$H$_4$CH$_2$OH | 4-MeC$_6$H$_4$CHO | 1 | 98 | 96 |
| 3 | 4-OMeC$_6$H$_4$CH$_2$OH | 4-OMeC$_6$H$_4$CHO | 1 | 96 | 93 |
| 4 | 4-NO$_2$C$_6$H$_4$CH$_2$OH | 4-NO$_2$C$_6$H$_4$CHO | 2 | 82 | 81 |
| 5 [c] | 4-Pyridinemethanol | Isonicolinaldehyde | 3 | 76 | 74 |
| 6 | C$_6$H$_5$CH(OH)CH$_3$ | C$_6$H$_5$COCH$_3$ | 1 | 92 | 92 |
| 7 | (C$_6$H$_5$)$_2$CHOH | (C$_6$H$_5$)$_2$CO | 3 | 82 | 82 |

[a] Reaction condition: Substrate (2 mmol), Ru(bbp)(pydic) **75** (2 × 10$^{-3}$ mmol), H$_2$O$_2$ (30%, 10 mmol), solvent free, 60 °C, 1 to 3 h. [b] Determined by GC. [c] Ru(bbp)(pydic) (4 × 10$^{-3}$ mmol).

Keggintype polyoxometallates, such as M$_3$(PW$_{12}$O$_{40}$)$_2$ [M = Ni, Zn, Co, Mn, Cu; denoted as MPW$_{12}$], FePW$_{12}$O$_{40}$ [FePW$_{12}$] (**76**), Ni$_3$(PMo$_{12}$O$_{40}$)$_2$ [NiPMo$_{12}$] (**77**), and Ni$_2$SiW$_{12}$O$_{40}$ [NiSiW$_{12}$] (**78**) were synthesized as catalysts, under solvent-free condition, for the selective oxidation of BzOH into BzH with hydrogen peroxide (Table 68) [109]. Among them, Ni$_3$(PW$_{12}$O$_{40}$)$_2$·26H$_2$O (**77**) provided high catalytic activity, TON of 550.6 mol/(mol cat.), and 87.3% selectivity. The above catalyst was also recovered and reused for three times.

**Table 68.** Oxidation of alcohols [a].

| Entry | Substrate | Product | Conv. (%) | Yield (%) | Sel. (%) |
|-------|-----------|---------|-----------|-----------|----------|
| 1 | $C_6H_5CH_2OH$ | $C_6H_5CHO$ | 90.9 | 79.4 | 87.3 |
| 2 | $4\text{-}OMeC_6H_4CH_2OH$ | $4\text{-}OMeC_6H_4CHO$ $4\text{-}OMeC_6H_4CHO$ [b] | 93.9 82 | 27.1 78.7 | 28.9 96 |

[a] Reaction Condition: Substrate (47.5 mmol, 5 mL), $Ni_3(PW_{12}O_{40})_2 \cdot 26H_2O$ **77** (0.5 g), $H_2O_2$ (30%, 93 mmol, 10 mL), solvent free, 90 °C (in an water bath), 3.5 h. [b] Conditions: substrate (47.5 mmol, 5 mL), $Ni_3(PW_{12}O_{40})_2 \cdot 26H_2O$ (0.5 g), $H_2O_2$ (30%, 93 mmol, 10 mL), solvent free, 70 °C, 3.5 h.

Appling *L*-aspartic acid coupled with imidazolium based ionic liquid [L-AAIL] (**79**), in the presence of hydrogen peroxide as an ideal oxidant, provided a green protocol for the selective oxidation of alcohols which could be recycled and reused for seven runs. The substituted primary benzylic alcohols were selectively oxidized to aldehydes with 88% to 96% yield (Table 69, Entries 1–6, 9) and the secondary alcohols also produced ketones with 76% to 84% yield (Table 69, Entries 7, 8, 10) [110].

**Table 69.** L-AAIL **79** catalyzed oxidation of alcohols by hydrogen peroxide at 25 °C [a].

| Entry | Substrate | Product | t/min | Yield (%) [b,c] |
|-------|-----------|---------|-------|-----------------|
| 1 | $C_6H_5CH_2OH$ | $C_6H_5CHO$ | 30 | 96 |
| 2 | $4\text{-}ClC_6H_4CH_2OH$ | $4\text{-}ClC_6H_4CHO$ | 38 | 93 |
| 3 | $4\text{-}BrC_6H_4CH_2OH$ | $4\text{-}BrC_6H_4CHO$ | 40 | 92 |
| 4 | $4\text{-}NH_2C_6H_4CH_2OH$ | $4\text{-}NH_2C_6H_4CHO$ | 38 | 94 |
| 5 | $4\text{-}OMeC_6H_4CH_2OH$ | $4\text{-}OMeC_6H_4CHO$ | 35 | 96 |
| 6 | $4\text{-}MeC_6H_4CH_2OH$ | $4\text{-}MeC_6H_4CHO$ | 35 | 88 |
| 7 | $C_6H_5CH(OH)CH_3$ | $C_6H_5COCH_3$ | 65 | 84 |
| 8 | $(C_6H_5)_2CHOH$ | $(C_6H_5)_2CO$ | 85 | 82 |
| 9 | Furfuryl alcohol | Furfural | 210 | 88 |
| 10 | 2-Pyridinemethanol | Picolinaldehyde | 190 | 90 |

[a] Reaction condition: Substrate (2 mmol), L-AAIL **79** (0.0025 mmol), $H_2O_2$ (5 mmol), solvent free, 25 °C, 30 to 210 min. [b] Quantified by HPLC of the crude product. [c] Isolated yield by flash chromatography.

Under solvent-free conditions, hydrogen peroxide as oxidant and metal dodecanesulfonate salts, $M(DS)_x$ [M = $Fe^{3+}$ (**80**), $Cu^{2+}$ (**81**), $Ni^{2+}$ (**82**) and $Sn^{2+}$ (**83**), DS = dodecanesulfonate, x = 3 or 2] as catalysts were conducted by biphasic catalysis for benzyl alcohol oxidation to benzaldehyde. The results are summarized in Table 70 [111] and indicate that $Fe(DS)_3$ (**80**) catalyst represented a surprisingly high activity, exhibiting nearly 100% conversion of BzOH and TON of 194.4 mol/(mol cat) as compare with other catalytic systems, due to its difference in Lewis acidity (Entry 1) [112]. In addition, this catalyst could be recovered and reused three times. Overoxidation of BzH into benzylic acid was minimized in biphasic operation systems due to reducing the contact opportunity between BzH in bulk organic phase and the catalyst on the interphase or $H_2O_2$ that was soluble in the aqueous phase.

**Table 70.** Catalytic activity of various dodecanesulfonate salts [a].

| Entry | Catalyst (0.25 mmol) | Conv. (%) | TON (mol/mol$_{cat}$) |
|-------|----------------------|-----------|----------------------|
| 1 | Fe(DS)$_3$ | 100 | 194.4 |
| 2 | Cu(DS)$_2$ | 47.3 | 92 |
| 3 | Ni(DS)$_2$ | 7.8 | 15.2 |
| 4 | Sn(DS)$_2$ | 4 | 7.8 |

[a] Reaction conditions: Ratio of H$_2$O$_2$ to alcohol (0.22:0.0486, 4.5:1), catalyst (0.2 g), solvent free, 90 °C, 6 h, stirring speed 250 rpm.

A series of conventional chromium(III) Schiff base complexes were immobilized on MCM-41. Without any organic solvent, phase transfer catalyst, or additive, the mentioned complexes showed much higher catalytic performance in benzyl alcohol conversion to benzaldehyde, due to their corresponding homogeneous analogs. The catalyst represented difference activity by varying ligand in the following order: 84-L4 > 84-L2 > 84-L1 > 84-L5 > 84-L3 (Figure 2) [113]. The difference could be attributed to ligand structures which **84**-L4 with π-extended coordination structure exhibiting the best catalytic performance.

**84**

L1: R = -CH$_2$CH$_2$-; L2: R = -CH$_2$CHNHCHCH$_2$-; L3: R =

L4: R =   L5: R =

**Figure 2.** Homogeneous chromiun (III) Schiff base complexes and their immobilized analouge were prepared as described in "experimental" section.

Different alcohol oxidations were also applied over the representative catalyst **84**-L4 (Table 71) [113]. All benzylic alcohols were oxidized with good yields, and the major products were the corresponding ketones or aldehydes. Overoxidation of aldehydes into their corresponding acids were also observed. The catalyst could be reusedfor four catalytic runs, with only a little change in the catalytic performance.

As illustrated in Table 72 [114], green oxidant hydrogen peroxide and cyclopentadienyl molybdenum acetylide catalyst, CpMo(CO)$_3$(C≡CPh) (**85**), were used to explore selective oxidations of different aromatic alcohols to aldehydes with very high conversion (90%) and aldehyde selectivity

(90%). The catalytic system could be reused even after five recycles with no decrease in alcohol conversion and aldehyde selectivity.

**Table 71.** Catalytic performance of the representative immobilized complex in the oxidation of alcohols [a].

$$\text{C}_6\text{H}_5\text{CH}_2\text{OH} \xrightarrow[\text{H}_2\text{O}_2 \text{ (0.125 mol), solvent free} \atop 50\,°\text{C; 4 h}]{\textbf{84}\text{-L4 (0.25 g)}} \text{C}_6\text{H}_5\text{CHO}$$

| Entry | Substrate | Product | Conv. (mol%) [b] | Yield (mol%) [c] |
|-------|-----------|---------|------------------|------------------|
| 1 | C$_6$H$_5$CH$_2$OH | C$_6$H$_5$CHO | 45 [d] | 45 |
| | | | 44 [e] | 44 |
| | | | 43 [f] | 42 |
| | | | 41 [g] | 38 |

[a] Reaction conditions: Substrate (0.05 mol), **84**-L4 (0.25 g), H$_2$O$_2$ (0.125 mol), solvent free, 50 °C, 4 h. [b] Conversion = (moles of substrate reacted/moles of substrate in the feed) × 100. [c] Yield = (moles of substrate converted to the products/moles of substrate in the feed) × 100. [d, e, f, g] Recycle runs: 1, 2, 3, 4 respectively.

**Table 72.** Oxidation of different alcohols [a].

$$\text{R}\text{—}\text{C}_6\text{H}_4\text{CH}_2\text{OH} \xrightarrow[\text{H}_2\text{O}_2 \text{ (30\%, 0.1 mol), solvent free} \atop 80\,°\text{C, 8 h}]{\text{CpMo(CO)}_3(\text{C}\equiv\text{CPh}) \textbf{ 85 } (1\text{-}0.01 \text{ mmol})} \text{R}\text{—}\text{C}_6\text{H}_4\text{CHO}$$

| Entry | Substrate | Product | Conv. (%) | Yield (%) | Sel. (%) | | TON |
|-------|-----------|---------|-----------|-----------|----------|------|-----|
| | | | | | Aldehyde | Acid | |
| 1 | C$_6$H$_5$CH$_2$OH | C$_6$H$_5$CHO | 86 | 79 | 92 | 8 | 396 |
| 2 | 4-MeC$_6$H$_4$CH$_2$OH | 4-MeC$_6$H$_4$CHO | 90 | 78 | 87 | 13 | 391 |
| 3 | 4-OMeC$_6$H$_4$CH$_2$OH | 4-OMeC$_6$H$_4$CHO | 90 | 90 | 90 | 10 | 392 |
| 4 | 3,5-OMeC$_6$H$_3$CH$_2$OH | 3,5-OMeC$_6$H$_3$CHO | 83 | 71 | 85 | 15 | 352 |
| 5 | 4-NO$_2$C$_6$H$_4$CH$_2$OH | 4-NO$_2$C$_6$H$_4$CHO | 60 | 53 | 88 | 12 | 264 |
| 6 | 4-ClC$_6$H$_4$CH$_2$OH | 4-ClC$_6$H$_4$CHO | 65 | 62 | 91 | 9 | 296 |
| 7 | 2,4-ClC$_6$H$_3$CH$_2$OH | 2,4-ClC$_6$H$_3$CHO | 78 | 70 | 90 | 10 | 343 |

[a] Reaction conditions: Substrate (0.05 mol), CpMo(CO)$_3$(C≡CPh) **85** (1 to 0.01 mmol), H$_2$O$_2$ (30%, 0.1 mol), solvent free, 80 °C, 8 h.

Nano-γ-Fe$_2$O$_3$ **86** as a selective and active catalyst for alcohol and olefin oxidation yielded the corresponding aldehyde, with good to excellent selectivity, without losing catalyst activity even after reusing 5 times. As illustrated in Table 73 [115], significant impact on the catalyst activity was provided by substitutions on the aromatic ring of the benzyl alcohol. Moreover, as compared with benzyl alcohol, 1-phenylethanol, diphenylmethanol, and 1-phenyl-1-propanol gave good conversions with lower selectivity (Table 73, Entries 4–6).

Some methyl-containing Cr(salen) complexes which immobilized on MCM-41 were prepared as catalysts and exhibited for the selective oxidation of benzyl alcohol (BzOH) with H$_2$O$_2$. All organic solvent-free systems, with no phase transfer catalyst or additive, showed much higher catalytic activity than their homogeneous analogue. Benzaldehyde (BzH), benzoic acid, and benzyl benzoate were only detected as products (Table 74) [116]. By increasing methyl content of catalysts, from Cr(salen)-MCM-41(CH$_3$)$_1$ (**87**) to Cr(salen)-MCM-41(CH$_3$)$_3$ (**89**), BzOH conversion and H$_2$O$_2$ efficiency decreased slowly, whereas BzH selectivity increased, as can be seen in the presence of Cr(salen)-MCM-41(CH$_3$)$_2$ (**88**) and **89**, BzOH was completely oxidized to BzH with no the formation of

byproducts. The best BzOH conversion of 65.0% with 100% selectivity to benzaldehyde (BzH) was reached using catalytic system **88** [116].

**Table 73.** Selective oxidation of alcohols to aldehydes and ketones with nano-γ-Fe$_2$O$_3$ **86** [a].

| Entry | Substrate | Product | Conv. (%) | Sel. (%) | TON |
|-------|-----------|---------|-----------|----------|-----|
| 1 | 4-(tert-Butyl)C$_6$H$_4$CH$_2$OH | 4-(tert-Butyl) C$_6$H$_4$CHO | 10 | >99 | 10 |
| 2 | 4-FC$_6$H$_4$CH$_2$OH | 4-FC$_6$H$_4$CHO | 87 | 51 | 44 |
| 3 | 4-ClC$_6$H$_4$CH$_2$OH | 4-ClC$_6$H$_4$CHO | 98 | 61 | 60 |
| 4 | C$_6$H$_5$CH(OH)CH$_3$ | C$_6$H$_5$COCH$_3$ | 42 | 60 | 25 |
| 5 | C$_6$H$_5$CH(OH)Et | C$_6$H$_5$COEt | 35 | 42 | 44 |
| 6 | (C$_6$H$_5$)$_2$CHOH | (C$_6$H$_5$)$_2$CO | 49 | 52 | 25 |
| 7 | 2-Biphenylmethanol | 2-Biphenylcarbaldehyde | 6 | >99 | 6 |

[a] Reaction conditions: Substrate (10 mmol, 1.08 g), nano-γ-Fe$_2$O$_3$ **86** (1 mol%, 16 mg), H$_2$O$_2$ (30 wt%, 1 mL, 10 mmol), solvent free, 75 °C, 12 h.

**Table 74.** Catalytic performance of the obtained complexes in the selective oxidation of benzyl alcohol [a].

| Entry | Catalyst | Conv. of BzOH (mol %) [b] | Sel. (mol%) [c] | | | H$_2$O$_2$ Efficiency (mol %) [d] |
|-------|----------|--------------------------|-----------------|---|---|----------------------------------|
| | | | BzH | Benzoic Acid | Benzyl Benzoate | |
| 1 | Blank | 3 | 100 | 0 | 0 | 5.1 |
| 2 | Cr(salen) | 12.2 | 65.4 | 23.5 | 11.1 | 15.3 |
| 3 | Cr(salen)-MCM-41 | 40.5 | 80.6 | 12.4 | 7 | 46.5 |
| 4 | Cr(salen)-MCM-41(CH$_3$)$_1$ | 66.1 | 92.3 | 5.6 | 2.1 | 77.3 |
| 5 | Cr(salen)-MCM-41(CH$_3$)$_2$ | 65 | 100 | 0 | 0 | 76.6 |
| 6 | Cr(salen)-MCM-41(CH$_3$)$_3$ | 61.3 | 100 | 0 | 0 | 70.9 |

[a] Reaction conditions: Benzyl alcohol (0.05 mol), heterogenized catalyst (0.25 g) or homogeneous complex (1 mol%), H$_2$O$_2$ (30%, 0.125 mol), solvent free, 50 °C, 4 h. [b] Conversion = (moles of BzOH reacted/moles of BzOH in the feed) × 100. [c] Selectivity = (moles of BzOH converted to the products/moles of BzOH reacted) × 100. [d] Efficiency of H$_2$O$_2$ for BzOH oxidation = (moles of H$_2$O$_2$ converted to the products/moles of H$_2$O$_2$ consumed) × 100.

Wang et al. reported the preparation of Mo$^{VI}$ oxo-diperoxo complex [MoO(O$_2$)$_2$(TEDA)$_2$] (**90**) (TEDA = 1,4-diazabicyclo [2.2.2]octane) [117], which successfully catalyzed the high yield oxidations of alcohols to their corresponding carbonyl groups by H$_2$O$_2$. The catalyst was active even for three recycling experiments with no overoxidation (Table 75).

Some tetra-alkylpyridinium octamolybdate (**91–94**) were used as catalysts for selective oxidations of benzyl alcohol. Among them, as illustrated in Table 76, [118], tetraalkylpyridinium octamolybdate exhibited high activity and selectivity due to oxygen transfer ability from hydrogen peroxide to the substrate. This matter is clear by 82.3% to 94.8% benzyl alcohol conversion and 87.9% to 96.7%

benzaldehyde selectivity. The simple and easy catalyst preparations and utilization, high recovery, and short reaction time made the catalytic system an ideal choice for future investigations.

**Table 75.** [MoO(O$_2$)$_2$(TEDA)$_2$]-catalyzed oxidation of selected alcohols using H$_2$O$_2$ as oxidant [a].

$$R\text{—}C_6H_4\text{—}CH_2OH \xrightarrow[\substack{H_2O_2 \text{ (30\%, 2 mmol), solvent free} \\ 80\,°C,\,6\,h}]{[MoO(O_2)_2(TEDA)_2]\,\mathbf{90}\,(0.01\,mol,\,1\,mol\,\%)} R\text{—}C_6H_4\text{—}CHO$$

| Entry | Substrate | Product | Yield (%) [b] | Sel. (%) |
|-------|-----------|---------|---------------|----------|
| 1 | C$_6$H$_5$CH$_2$OH | C$_6$H$_5$CHO | 96 | 98 |
| 2 | 4-MeC$_6$H$_4$CH$_2$OH | 4-MeC$_6$H$_4$CHO | 96 | 96 |

[a] Reaction conditions: Substrate (1 mmol), [MoO(O$_2$)$_2$(TEDA)$_2$] **90** (0.01 mol, 1 mol %), H$_2$O$_2$ (30%, 2 mmol), solvent free, 80 °C, 6 h, control experiments without complex **90** showed no oxidation under the same reaction conditions.
[b] Determined by GC using an internal standard technique.

**Table 76.** Oxidation of benzyl alcohol to benzaldehyde by hydrogen peroxide over tetra-alkylpyridinium octamolybdate catalysts [a].

$$C_6H_5CH_2OH \xrightarrow[\substack{H_2O_2 \text{ (15\%, 25 g, 0.12 mol)} \\ reflux,\,0.4\text{-}1.7\,h}]{\substack{[R\,(\pi\text{-}C_5H_5N)]_4Mo_8O_{26}\,(0.5\text{-}0.8\,g) \\ Where\,R=\,n\text{-}C_4H_9,\,n\text{-}C_8H_{17},\,n\text{-}C_{14}H_{29},\,n\text{-}C_{16}H_{33}}} C_6H_5CHO$$

| Entry | Catalyst | Amount of Catalyst/g | BzOH/H$_2$O$_2$ Molar Ratio | t/h | Conv. (%) | Sel. (%) |
|-------|----------|----------------------|------------------------------|-----|-----------|----------|
| 1 | | 0.5 | 1:1.2 | 1.7 | 93.6 | 90.5 |
| 2 | PyC4 | 0.8 | 1:1.1 | 1 | 86.4 | 91.8 |
| 3 | | 0.8 | 1:1 | 0.5 | 82.3 | 94.6 |
| 4 | | Recycle [b] | 1:1 | 0.6 | 83.9 | 93.3 |
| 5 | PyC8 | 0.5 | 1:1.1 | 0.6 | 90.5 | 89.4 |
| 6 | PyCl4 | 0.5 | 1:1 | 0.4 | 85.5 | 96.7 |
| 7 | PyCl6 | 0.5 | 1:1.1 | 0.6 | 92.7 | 89.5 |
| 8 | PyCl6 | Recycle [c] | 1:1.1 | 1 | 94.8 | 87.9 |
| 9 | PyC4 | 0.8 | 1:2 | 6.5 | 99.5 | 76.5 |
| 10 | PyC4 | 0.8 | 1 [d]:1.1 | 0.4 | 22.6 | 99.6 |

[a] Reaction conditions: Benzyl alcohol (10.8 g, 0.1 mol), catalyst (0.5 to 0.8 g), H$_2$O$_2$ (15%, 25 g, 0.12 mol), reflux, 0.4 to 1.7 h. [b] Catalyst of Entry 4 was recycled. [c] Catalyst of Entry 8 was recycled. [d] Here, the alcohol is cyclohexanol. The catalysts of PyC4, PyC6, PyCl4, and PyCl6 are [n-C$_4$H$_9$(π-C$_5$H$_5$N)]$_4$Mo$_8$O$_{26}$, [n-C$_8$H$_{17}$(π-C$_5$H$_5$N)]$_4$Mo$_8$O$_{26}$, [n-C$_{14}$H$_{29}$(π-C$_5$H$_5$N)]$_4$Mo$_8$O$_{26}$, and [n-C$_{16}$H$_{33}$(π-C$_5$H$_5$N)]$_4$Mo$_8$O$_{26}$, respectively.

Na$_4$H$_3$[SiW$_9$Al$_3$(H$_2$O)$_3$O$_{37}$]·12H$_2$O(SiW$_9$Al$_3$) (**95**) was synthesized and applied as the catalyst for organic-solvent-free selective oxidations of alcohol to ketones using H$_2$O$_2$ without any phase-transfer catalyst under mild, safe, and simple reaction conditions (Table 77) [119]. Benzaldehydes was prepared with selective benzylic alcohol oxidations in moderate to good yields without overoxidation. Additionally, *N*-atom of 2-pyridinemethanol and *S*-atom of 2-thiophenemethanol showed no oxidation.

**Table 77.** The selective oxidation of alcohols with $H_2O_2$ catalyzed by $SiW_9Al_3$ **95** without solvent [a].

R—C$_6$H$_4$—CH$_2$OH → $SiW_9Al_3$ **95** (1.7 µmol), $H_2O_2$ (30%, 30-50 mmol), solvent free, 60-95 °C, 2-12 h. → R—C$_6$H$_4$—CHO

| Entry | Substrate | Product | t/h | Conv. (%) | Sel. (%) |
|-------|-----------|---------|-----|-----------|----------|
| 1 | $4\text{-OMeC}_6\text{H}_4\text{CH}_2\text{OH}$ | $4\text{-OMeC}_6\text{H}_4\text{CHO}$ | 6 | 100 | 99 |
| 2 | $4\text{-MeC}_6\text{H}_4\text{CH}_2\text{OH}$ | $4\text{-MeC}_6\text{H}_4\text{CHO}$ | 7 | 100 | 99 |
| 3 | $C_6\text{H}_5\text{CH}_2\text{OH}$ | $C_6\text{H}_5\text{CHO}$ | 5 | 100 | 99 |
| 4 | $4\text{-ClC}_6\text{H}_4\text{CH}_2\text{OH}$ | $4\text{-ClC}_6\text{H}_4\text{CHO}$ | 6 | 100 | 99 |
| 5 | $4\text{-NO}_2\text{C}_6\text{H}_4\text{CH}_2\text{OH}$ | $4\text{-NO}_2\text{C}_6\text{H}_4\text{CHO}$ | 12 | 74 | 99 |
| 6 | 2-Pyridinemethanol | Picolinaldehyde | 5 | 100 | 99 |
| 7 | Furfuryl alcohol | Furfural | 2 | 100 | 99 |
| 8 | 2-Thiophenemethanol | 2-Thiophenecarboxaldehyde | 4 | 100 | 99 |

[a] Reaction condition: Substrate (20 mmol), $SiW_9Al_3$ **95** (1.7 µmol), $H_2O_2$ (30%, 30 to 50 mmol), solvent free, 60 to 95 °C, 2 to 12 h.

## 2.3. Oxidation of Benzylic and Heterocyclic Alcohols in the Presence of Various Solvents

### 2.3.1. Acetonitrile Solvent

Keggin-type polyoxometalate $[n\text{-}C_4H_9)_4N]_x[PW_{11}ZnO_{39}]\cdot nH_2O$ was successfully immobilized on imidazole functionalized ionic liquid-modified mesoporous MCM-41 by physical adsorption ($PW_{11}Zn@MCM\text{-}41\text{-}Im$) (**96**) [120]. The supported ionic liquid catalyst was easily recovered by simple filtration and reused in four reaction runs. Different alcohols were efficiently oxidized in reflux condition, as shown in Table 78, with high yields and good selectivity. A slight influence on the reaction of benzylic alcohols and electronic nature (electron-withdrawing and electron-donating groups) of the substituent were observed (Table 78, Entries 2–6).

**Table 78.** Oxidation of alcohols with $H_2O_2$ catalyzed by $PW_{11}Zn@MCM\text{-}41\text{-}Im$ **96** under reflux conditions [a].

R—C$_6$H$_4$—CH$_2$OH → $PW_{11}Zn@MCM\text{-}41\text{-}Im$ **96** (50 mg, 0.003 mmol), $H_2O_2$ (30%, 5 mmol), $CH_3CN$ (3 mL) reflux, 10-15 h → R—C$_6$H$_4$—CHO

| Entry | Substrate | Product | t (h) | Yield (%) [b] | TOF (h$^{-1}$) |
|-------|-----------|---------|-------|---------------|----------------|
| 1 | $C_6\text{H}_5\text{CH}_2\text{OH}$ | $C_6\text{H}_5\text{CHO}$ | 10 | 94 | 15.83 |
| 2 | $2\text{-ClC}_6\text{H}_4\text{CH}_2\text{OH}$ | $2\text{-ClC}_6\text{H}_4\text{CHO}$ | 14 | 88 | 10.47 |
| 3 | $4\text{-ClC}_6\text{H}_4\text{CH}_2\text{OH}$ | $4\text{-ClC}_6\text{H}_4\text{CHO}$ | 12 | 94 | 13.09 |
| 4 | $2\text{-BrC}_6\text{H}_4\text{CH}_2\text{OH}$ | $2\text{-BrC}_6\text{H}_4\text{CHO}$ | 15 | 90 | 10 |
| 5 | $4\text{-BrC}_6\text{H}_4\text{CH}_2\text{OH}$ | $4\text{-BrC}_6\text{H}_4\text{CHO}$ | 12 | 92 | 12.77 |
| 6 | $2,4\text{-ClC}_6\text{H}_3\text{CH}_2\text{OH}$ | $2,4\text{-ClC}_6\text{H}_3\text{CHO}$ | 13 | 60 | 7.69 |
| 7 | $2\text{-NO}_2\text{C}_6\text{H}_4\text{CH}_2\text{OH}$ | $2\text{-NO}_2\text{C}_6\text{H}_4\text{CHO}$ | 12 | 75 | 10.41 |
| 8 | $4\text{-NO}_2\text{C}_6\text{H}_4\text{CH}_2\text{OH}$ | $4\text{-NO}_2\text{C}_6\text{H}_4\text{CHO}$ | 12 | 70 | 9.72 |
| 9 | $4\text{-OHC}_6\text{H}_4\text{CH}_2\text{OH}$ | $4\text{-OHC}_6\text{H}_4\text{CHO}$ | 12 | 82 | 11.08 |
| 10 | $2\text{-OMeC}_6\text{H}_4\text{CH}_2\text{OH}$ | $2\text{-OMeC}_6\text{H}_4\text{CHO}$ | 12 | 92 | 12.77 |
| 11 | $4\text{-OMeC}_6\text{H}_4\text{CH}_2\text{OH}$ | $4\text{-OMeC}_6\text{H}_4\text{CHO}$ | 14 | 94 | 9.64 |
| 12 | $2\text{-MeC}_6\text{H}_4\text{CH}_2\text{OH}$ | $2\text{-MeC}_6\text{H}_4\text{CHO}$ | 12 | 94 | 11.05 |
| 13 | $4\text{-Tert-butylC}_6\text{H}_4\text{CH}_2\text{OH}$ | $4\text{-Tert-butylC}_6\text{H}_4\text{CHO}$ | 11 | 65 | 9.84 |

[a] Reaction conditions: Substrates (0.5 mmol), $PW_{11}Zn@MCM\text{-}41\text{-}Im$ **96** (50 mg, 0.003 mmol), $H_2O_2$ (30%, 5 mmol), $CH_3CN$ (3 mL), reflux, 10 to 15 h. [b] GC yield based on the starting alcohol.

Three paramagnetic metal complexes of 3-hydroxy-3,3′-biindoline-2,2′-dione (dihydroindolone, $H_4ID$) ($MH_2ID$) with $Ni^{2+}$, $Cu^{2+}$, and $VO^{2+}$ ions with were synthesized and applied in oxidation

reactions using aqueous $H_2O_2$ in acetonitrile. Although all the $NiH_2ID$ (**97**), $CuH_2ID$ (**98**), and $VOH_2ID$ (**99**) catalysts showed good catalytic activity, chemo-, and regioselectivity, $VOH_2ID$ had the highest potential due to more Lewis acid character and the high oxidation number of the central $V^{4+}$ ion in **99** as compared with both $Ni^{2+}$- and $Cu^{2+}$-species in the same homogenous aerobic atmosphere [121]. As illustrated in Table 79, **99** afforded the best amount of the selective product such as benzaldehyde (85%) [122].

**Table 79.** Oxidation process catalyzed by $VOH_2ID$ **99** using an aqueous $H_2O_2$ in acetonitrile [a].

| Entry | Substrate | Product | t/h | Conv. (%) | | | Conv. (%) | Sel. (%) | TON [d] | TOF [e] |
|---|---|---|---|---|---|---|---|---|---|---|
| | | | | R [b] (Residual) | P [c] | Side Products | | | | |
| 1 | | | 1 | 50 | 47 | 3 | 50 | 94 | 23.5 | 23.5 |
| 2 | $C_6H_5CH_2OH$ | $C_6H_5CHO$ | 2 | 28 | 64 | 8 | 72 | 88 | 32 | 16 |
| 3 | | | 3 | 3 | 85 | 12 | 97 | 87 | 42.5 | 14.1 |
| 4 | | | 4 | 0 | 71 | 29 | 100 | 71 | 35.5 | 8.87 |

[a] Reaction conditions: Benzyl alcohol (R) (1 mmol), $VOH_2ID$ **99** (0.02 mmol), $H_2O_2$ (3 mmol), $CH_3CN$ (10 mL), 85 °C, 4 h. [b] Residual amount (%) of the reactant after finishing the catalytic process (R). [c] Selectivity percentage of benzaldehyde (P), and the other side products. [d] TON (turn over number) is the ratio of moles of product obtained to the moles of catalyst. [e] Corresponding TOF (turn over frequency) (TON/h) is shown as mol (mol catalyst)$^{-1}$ h$^{-1}$.

Iron catalysts supported on porous furfuryl alcohol derived resins were synthesized and applied for the selective comparison of benzyl alcohol to benzaldehyde. As shown in Table 80, $FeCl_2$ (0.5 mol%) as a reference reaction was tested for the oxidation of 0.77 M benzyl alcohol with 2.3 eq. $H_2O_2$ in acetonitrile [123]. After microwave irradiation (5 min), 53% conversion with 78% selectivity to benzaldehyde was detected. Loading iron catalysts supported on P420 (Fe/P420 (**100**)) or P500resin (Fe/P500 (**101**)) with 10 times lower Fe content (0.05 mol%), similar selectivity with lower conversion in the product mixture was obtained. (Tables 80 and 81) [123]. In addition, the selectivity in the final reaction mixture by preventing further oxidation to benzoic acid was optimized by continuous addition of $H_2O_2$ between 1.5 and 2.3 eq.

**Table 80.** Benzyl alcohol conversion and selectivity using Fe nanoparticles supported on P420 resin **100** as compared with free dissolved $FeCl_2$, using 1.5 and 2.3 equiv. $H_2O_2$ [a].

| Entry | Substrate | Product | Catalyst (mol%) | | t/min | $H_2O_2$ (equiv.) | Conv. (%) | Sel. (%) |
|---|---|---|---|---|---|---|---|---|
| | | | Fe/P420 | $FeCl_2$ | | | | BzH |
| 1 | | | - | 0.5 | 5 | 2.3 | 53 | 78 |
| 2 | | | 0.03 | - | 5 | 2.3 | | |
| 3 | $C_6H_5CH_2OH$ | $C_6H_5CHO$ | 0.05 | - | 5 | 2.3 | 22 | 78 |
| 4 | | | 0.23 | - | 1.2 | 2.3 | 20 | |
| 5 | | | 0.05 | - | 5 | 1.5 | | |
| 6 | | | 0.12 | - | 2.3 | 1.5 | | |

[a] Reaction conditions: Substrate (0.77 M, 0.2 mL, 1.92 mmol), $FeCl_2$ (0.5 mol%), Fe/P420 **100** (0.3 to 0.0.23 mol%), $H_2O_2$ (1.5 and 2.3 equiv., 0.3 mL of a 33 or 50 *w/v*% aqueous $H_2O_2$ solution), $CH_3CN$ (2 mL), microwave irradiation (300 W), autogenous pressure (5 to 17 bar with an average of 16 bar), 90 to 132 °C, 1.2 to 5 min.

**Table 81.** Benzyl alcohol conversion and selectivity using Fe nanoparticles supported on P500 **101** as compared with dissolved FeCl$_2$, using 1.5 and 2.3 equiv. H$_2$O$_2$ [a].

$$
\begin{array}{c}
\text{FeCl}_2 \ (0.5 \ \text{mol\%}) \\
\text{Fe/P500 } \mathbf{101} \ (0.3\text{-}0.0.12 \ \text{mol\%}) \\
\text{C}_6\text{H}_5\text{CH}_2\text{OH} + \text{H}_2\text{O}_2 \ (33 \text{ or } 50 \text{ w/v\%}, 1.5 \text{ and } 2.3 \text{ equiv.}) \longrightarrow \text{C}_6\text{H}_5\text{CHO} \\
\overline{\text{CH}_3\text{CN} \ (2 \ \text{mL}), \text{ microwave irradiation (300 W)}} \\
\text{autogenous pressure (5 to 17 bar)} \\
90\text{-}132 \ ^\circ\text{C}, 1.2\text{-}5 \ \text{min.}
\end{array}
$$

| Entry | Substrate | Product | Catalyst (mol%) Fe/P500 | Catalyst (mol%) FeCl$_2$ | t/min | H$_2$O$_2$ (equiv.) | Conv. (%) |
|---|---|---|---|---|---|---|---|
| 1 | | | - | 0.5 | 5 | 2.3 | |
| 2 | | | 0.03 | - | 5 | 2.3 | 6–9 |
| 3 | C$_6$H$_5$CH$_2$OH | C$_6$H$_5$CHO | 0.05 | - | 5 | 2.3 | |
| 4 | | | 0.05 | - | 5 | 1.5 | |
| 5 | | | 0.12 | - | 2.3 | 1.5 | |
| 6 | | | 0.23 | - | 1.5 | 1.5 | |

[a] Reaction conditions: Substrate (0.77 M, 0.2 mL, 1.92 mmol), FeCl$_2$ (0.5 mol%), Fe/P500 **101** (0.3 to 0.0.12 mol%), H$_2$O$_2$ (1.5 and 2.3 equiv., 0.3 mL of a 33 or 50 *w/v*% aqueous H$_2$O$_2$ solution), CH$_3$CN (2 mL), microwave irradiation (300 W), autogenous pressure (5 to 17 bar with an average of 16 bar), 90 to 132 °C, 2.3 to 5 min.

Metal molybdates were coupled with Co, Ni, and Cu with Mo, respectively, for the preparation of bimetallic complexes which was used successfully in alcohol oxidation reactions by Xinhua et al., in 2018 [124]. The 100% selectivity in conversion of the alcohols to corresponding carbonyl compounds showed noticeable performance of theses catalysts which could be reused three times. The results represented that CoMoO$_4$ (**102**) catalyst provided lower conversion and much higher selectivity (approximately 100%). In addition, non-terminal alcohols could be oxidized to ketones sufficiently (Table 82, Entries 2 and 3). Although NiMoO$_4$ (**103**) had near 100% selectivity, Cu$_4$Mo$_3$O$_{12}$ (**104**) provided no product (Table 82, Entry 1) due to the existence of a bimetallic combination effect in the oxidation reactions.

**Table 82.** Selective oxidation of different alcohols [a].

$$
\begin{array}{c}
\text{R-C}_6\text{H}_4\text{CH}_2\text{OH} \\
\text{R-C}_6\text{H}_4\text{CH(OH)R'}
\end{array}
\xrightarrow[\substack{\text{CH}_3\text{CN (5 mL)} \\ 70 \ ^\circ\text{C}, 18 \ \text{h}}]{\substack{\text{CoMoO}_4 \ \mathbf{102} \ (0.05 \ \text{g}, 23 \ \text{mol\%}) \\ \text{H}_2\text{O}_2 \ (30\%, 0.23 \ \text{g}, 2 \ \text{mmol})}}
\begin{array}{c}
\text{R-C}_6\text{H}_4\text{CHO} \\
\text{R-C}_6\text{H}_4\text{COR'}
\end{array}
$$

| Entry | Substrate | Product | Conv. (%) [b] | Yield (%) [b] |
|---|---|---|---|---|
| 1 | C$_6$H$_5$CH$_2$OH | C$_6$H$_5$CHO | 43<br>37 [c,d]<br>- [d] | 43<br>36 [c,d]<br>- [d] |
| 2 | C$_6$H$_5$CH(OH)CH$_3$ | C$_6$H$_5$COCH$_3$ | 37 | 37 |
| 3 [c] | (C$_6$H$_5$)$_2$CHOH | (C$_6$H$_5$)$_2$CO | 30 | 30 |
| 4 | 4-BrC$_6$H$_4$CH$_2$OH | 4-BrC$_6$H$_4$CHO | 46 | 46 |
| 5 [e] | 1-Naphthylmethanol | 1-Naphthaldehyde | 13 | 13 |
| 6 [e] | 4-NO$_2$C$_6$H$_4$CH$_2$OH | 4-NO$_2$C$_6$H$_4$CHO | 25 | 25 |

[a] Reaction conditions: Substrate (1 mmol), CoMoO$_4$ **102** (0.05 g, 23 mol%), H$_2$O$_2$ (30%, 0.23 g, 2 mmol), CH$_3$CN (5 mL), 70 °C, 18 h. [b] Calculated by GC with nonane as the internal standard. [c] With NiMoO$_4$ as the catalyst (0.05 g, 23 mol%). [d] With Cu$_4$Mo$_3$O$_{12}$ as the catalyst (0.04 g, 6 mol%). [e] Isolated by column chromatography.

Baghbanian et al. [125] immobilized 12-tungstophosphoric acid onto poly(N-vinylimidazole) which was modified by magnetic nanozeolite (MNZ@PVImW) (**105**). The results of successful catalytic oxidations of benzylic alcohols in the presence of synthesized catalyst under optimized reaction conditions are summarized in Table 83. In addition, without loss of catalytic activity, the catalyst could be reused eight times.

**Table 83.** Oxidation of various alcohols catalyzed by MNZ@PVImW **105** [a].

| Entry | Substrate | Product | | t/min | Conv. (%) [b] | Yield (%) |
|---|---|---|---|---|---|---|
| | | **A** | **B** | | | |
| 1 | $C_6H_5CH_2OH$ | $C_6H_5CHO$ | $C_6H_5CO_2H$ | 30 | 96 | 95 |
| 2 | $4\text{-}MeC_6H_4CH_2OH$ | $4\text{-}MeC_6H_4CHO$ | $4\text{-}MeC_6H_4CO_2H$ | 25 | 92 | 92 |
| 3 | $2\text{-}MeC_6H_4CH_2OH$ | $2\text{-}MeC_6H_4CHO$ | $2\text{-}MeC_6H_4CO_2H$ | 25 | 91 | 90 |
| 4 | $4\text{-}OMeC_6H_4CH_2OH$ | $4\text{-}OMeC_6H_4CHO$ | $4\text{-}OMeC_6H_4CO_2H$ | 25 | 96 | 95 |
| 5 | $3,4\text{-}OMeC_6H_3CH_2OH$ | $3,4\text{-}OMeC_6H_3CHO$ | $3,4\text{-}OMeC_6H_3CO_2H$ | 20 | 91 | 90 |
| 6 | $4\text{-}NO_2C_6H_4CH_2OH$ | $4\text{-}NO_2C_6H_4CHO$ | $4\text{-}NO_2C_6H_4CO_2H$ | 60 | 82 | 80 |
| 7 | $3\text{-}NO_2C_6H_4CH_2OH$ | $3\text{-}NO_2C_6H_4CHO$ | $3\text{-}NO_2C_6H_4CO_2H$ | 90 | 79 | 75 |
| 8 | $4\text{-}ClC_6H_4CH_2OH$ | $4\text{-}ClC_6H_4CHO$ | $4\text{-}ClC_6H_4CO_2H$ | 45 | 92 | 91 |
| 9 | $3\text{-}ClC_6H_4CH_2OH$ | $3\text{-}ClC_6H_4CHO$ | $3\text{-}ClC_6H_4CO_2H$ | 45 | 92 | 90 |
| 10 | $4\text{-}BrC_6H_4CH_2OH$ | $4\text{-}BrC_6H_4CHO$ | $4\text{-}BrC_6H_4CO_2H$ | 40 | 92 | 91 |
| 11 | $3\text{-}BrC_6H_4CH_2OH$ | $3\text{-}BrC_6H_4CHO$ | $3\text{-}BrC_6H_4CO_2H$ | 40 | 90 | 89 |
| 12 | $C_6H_5CH(OH)CH_3$ | $C_6H_5COCH_3$ | - | 30 | 90 | 88 |
| 13 | $(C_6H_5)_2CHOH$ | $(C_6H_5)_2CO$ | - | 45 | 92 | 90 |
| 14 | $4\text{-}OMeC_6H_4CH(OH)C_6H_5$ | $4\text{-}OMeC_6H_4COC_6H_5$ | - | 40 | 91 | 90 |
| 15 | $4\text{-}ClC_6H_4CH(OH)C_6H_5$ | $4\text{-}ClC_6H_4COC_6H_5$ | - | 40 | 88 | 85 |

[a] Reaction conditions: Substrate (1 mmol), MNZ@PVImW **105** (2 mg), $H_2O_2$ (4 mmol), $CH_3CN$ (3 mL), 40 °C, 20 to 90 min. [b] Conversions were calculated based on initial mmol of benzyl alcohol.

A simple and efficient catalytic oxidation of benzylic and hetero-aryl alcohols to their corresponding carbonyl compound using recoverable heterobimetallic sodium-dioxidovanadium (V) complexes (**106**) was reported (Table 84) [126].

**Table 84.** Catalytic oxidation of alcohols by heterobimetallic vanadium(V) complexes **106** [a].

| Entry | Substrate | Product | t/h | Yield (%) [b] | TON | TOF (h$^{-1}$) | Yield (%) [c] |
|---|---|---|---|---|---|---|---|
| 1 | $C_6H_5CH_2OH$ | $C_6H_5CHO$ | 3.4 | 89 | 445 | 130.88 | 93 |
| 2 | $4\text{-}OMeC_6H_4CH_2OH$ | $4\text{-}OMeC_6H_4CHO$ | 4 | 91 | 455 | 113.75 | 94 |
| 3 | $4\text{-}NO_2C_6H_4CH_2OH$ | $4\text{-}NO_2C_6H_4CHO$ | 4.3 | 80 | 400 | 93.02 | 87 |
| 4 | $C_6H_5CH(OH)CH_3$ | $C_6H_5COCH_3$ | 6 | No reaction | - | - | - |
| 5 | $C_6H_5CH(OH)Et$ | $C_6H_5COEt$ | 7.2 | No reaction | - | - | - |
| 6 | $3\text{-}ClC_6H_4CH_2OH$ | $3\text{-}ClC_6H_4CHO$ | 4.5 | 74 | 370 | 82.22 | 79 |
| 7 | $4\text{-}MeC_6H_4CH_2OH$ | $4\text{-}MeC_6H_4CHO$ | 4.5 | 88 | 400 | 97.78 | 90 |

**Table 84.** *Cont.*

| Entry | Substrate | Product | t/h | Yield (%) [b] | TON | TOF (h$^{-1}$) | Yield (%) [c] |
|-------|-----------|---------|-----|----------|-----|----------|----------|
| 8 | 3-MeC$_6$H$_4$CH$_2$OH | 3-MeC$_6$H$_4$CHO | 5 | 86 | 430 | 86 | 88 |
| 9 | 2-MeC$_6$H$_4$CH$_2$OH | 2-MeC$_6$H$_4$CHO | 5 | 84 | 420 | 84 | 86 |
| 10 | C$_6$H$_5$CH(OH)COC$_6$H$_5$ | (C$_6$H$_5$)$_2$(CO)$_2$ | 4 | 86 | 430 | 107.5 | 88 |
| 11 | Furfuryl alcohol | Furfural | 3 | 85 | 425 | 141.67 | 89 |
| 12 | 2-Pyridinemethanol | Picolinaldehyde | 4 | 89 | 445 | 111.25 | 92 |
| 13 | 2-Thiophenemethanol | 2-Thiophenecarboxaldehyde | 2.3 | 78 | 390 | 169.57 | 82 |
| 14 | 2,3,4-OMeC$_6$H$_2$CH$_2$OH | 2,3,4-OMeC$_6$H$_2$CHO | 3.5 | 94 | 470 | 134.29 | 96 |

[a] Reaction condition: Substrate (4.84 mmol), heterobimetallic dioxido-vanadium(V) complexes **106** (0.01 mmol), H$_2$O$_2$ (15%, 4.85 mmol), 70 °C, 2.3 to 7.2 h. [b] Isolated yield. [c] GC Yield.

The hydrothermal method was applied for the preparation of MCM-41 nanostructure-modified with vanadium, iron, and cobalt. Under optimized conditions, a molar ratio substrate/oxidant of 4/1 and 7 h of reaction, as summarized in Table 85, BzH was the main product with 7% and 12% yield for V-M(60) (**107**) and Fe-M(60) (**108**), respectively [127]. High TON (1100 mol/mol V), 95% selectivity to BzH, and 31.7% yield made V-M(60) performance better than **108** and Co-M(60) (**109**) due to effective dispersion of vanadium species in the framework that could be considered as the active sites for the oxidation reaction. Moreover, the catalyst could be effectively reused and recovered for at least three cycles without loss of its activity and selectivity.

**Table 85.** Benzyl alcohol oxidation with H$_2$O$_2$ on M-M(x) catalysts under standards conditions [a].

| Entry | Sample | Conv. of BzOH | Sel. (%) | | | BzH Yield (%) | TON |
|-------|--------|---------------|----------|-----|-----|---------------|-----|
| | | | BzH | BzA | BzB | | |
| 1 | V-M(60) | 7.1 [b] (28.3) [c] | 100 | - | - | 7.1 | 235 |
| 2 | Fe-M(60) | 13.3 [b] (53.2) [c] | 90.5 | 6.0 | 3.5 | 12 | 50.5 |
| 3 | Co-M(60) | 0.9 [b] (3.5) [c] | 100 | - | - | 0.9 | 2.1 |

[a] Reaction conditions: BzOH/H$_2$O$_2$ molar ratio (4:1), M-M(x) (100 mg), CH$_3$CN (91.15 mmol, Sintorgan, 99.5%), 70 °C, 7 h. [b] BzOH conversion (%). [c] BzOH conversion (mol% of max).

Using a synthetic manganese catalyst for catalytic oxidation and oxidative kinetic resolution (OKR) of secondary alcohols in the presence of an environmentally benign oxidant hydrogen peroxide and a small amount of additive sulfuric acid provided the high yields of products (up to 93%) with excellent enantioselectivity (>90% *ee* in the OKR of secondary alcohols). Moreover, **111** provided a higher enantiomeric excess (90% *ee*) than **110** (65%) in 1-phenylethanol oxidation (Table 86, Entry 1 and footnote c) [128].

A variety of picolinic/quinaldinic acids as ligands mixed in situ with Fe(OAc)$_2$ (**112**) catalyzed the H$_2$O$_2$ oxidation of 1-phenylethanol especially with 6-methylpicolinic acid (6-MepicH) and 4-chloropicolinic acid (4-ClpicH) (Table 87) [129]. The oxidation possesses proceeded using 35% aq. H$_2$O$_2$ in CH$_3$CN solution with isolated Fe complexes (**113**) as the catalyst (Table 88) [129]. In addition, the results showed that the redox potential of Fe III and the lability of picolinate or quinaldinate ligand were important factors for the catalytic reaction.

**Table 86.** Substrate scope of oxidative kinetic resolution of secondary alcohols using the **111**/$H_2O_2$/$H_2SO_4$ catalytic system. [a,b]

**110:** Ar = Ph
**111:** Ar = 3,5-di-$^t$Bu-Ph

| Entry | Substrate | Product | $H_2O_2$ (equiv.) | Conv. (%) | *ee* (%) |
|---|---|---|---|---|---|
| 1 [c] | $C_6H_5CH(OH)CH_3$ | $C_6H_5COCH_3$ | 0.8 | 69 | 90 |
| 2 | 4-$ClC_6H_4CH(OH)CH_3$ | 4-$ClC_6H_4COCH_3$ | 0.8 | 74 | 92 |
| 3 | 3-$ClC_6H_4CH(OH)CH_3$ | 3-$ClC_6H_4COCH_3$ | 0.9 | 69 | 92 |
| 4 | 2-$ClC_6H_4CH(OH)CH_3$ | 2-$ClC_6H_4COCH_3$ | 0.9 | 68 | 96 |
| 5 [d] | 4-$NO_2C_6H_4CH(OH)CH_3$ | 4-$NO_2C_6H_4COCH_3$ | 0.9 | 60 | 96 |
| 6 [d] | 4-$PhC_6H_4CH(OH)CH_3$ | 4-$PhC_6H_4COCH_3$ | 0.9 | 65 | 93 |
| 7 | $C_6H_5CH(OH)Et$ | $C_6H_5COEt$ | 0.8 | 71 | 92 |
| 8 | 4-$BrC_6H_4CH(OH)Et$ | 4-$BrC_6H_4COEt$ | 0.8 | 69 | 96 |
| 9 | 4-$ClC_6H_4CH(OH)Et$ | 4-$ClC_6H_4COEt$ | 0.7 | 62 | 90 |
| 10 | 4-$FC_6H_4CH(OH)Et$ | 4-$FC_6H_4COEt$ | 0.8 | 68 | 96 |
| 11 | 3-$FC_6H_4CH(OH)Et$ | 3-$FC_6H_4COEt$ | 0.9 | 69 | 94 |
| 12 | 2-$FC_6H_4CH(OH)Et$ | 2-$FC_6H_4COEt$ | 0.8 | 61 | 90 |
| 13 | $C_6H_5CH(OH)iPr$ | $C_6H_5COiPr$ | 0.8 | 66 | 92 |
| 14 | $C_6H_5CH(OH)c$-Hex | $C_6H_5COc$-Hex | 0.8 | 65 | 92 |
| 15 | $C_6H_5CH(OH)(CH_2)_5Me$ | $C_6H_5CO(CH_2)_5Me$ | 0.8 | 64 | 93 |

[a] Reaction conditions: An $CH_3CN$ (0.5 mL) solution containing $H_2O_2$ (30%, 0.70 to 0.90 equiv.) was added dropwise into an $CH_3CN$ (1 mL) solution containing secondary alcohol (0.5 mmol), **111** (0.2 mol%), and $H_2SO_4$ (1 mol%) by using a syringe pump at 0 °C for 1 h. [b] Conversion yields and ee values were determined by GC with a CP-Chirasil-Dex CB column. [c] When **110** was used as a catalyst under the identical reaction conditions, the conversion yields and *ee* values were 66% and 65%, respectively. [d] Conversion yields were calculated from the isolated products and the ee values were determined by HPLC with IA column.

**Table 87.** Iron-catalyzed oxidation of 1-phenylethanol by various combinations of picolinic/quinaldinic acids [a].

| Entry | Picolinic | Quinaldinic Acids | Conv. (%) [b] | Yield (%) [b] | Sel. (%) [c] |
|-------|-----------|-------------------|---------------|---------------|--------------|
| 1 | 6-MepicH | picH | 78 | 71 | 91 |
| 2 | 6-MepicH | 4-OMepicH | 74 | 69 | 93 |
| 3 | 6-MepicH | 4-MepicH | 70 | 63 | 93 |
| 4 | 6-MepicH | 4-ClpicH | 85<br>94 [d] | 81<br>93 [d]<br>91 [e] | 95<br>99 [d] |
| 5 | qnH | picH | 46 | 39 | 85 |
| 6 | qnH | 4-OMepicH | 35 | 25 | 71 |
| 7 | qnH | 4-MepicH | 39 | 31 | 79 |
| 8 | qnH | 4-ClpicH | 73 | 65 | 89 |
| 9 [f] | picH | - | 57 | 33 | 59 |
| 10 [f] | 4-OMepicH | - | 13 | 8 | 59 |
| 11 [f] | 4-MepicH | - | 44 | 40 | 91 |
| 12 [f] | 4-ClpicH | - | 39 | 36 | 93 |
| 13 [f] | 6-MepicH | - | 25 | 20 | 77 |
| 14 [f] | qnH | - | 9 | 6 | 60 |

[a] Reaction conditions: 1-Phenylethanol (12 g, 0.1 mol), Fe(OAc)$_2$ **112** (0.5 equiv.), picolinic/quinaldinic acids (0.5 equiv. + 0.5 equiv.), H$_2$O$_2$ (35%, 2 equiv.), CH$_3$CN (2 mL), 25 °C, 30 min, and further stirring for 5 min, unless otherwise stated. [b] Determined by GC by using biphenyl as an internal standard. Average of two runs. [c] Yield/conversion × 100. [d] H$_2$O$_2$ (35%) in CH$_3$CN was used. [e] Yield of isolated product, 10 g scale. [f] 0.10 equiv. of acid were used.

The coprecipitation approach was applied for hybrid chromium (VI)-based magnetic nanocomposite catalyst (Fe$_3$O$_4$@SiO$_2$@PPh$_3$@Cr$_2$O$_7^{2-}$ (**114**)) preparation by Maleki et al., in 2016 [130]. This catalyst was used for the first report of using magnetic nanocomposites with ultrasonic irradiation for the oxidation of benzyl alcohol to benzaldehyde with hydrogen peroxide at room temperature (Table 89). Total conversion of benzyl alcohol to benzaldehyde and the reusability of the catalyst at least five times were the benefit of this system.

Yadollahi et al. [131] first reported using transition metal-substituted polyoxometalate [PW$_{11}$ZnO$_{39}$]$^{5-}$ (PW$_{11}$Zn) (POMs) supported on activated carbon (AC) for oxidation of alcohols in the presence of H$_2$O$_2$ and CH$_3$CN as solvent (Table 90). In addition, with high selectivity, the oxidation of *p*-hydroxybenzyl alcohol without oxidation of the hydroxyl group was carried out (Table 90, Entry 13) and showed no change in the catalytic activity/selectivity of PW$_{11}$Zn@AC (**115**) for at least five catalytic cycles of sequence loading.

**Table 88.** Catalytic oxidation of 1-phenylethanol by isolated Fe complexes **113** [a].

| Entry | Fe complex | Conv. (%)[b] | Yield (%) [b] | Sel. (%) [c] |
|-------|-----------|--------------|---------------|--------------|
| 1 | [Fe(6-Mepic)$_2$(pic)] | 69 | 59 | 85 |
| 2 | [Fe(6-Mepic)$_2$(4-Mepic)] | 66 | 56 | 85 |
| 3 | [Fe(6-Mepic)$_2$(4-Clpic)] | 76 (67) [d] | 65 (63) [d] | 86 (95) [d] |
| 4 | [Fe(qn)$_2$(4-Mepic)] | 39 | 33 | 85 |
| 5 | [Fe(qn)$_2$(4-Clpic)] | 76 | 64 | 84 |
| 6 | [Fe(4-OMepic)$_3$] | 24 | 4 | 17 |
| 7 | [Fe(4-Mepic)$_3$] | 8 | 6 | 75 |
| 8 | [Fe(4-Clpic)$_3$] | 14 | 10 | 71 |
| 9 | [Fe(pic)$_3$] | 24 | 13 | 54 |
| 10 | [Fe(6-Mepic)$_3$] | 22 | 17 | 74 |

[a] Reaction conditions: 1- Phenylethanol (12 g, 0.1 mol), Fe complex **113** (0.05 equiv.), H$_2$O$_2$ (35%, 2 equiv.), CH$_3$CN (2 mL), 25 °C, 30 min, and further stirring for 5 min, unless otherwise stated. [b] Determined by GC using biphenyl as an internal standard. Average of two runs. [c] Yield/conversion × 100. [d] 0.1 equiv. of CH$_3$COOH was added.

**Table 89.** The oxidations of benzyl alcohol to benzaldehyde in the presence of Fe$_3$O$_4$@SiO$_2$@PPh$_3$@Cr$_2$O$_7$$^{2-}$ nanocomposite **114** [a].

| Entry | Substrate | Product | Conv. (%) [b] | TON | TOF (min$^{-1}$) |
|-------|-----------|---------|---------------|-----|------------------|
| 1 | C$_6$H$_5$CH$_2$OH | C$_6$H$_5$CHO | 100 | 416 | 21.83 |

[a] Reaction conditions: Substrate (5 mmol), Fe$_3$O$_4$@SiO$_2$@PPh$_3$@Cr$_2$O$_7$$^{2-}$ **114** (0.05 g), H$_2$O$_2$ (30%, 0.6 mL), CH$_3$CN (2 mL), room temperature, 20 min, ultrasonic irradiation. [b] Obtained by gas chromatography mass spectrometry analysis.

Two thiosemicarbazide Schiff bases, 1-(4-dimethylaminobenzylidene) thiosemicarbazide (ABTSC) and 1-(2-pyridincarboxyl-idene) thiosemicarbazide (TCTS), were applied for the coordination with Co(II), Ni(II), Zn(II), Cd(II), and Ag(I) transition metal salts, chloride, and acetate, and then under optimized conditions proceeded successfully for the oxidation of benzylic alcohols, Table 91 [132]. The environmentally friendly oxidation catalytic systems exhibited great activities in successive runs without loss in activity. The best results under the optimum conditions were obtained with Co(ABTSC)$_2$(OAc)$_2$ (**116**) catalyst as in 4-methoxybenzyl alcohol oxidation to aldehyde with 95% conversion and 100% benzaldehyde selectivity.

**Table 90.** Oxidation of various alcohols with hydrogen peroxide catalyzed by PW$_{11}$Zn@AC **115** [a].

$$R\text{-}C_6H_4CH_2OH \xrightarrow[\text{H}_2\text{O}_2 (30\%,1 \text{ mL}), \text{CH}_3\text{CN} (3 \text{ mL})]{\text{PW}_{11}\text{Zn@AC } \textbf{115} (5 \text{ }\mu\text{mol})} R\text{-}C_6H_4CHO$$

reflux, 25-240 min

| Entry | Substrate | Product | t/min [b] | TOF (h$^{-1}$) [c] |
|-------|-----------|---------|-----------|---------------------|
| 1 | C$_6$H$_5$CH$_2$OH | C$_6$H$_5$CHO | 45 | 267 |
| 2 | 2-MeC$_6$H$_4$CH$_2$OH | 2-MeC$_6$H$_4$CHO | 35 | 343 |
| 3 | 4-MeC$_6$H$_4$CH$_2$OH | 4-MeC$_6$H$_4$CHO | 25 | 480 |
| 4 | 2-OMeC$_6$H$_4$CH$_2$OH | 2-OMeC$_6$H$_4$CHO | 35 | 343 |
| 5 | 4-OMeC$_6$H$_4$CH$_2$OH | 4-OMeC$_6$H$_4$CHO | 25 | 480 |
| 6 | 3-OMeC$_6$H$_4$CH$_2$OH | 3-OMeC$_6$H$_4$CHO | 70 | 171 |
| 7 | 2-NO$_2$C$_6$H$_4$CH$_2$OH | 2-NO$_2$C$_6$H$_4$CHO | 180 | 67 |
| 8 | 4-NO$_2$C$_6$H$_4$CH$_2$OH | 4-NO$_2$C$_6$H$_4$CHO | 240 | 50 |
| 9 | 4-BrC$_6$H$_4$CH$_2$OH | 4-BrC$_6$H$_4$CHO | 50 | 240 |
| 10 | 4-FC$_6$H$_4$CH$_2$OH | 4-FC$_6$H$_4$CHO | 25 | 480 |
| 11 | 4-ClC$_6$H$_4$CH$_2$OH | 4-ClC$_6$H$_4$CHO | 35 | 343 |
| 12 | 2,4-ClC$_6$H$_3$CH$_2$OH | 2,4-ClC$_6$H$_3$CHO | 90 | 133 |
| 13 | 4-OHC$_6$H$_4$CH$_2$OH | 4-OHC$_6$H$_4$CHO | 50 | 240 |
| 14 | 3-Br-2-HOC$_6$H$_3$CH$_2$OH | 3-Br-2-HOC$_6$H$_3$CH$_2$OH | 25 | 480 |

[a] Reaction conditions: Substrate (1 mmol), PW$_{11}$Zn@AC **115** (5 µmol), H$_2$O$_2$ (30%, 1 mL), CH$_3$CN (3 mL), reflux, 25 to 240 min. [b] Yields are quantitative and refer to GC yields. [c] Turnover frequency.

**Table 91.** Oxidation benzyl alcohol derivatives for complex [Co(ABTSC)$_2$(OAc)$_2$] **116** [a].

$$R\text{-}C_6H_4CH_2OH \xrightarrow[\text{H}_2\text{O}_2 (30\%, 3 \text{ mmol}), \text{CH}_3\text{CN} (3 \text{ mL})]{[\text{Co(ABTSC)}_2(\text{OAc})_2] \textbf{116} (0.03 \text{ mmol})} R\text{-}C_6H_4CHO$$

82 °C, 80 min

**116** M= Co, X= OAc

| Entry | Substrate | Product | Yield (%) | Sel. (%) |
|-------|-----------|---------|-----------|----------|
| 1 | C$_6$H$_5$CH$_2$OH | C$_6$H$_5$CHO | 85 | 100 |
| 2 | 2-NO$_2$C$_6$H$_4$CH$_2$OH | 2-NO$_2$C$_6$H$_4$CHO | 65 | 97 |
| 3 | 3-NO$_2$C$_6$H$_4$CH$_2$OH | 3-NO$_2$C$_6$H$_4$CHO | 65 | 97 |
| 4 | 4-NO$_2$C$_6$H$_4$CH$_2$OH | 4-NO$_2$C$_6$H$_4$CHO | 75 | 100 |
| 5 | 2-OMeC$_6$H$_4$CH$_2$OH | 2-OMeC$_6$H$_4$CHO | 85 | 100 |
| 6 | 3-OMeC$_6$H$_4$CH$_2$OH | 3-OMeC$_6$H$_4$CHO | 90 | 100 |
| 7 | 4-OMeC$_6$H$_4$CH$_2$OH | 4-OMeC$_6$H$_4$CHO | 95 | 100 |

[a] Reaction conditions: Substrate (1 mmol), [Co(ABTSC)$_2$(OAc)$_2$] **116** (0.03 mmol), H$_2$O$_2$ (30%, 3 mmol), CH$_3$CN (3 mL), 82 °C, 80 min.

Nanocrystalline cobalt aluminate ($CoAl_2O_4$), using both conventional (sample A: $CoAl_2O_4$-CCM) (**117**) and microwave combustion method (sample B: $CoAl_2O_4$-MCM) (**118**), was synthesized for comparative investigation in various alcohol oxidations to corresponding carbonyl compounds, as shown in Tables 92 and 93 [133]. **117** showed weaker ferromagnetic in nature as compared with $CoAl_2O_4$-MCM and both also showed a lower conversion rate in the oxidation of benzyl alcohol [134]. This result was confirmed more powerful for **118** and similar results were achieved within a shorter time. In addition, both catalytic systems catalytic were reusable activity/selectivity during at least five catalytic cycles.

**Table 92.** Oxidation of substituted alcohols to aldehydes using cobalt aluminate ($CoAl_2O_4$) (sample A) **117** under the optimum conditions [a].

| Entry | Substrate | Product | Conv. (%) | Sel. (%) |
|-------|-----------|---------|-----------|----------|
| 1 | 4-OMeC$_6$H$_4$CH$_2$OH | 4-OMeC$_6$H$_4$CHO | 74.11 | 81.34 |
| 2 | 4-NO$_2$C$_6$H$_4$CH$_2$OH | 4-NO$_2$C$_6$H$_4$CHO | 63.09 | 79.12 |
| 3 | C$_6$H$_5$CH(OH)CH$_3$ | C$_6$H$_5$COCH$_3$ | 63.45 | 83.25 |
| 4 | C$_6$H$_5$CH(OH)Et | C$_6$H$_5$COEt | 58.34 | 80.94 |
| 5 | C$_6$H$_5$CH$_2$OH | C$_6$H$_5$CHO | 80.91 | 98.68 |

[a] Reaction conditions: Substrate (5 mmol), $CoAl_2O_4$ (sample A) **117** (0.5 g), $H_2O_2$ (5 mmol), $CH_3CN$ (5 mmol), 80 °C, 8 h.

**Table 93.** Oxidation of substituted alcohols to aldehydes using cobalt aluminate ($CoAl_2O_4$) (sample B) **118** under optimum conditions [a].

| Entry | Substrate | Product | Conv. (%) | Sel. (%) |
|-------|-----------|---------|-----------|----------|
| 1 | 4-OMeC$_6$H$_4$CH$_2$OH | 4-OMeC$_6$H$_4$CHO | 73.71 | 78.84 |
| 2 | 4-NO$_2$C$_6$H$_4$CH$_2$OH | 4-NO$_2$C$_6$H$_4$CHO | 58.07 | 74.89 |
| 3 | C$_6$H$_5$CH(OH)CH$_3$ | C$_6$H$_5$COCH$_3$ | 78.54 | 85.43 |
| 4 | C$_6$H$_5$CH(OH)Et | C$_6$H$_5$COEt | 71.23 | 81.87 |
| 5 | C$_6$H$_5$CH$_2$OH | C$_6$H$_5$CHO | 95.98 | 98.90 |

[a] Reaction conditions: Substrate (5 mmol), $CoAl_2O_4$ (sample B) **118** (0.5 g), $H_2O_2$ (5 mmol), $CH_3CN$ (5 mmol), 80 °C, 8 h.

The sol-gel technique via hydrolysis of tetraethyl orthosilicate (TEOS) was applied for the preparation of mono substituted Keggin type POMs, [*n*-C$_4$H$_9$)$_4$N]$_x$[PW$_{11}$MO$_{39}$]·nH$_2$O (PWM) (M = Cr, Mn, Fe, Co, Ni, and Cu) in silica matrix (PWM/SiO$_2$) and were examined in the oxidation reactions of different aromatic alcohols with aqueous 30% $H_2O_2$. Because for the preparation of benzaldehyde, PWFe/SiO$_2$ (**119**) composite showed the highest activity (80% yields) among all of catalytic systems,

further oxidation was performed by this catalyst (Table 94) [33]. Benzyl alcohol its derivatives with electron donating and withdrawing groups in *para-*, *orto-* and *meta*-positions, provided yields as excellent as primarily aromatic alcohols to the corresponding aldehydes. The **119** catalyst could also be recovered for five oxidation runs.

**Table 94.** Oxidation of alcohols with hydrogen peroxide catalyzed by $PWFe/SiO_2$ **119** [a].

| Entry | Substrate | Product | t/h | Yield (%) [b] |
|-------|-----------|---------|-----|---------------|
| 1 | $C_6H_5CH_2OH$ | $C_6H_5CHO$ | 4 | 92 |
| 2 | $2\text{-}MeC_6H_4CH_2OH$ | $2\text{-}MeC_6H_4CHO$ | 3 | 98 |
| 3 | $3\text{-}MeC_6H_4CH_2OH$ | $3\text{-}MeC_6H_4CHO$ | 3 | 100 |
| 4 | $4\text{-}MeC_6H_4CH_2OH$ | $4\text{-}MeC_6H_4CHO$ | 2.5 | 100 |
| 5 | $4\text{-}(tert\text{-Butyl})C_6H_4CH_2OH$ | $4\text{-}(tert\text{-Butyl})MeC_6H_4CHO$ | 4 | 65 |
| 6 | $2\text{-}OMeC_6H_4CH_2OH$ | $2\text{-}OMeC_6H_4CHO$ | 4.5 | 92 |
| 7 | $4\text{-}OMeC_6H_4CH_2OH$ | $4\text{-}OMeC_6H_4CHO$ | 2.5 | 100 |
| 8 | $3\text{-}BrC_6H_4CH_2OH$ | $3\text{-}BrC_6H_4CHO$ | 3.15 | 98 |
| 9 | $4\text{-}BrC_6H_4CH_2OH$ | $4\text{-}BrC_6H_4CHO$ | 3 | 99 |
| 10 | $2\text{-}BrC_6H_4CH_2OH$ | $2\text{-}BrC_6H_4CHO$ | 4.5 | 96 |
| 11 | $2\text{-}NO_2C_6H_4CH_2OH$ | $2\text{-}NO_2C_6H_4CHO$ | 3 | 60 |
| 12 | $4\text{-}NO_2C_6H_4CH_2OH$ | $4\text{-}NO_2C_6H_4CHO$ | 3 | 55 |
| 13 | $4\text{-}OHC_6H_4CH_2OH$ | $4\text{-}OHC_6H_4CHO$ | 4 | 85 |
| 14 | $2\text{-}OHC_6H_4CH_2OH$ | $2\text{-}OHC_6H_4CHO$ | 5 | 75 |
| 15 | $4\text{-}ClC_6H_4CH_2OH$ | $4\text{-}ClC_6H_4CHO$ | 3.15 | 99 |
| 16 | $2,4\text{-}ClC_6H_3CH_2OH$ | $2,4\text{-}ClC_6H_3CHO$ | 50 | 96 |
| 17 | $3\text{-}IC_6H_4CH_2OH$ | $3\text{-}IC_6H_4CHO$ | 4 | 90 |
| 18 | $C_6H_5CH(OH)Et$ | $C_6H_5COEt$ | 2.5 | 99 |

[a] Reaction condition: Substrate (1 mmol), $PWFe/SiO_2$ **119** (0.03 mmol), $H_2O_2$ (30%, 1 mL), $CH_3CN$ (3 mL), reflux, 2.5 to 50 h. [b] Yields refer to GC yields.

Oxidation of *para*-substituted phenyl methyl sulfides and benzyl alcohols with $H_2O_2$ in acetonitrile solution were examined by a series of diiron (III) complexes of 1,3-bis(2′-arylimino)isoindoline, [(Fe(L)Cl)$_2$O], and 1,4-di-(2′-aryl)aminophthalazine, [Fe$_2$(μ-OMe)$_2$(H$_2$L)Cl$_4$], 1,4-di-(4′-methyl-2′-thiazolyl)aminophthalazine, and 1,4-di-(2′-benzthiazolyl)-aminophthalazine (Table 95) [135]. The results of the [Fe$_2$(μ-O)(L$^{3,5,7}$)$_2$Cl$_2$] (**120–122**) catalysts confirmed that complex **120** provided the highest conversion because of catalyzing both oxygen-atom transfer and hydrogen-atom abstraction. In addition, among phthalazine-based complexes, the best oxidative activity was found with **124**, since benzyl alcohol gave the benzaldehyde with 100% selectivity by catalysts in all these reactions, and no other products were provided.

The oxidation of alcohols employing $H_2O_2$ as oxidant by manganese-containing catalytic system [Mn$^{IV,IV}$$_2$O$_3$(tmtacn)$_2$]$^{2+}$ (**126**)/carboxylic acid (where tmtacn = $N,N′,N″$-trimethyl-1,4,7-triazacyclononane was reported (Table 96) [136]. Co-catalyst trichloroacetic acid provided the most active catalyst system and side reactions due to side products were not observed [137,138].

**Table 95.** Results of the oxidation of benzyl alcohols catalyzed by isoindoline- ($[Fe_2(\mu\text{-O})(L^{3,5,7})_2Cl_2]$) (**120–122**) and phthalazine-based ($[Fe_2(\mu\text{-OMe})_2(H_2^{4,6,8})Cl_4]$) (**123–125**) diiron complexes [a].

| Entry | Catalyst | Substrate 4′R-PhCH$_2$OH | t/h | Conv. (%) | Product distribution (%) | |
|-------|----------|--------------------------|-----|-----------|--------------------------|---|
| | | | | | 4′R-PhCHO | 4′R-PhCO$_2$H |
| 1 | $[Fe_2(\mu\text{-O})(L^3)_2Cl_2]$ | H | 1 | 2.4 | 100 | 0 |
| | | H | 2 | 12.5 | 100 | 0 |
| 2 | $[Fe_2(\mu\text{-O})(L^5)_2Cl_2]$ | H | 1 | 0.97 | 100 | 0 |
| | | H | 2 | 0.97 | 100 | 0 |
| 3 | $[Fe_2(\mu\text{-O})(L^7)_2Cl_2]$ | H | 1 | 0.15 | 100 | 0 |
| | | H | 2 | 1.2 | 100 | 0 |
| 4 | $[Fe_2(\mu\text{-OMe})_2(H_2L^4)Cl_4]$ | H | 1 | 7 | 100 | 0 |
| | | H | 2 | 11 | 100 | 0 |
| 5 | $[Fe_2(\mu\text{-OMe})_2(H_2L^6)Cl_4]$ | H | 1 | 9.5 | 100 | 0 |
| | | H | 2 | 12 | 100 | 0 |
| | | OMe | 1 | 10 | 100 | 0 |
| | | OMe | 2 | 13 | 100 | 0 |
| | | Me | 1 | 9 | 100 | 0 |
| | | Me | 2 | 11 | 100 | 0 |
| | | Cl | 1 | 9 | 100 | 0 |
| | | Cl | 2 | 10 | 100 | 0 |
| | | NO$_2$ | 1 | 4 | 100 | 0 |
| | | NO$_2$ | 2 | 5 | 100 | 0 |
| 6 | $[Fe_2(\mu\text{-OMe})_2(H_2L^8)Cl_4]$ | | 0.17 [b] | 3 | 100 | 0 |

[a] Reaction conditions: Substrate (1 mmol), ($[Fe_2(\mu\text{-O})(L^{3,5,7})_2Cl_2]$) (**120–122**) and ($[Fe_2(\mu\text{-OMe})_2(H_2^{4,6,8})Cl_4]$) (**123–125**) (10 µmol), H$_2$O$_2$ (2.5 mmol), CH$_3$CN (5 mL), 20 °C, 1 to 2 h. [b] Complex precipitation after 10 min.

**Table 96.** Oxidation of secondary alcohols to ketones using H$_2$O$_2$ catalyzed by $[Mn^{IV,IV}_2O_3(tmtacn)](PF_6)_2$ (**126**) [a].

| Entry | Substrate | Product | Conv. (%) [b] | Yield (%) [c] |
|-------|-----------|---------|---------------|---------------|
| 1 | C$_6$H$_5$CH(OH)CH$_3$ | C$_6$H$_5$COCH$_3$ | 100 | 77 |
| 2 | C$_6$H$_5$CH(OH)Et | C$_6$H$_5$COEt | 100 | 63 |
| 3 | (C$_6$H$_5$)$_2$CHOH | (C$_6$H$_5$)$_2$CO | 100 | 93 |

[a] Reaction conditions: Substrate (3 mmol), $[Mn^{IV,IV}_2O_3(tmtacn)](PF_6)_2$ **126** (0.1 mol%), Cl$_3$CCO$_2$H (1 mol%), H$_2$O$_2$ (50%, 1.45 equiv.), CH$_3$CN ([substrate] = 1M), room temperature, 16 h. [b] Conversion was determined by $^1$H NMR spectroscopy. [c] Isolated yields.

As illustrated in Table 97, an in situ prepared catalyst based on manganese(II) salts (Mn(ClO$_4$)$_2$·6H$_2$O (**127**)), pyridine-2-carboxylic acid, and butanedione with H$_2$O$_2$ as oxidant at ambient temperatures as catalyst system provided good-to-excellent yields and conversions with high TONs (up to 10,000) [139]. In addition, secondary alcohols were converted to ketones selectively in substrates bearing multiple alcohol groups which reduced the need to introduction protecting groups prior to the oxidation and subsequent removal. In general, the results confirmed that benzyl CH oxidation proceeded in preference to aliphatic CH oxidation.

**Table 97.** Oxidation of secondary alcohols [a].

| Entry | Substrate | Product | Conv. (%) [b] | Yield (%) [c] |
|-------|-----------|---------|---------------|---------------|
| 1 | C$_6$H$_5$CH(OH)CH$_3$ | C$_6$H$_5$COCH$_3$ | 97 | 90 |
| 2 | C$_6$H$_5$CH(OH)Et | C$_6$H$_5$COEt | 92 | 77 |
| 3 | (C$_6$H$_5$)$_2$CHOH | (C$_6$H$_5$)$_2$CO | 90 | 80 |
| 4 | 1-Tetralol | 1-Tetralone | 88 | 75 |
| 5 [d] | 4-OMeC$_6$H$_4$CH(OH)CH$_3$ | 4-OMeC$_6$H$_4$COCH$_3$ | 70 | 64 |
| 6 | 4-BrC$_6$H$_4$CH(OH)CH$_3$ | 4-BrC$_6$H$_4$COCH$_3$ | 78 | 76 |

[a] Reaction conditions: Substrate (0.5 M, 1 mmol), Mn(ClO$_4$)$_2$·6H$_2$O **127** (50 μM, 0.01 mol%), pyridine-2-carboxylic acid (2.5 mM, 0.5 mol%), NaOAc (5 mM, 1 mol%), butanedione (0.25 M, 0.5 equiv.), H$_2$O$_2$ (50%, 1.5 M, 3 equiv.) in CH$_3$CN, room temperature, 12 to 16 h. [b] Conversions and yields based on the substrate, determined by $^1$H NMR spectroscopy. [c] Isolated yields, unless stated otherwise. [g] Substrate (0.25 M, 0.5 mmol).

Chemoselective oxidation of alcohols to their corresponding carbonyl compounds in short reaction times and high yields were reported by an efficient protocol including H$_2$O$_2$ as oxidant at 80 °C in acetonitrile in the presence of nanoparticles of Fe$_2$O$_3$-SiO$_2$ (**128**) (Table 98) [140]. The catalyst was successfully recycled at least five times with no substantial loss in the activity of the reused catalyst.

**Table 98.** Oxidation of various alcohols catalyzed by (9.6 wt% Fe) Fe$_2$O$_3$-SiO$_2$/Imidazole/H$_2$O$_2$ [a].

| Entry | Substrate | Product | Conv. (%) | Sel. (%) [b] | TON |
|-------|-----------|---------|-----------|--------------|-----|
| 1 [d] | C$_6$H$_5$CH$_2$OH | C$_6$H$_5$CHO | 94 | 100 | 55 |
| 2 [d] | 4-ClC$_6$H$_4$CH$_2$OH | 4-ClC$_6$H$_4$CHO | 100 | 100 | 58 |

[a] Reaction conditions: Substrate (1 mmol), Fe$_2$O$_3$-SiO$_2$ **128** (9.6 wt% Fe, 10 mg, 17.2 μmol), imidazole (40 μmol), H$_2$O$_2$ (30%, 14 mmol), CH$_3$CN (3 mL), 80 ± 1 °C, 5 h. Conversions and yields are based on the starting substrate. [b] All products were identified by comparison of their physical and spectral data with those of authentic samples or GC-MS. [c] Reaction time = 4 h.

In the presence of H$_2$O$_2$, a reusable (three times) and high stable heteropolyoxometalates [C$_7$H$_7$N(CH$_3$)$_3$]$_3$PMo$_4$O$_{16}$ (BTPM) (**140**) catalyst was catalyzed for selective oxidation of BzH with selectivity more than 99% and 92.8% of BzOH conversion (Scheme 1) [141].

$$[C_7H_7N(CH_3)_3]_3PMo_4O_{16} \xrightarrow[\text{organic solvents}]{H_2O_2} [C_7H_7N(CH_3)_3]_3PO_4\{MoO(O_2)_2\}_4$$

insoluble             soluble

**Scheme 1.** Process of the catalytic oxidation of BzOH.

Vanadium phosphorus oxide (VPO) (**129**) was introduced as an operational catalyst in the presence of hydrogen peroxide for the preparation of aldehydes and ketones from alcohol oxidations without employing any sacrificial oxidants or base at 65 to 70 °C in acetonitrile solvent (Table 99) [142]. Although both activated and non-activated alcohols were selectively and efficiently converted to the corresponding carbonyls with no producing byproducts, activated alcohols such as benzhydrol, benzyl alcohol, and 1-phenylethanol (Entries 1, 2, and 3, respectively) afforded a much higher conversion. The catalyst could also be easily reused for five successive cycle runs with a slight decrease in its efficiency.

**Table 99.** Oxidation of various alcohols over VPO **129** catalyst using $H_2O_2$ and acetonitrile [a].

| Entry | Substrate | Product | Conv. (%) | Sel. (%) | TON |
|-------|-----------|---------|-----------|----------|-----|
| 1 | $(C_6H_5)_2CHOH$ | $(C_6H_5)_2CO$ | 52 | 100 | 80 |
| 2 | $C_6H_5CH_2OH$ | $C_6H_5CHO$ | 66 | 78 [b] | 102 |
| 3 | $C_6H_5CH(OH)CH_3$ | $C_6H_5COCH_3$ | 77 | 100 | 118 |
| 4 | $2\text{-}FC_6H_4CH_2OH$ | $2\text{-}FC_6H_4CHO$ | 42 | 62 [c] | 65 |

[a] Reaction conditions: Substrate (10 mmol), VPO **129** (10 mg), $H_2O_2$ (30%, 40 mmol), $CH_3CN$ (10 mL), 65 °C, 4 h, stir, $N_2$. [b] Remaining benzoic acid. [c] Remaining fluoro benzoic acid.

An environmentally friendly protocol including hydrogen peroxide, liquid phase at atmospheric pressure over $Fe^{3+}$/montmorillonite-K10 (**130**) catalyst synthesized by the ion-exchange method at pH = 4 in acetonitrile solvent was proposed for the oxidation of various primary and secondary aromatic, as illustrated in Table 100 [143]. Oxidation of the -OH group adjacent to the benzene ring (activated alcohols) was easier such as in 1-phenylethanol (Entry 1) and benzyl alcohols (Entries 2–3) and also it showed higher conversions (86% to 95%). The catalyst was also reused for five times.

**Table 100.** Oxidation of various alcohols over $Fe^{3+}$/K10 **130** catalyst using $H_2O_2$ and acetonitrile [a].

| Entry | Substrate | Product | Conv. (%) | Sel. (%) | TON |
|-------|-----------|---------|-----------|----------|-----|
| 1 | $C_6H_5CH(OH)CH_3$ | $C_6H_5COCH_3$ | 86 | 95 [b] | 48 |
| 2 | $C_6H_5CH_2OH$ | $C_6H_5CHO$ | >95 | 32 [c] | 54 |
| 3 | $3\text{-}ClC_6H_4CH_2OH$ | $3\text{-}ClC_6H_4CHO$ | >95 | 5 [c] | 54 |
| 4 | $4\text{-}FC_6H_4CH_2OH$ | $4\text{-}FC_6H_4CHO$ | >95 | 41 [d] | 54 |

[a] Reaction conditions: Substrate (12.5 mmol), $Fe^{3+}$/K10 **130** (0.25 g), $H_2O_2$ (30%, 50 mmol), $CH_3CN$ (10 mL), 65 °C, 8 h, stir, air. [b] 4 h, remaining benzaldehyde. [c] 4 h, remaining respective benzoic acids. [d] 0.5 h, remaining fluoro benzoic acid.

### 2.3.2. Toluene Solvent

Oxo-vanadium (V=O) was immobilized on the surface of $Fe_3O_4$@PDA [$VO(PDA)@Fe_3O_4$] as a magnetic adsorbent and stabilizing agent. $VO(PDA)@Fe_3O_4$ (**131**) nanoparticles demonstrated high catalytic activity as a recyclable catalyst in the chemoselective oxidation of benzylic alcohols to aldehydes using hydrogen peroxide in high yield (Table 101) [144]. The synthesized catalyst could be reused seven times and could be easily separated.

**Table 101.** Oxidation of benzylic alcohols catalyzed by $VO(PDA)@Fe_3O_4$ **131** [a].

| Entry | Substrate | Product | t (h) | Yield (%) |
|-------|-----------|---------|-------|-----------|
| 1 | $C_6H_5CH_2OH$ | $C_6H_5CHO$ | 6 | 96 |
| 2 | $4\text{-}ClC_6H_4CH_2OH$ | $4\text{-}ClC_6H_4CHO$ | 8 | 90 |
| 3 | $4\text{-}MeC_6H_4CH_2OH$ | $4\text{-}MeC_6H_4CHO$ | 8 | 88 |
| 4 | $4\text{-}NO_2C_6H_4CH_2OH$ | $4\text{-}NO_2C_6H_4CHO$ | 12 | 90 |
| 5 | $(C_6H_5)_2CHOH$ | $(C_6H_5)_2CO$ | 8 | 92 |
| 6 | $C_6H_5CH(OH)CH_3$ | $C_6H_5COCH_3$ | 10 | 90 |

[a] Reaction conditions: Substrate (1 mmol), $VO(PDA)@Fe_3O_4$ **131** (60 mg, 1.5 mol%), $H_2O_2$ (30%, 2.5 mmol), toluene (5 mL), reflux, 6 to 12 h.

The deposition-precipitation method was applied for the preparation of a series of bimetallic copper-nickel (CuNix, x = 0.1, 0.2, 0.5, and 1) nanoparticles supported on activated carbon (AC) for assessing the effect of different ratios of Ni added into the Cu catalyst. A comparison between bimetallic Cu-Ni nanoparticles and monometallic Cu and Ni nanoparticles showed that the catalytic activity increased in the presence of the $CuNi_1$/AC (**132**) catalyst for the oxidation of benzyl alcohols to the corresponding aldehyde in a short reaction time at 80 °C (Table 102) [145].

**Table 102.** Catalytic performance of the catalysts for the oxidation of benzyl alcohol to benzaldehyde [a].

| Entry | Catalyst | Conv. (%) |
|-------|----------|-----------|
| 1 | Cu/AC | 13.8 |
| 2 | Ni/AC | 33.6 |
| 3 | $CuNi_{0.1}/AC$ | 35.1 |
| 4 | $CuNi_{0.2}/AC$ | 36.5 |
| 5 | $CuNi_{0.5}/AC$ | 37.1 |
| 6 | $CuNi_1/AC$ | 46.8 |

[a] Reaction conditions: Benzyl alcohol (2.6 mL), catalyst (0.1 g), $H_2O_2$ (30%, 3.3 mL), toluene (20 mL), 80 °C, 2 h.

Redox reaction between in situ generated zero-valent cobalt nanomaterial and $KMnO_4$ produced gram-scale synthesis of recoverable heterobimetallic Co-Mn oxide ($Co_2Mn_3O_8$) (**133**) in water was successfully applied for oxidation of benzyl, aliphatic, cinamyl, pyridine, and thiophene moiety alcohols to aldehydes/ketones, both in the presence of hydrogen peroxide, with excellent selectivity (>99%) and yield (Table 103) [146].

**Table 103.** Substrate scope for $Co_2Mn_3O_8$-promoted oxidation of a variety of alcohols in the presence of hydrogen peroxide [a].

| Entry | Substrate | Product | t/h | Conv. (%) | TOF ($h^{-1}$) |
|-------|-----------|---------|-----|-----------|----------------|
| 1 | $4\text{-}OMeC_6H_4CH_2OH$ | $4\text{-}OMeC_6H_4CHO$ | 10 | 85 | 170 [b] |
| 2 | Indole-3-carbinol | Indole-3-carboxaldehyde | 12 | 78 | 130 [b] |
| 3 | 2-Thiophenemethanol | 2-Thiophenecarboxaldehyde | 12 | 72 | 120 [b] |
| 4 | 2-Pyridinemethanol | Picolinaldehyde | 10 | 80 | 160 [b] |
| 5 | $C_6H_5CH(OH)CH_3$ | $C_6H_5COCH_3$ | 10 | 95 | 190 [c] |
| 6 | $(C_6H_5)_2CHOH$ | $(C_6H_5)_2CO$ | 10 | 85 | 170 [c] |

[a] Reaction conditions: Substrate (10 mmol), $Co_2Mn_3O_8$ **133** (0.05 mol%), $H_2O_2$ (0.7 mL), toluene (30 mL), 130 °C, 10 to 12 h. [b] Yields were calculated from GC with o-xylene as an internal standard. [c] NMR yield.

The selective oxidation of alcohols catalyzed successfully by $Fe_3O_4@mSiO_2/NH\text{-}PV_2W$ (**134**) as a novel magnetically recyclable catalyst was prepared by immobilization of divanadium-substituted Keggin phosphotungstic acid $H_5PV_2W_{10}O_{40}$ on mesoporous silica-coated $Fe_3O_4$ core-shell nanoparticles (Table 104) [147]. With an external magnetic field, the catalyst was easily separated after the completion of the reactions and without any loss of its catalytic activity, the recovered catalyst could be reused at least five times. Although the catalyst showed high catalytic activities with 98% and 99% selectivity for conversion of benzyl alcohol to benzaldehyde, the secondary benzyl alcohols were reluctantly oxidized to corresponding ketones such as 40% and 36% conversion oxidation for 1-phenylethanol, as shown in Entry 4.

**Table 104.** Selective oxidation of different alcohols catalyzed by $Fe_3O_4@SiO_2/NH-V_2PW$ (20%) **134** [a].

| Entry | Substrate | Product | Conv. (%) | Sel. (%) |
|-------|-----------|---------|-----------|----------|
| 1 | $C_6H_5CH_2OH$ | $C_6H_5CHO$ | 98 | 99 |
| 2 | $3\text{-}MeC_6H_4CH_2OH$ | $3\text{-}MeC_6H_4CHO$ | 97 | 99 |
| 3 | $4\text{-}MeC_6H_4CH_2OH$ | $4\text{-}MeC_6H_4CHO$ | 98 | 99 |
| 4 | $C_6H_5CH(OH)CH_3$ | $C_6H_5COCH_3$ | 40 | 99 |

[a] Reaction condition: Substrate (1 mM), $Fe_3O_4@SiO_2/NH-V_2PW$ **134** (0.02 g), $H_2O_2$ (2 mL), toluene (2 mL), 80 °C, 8 h.

## 2.3.3. Acetic Acid Solvent

A simple combination included NaBr (**135**), $H_2O_2$, and AcOH was overlooked among many reported catalytic systems for catalytic oxidations of alcohols [148–150]. Oxidation of secondary benzylic alcohols bearing substituents such as Cl, Br, $NO_2$, and diarylmethanol derivatives in aqueous hydrogen peroxide as terminal oxidant were explored by this system with excellent yields (Table 105) [151].

**Table 105.** Substrate scope under optimized conditions [a].

| Entry | Substrate | Product | Yield (%) |
|-------|-----------|---------|-----------|
| 1 | $C_6H_5CH(OH)n\text{-}Oct$ | $C_6H_5COn\text{-}Oct$ | 98 |
| 2 | $4\text{-}ClC_6H_4CH(OH)n\text{-}Pr$ | $4\text{-}ClC_6H_4COn\text{-}Pr$ | 93 |
| 3 | $4\text{-}BrC_6H_4CH(OH)CH_3$ | $4\text{-}BrC_6H_4COCH_3$ | >99 |
| 4 | $4\text{-}NO_2C_6H_4CH(OH)CH_3$ | $4\text{-}NO_2C_6H_4COCH_3$ | 95 |
| 5 | $3\text{-}NO_2C_6H_4CH(OH)CH_3$ | $3\text{-}NO_2C_6H_4COCH_3$ | 95 |
| 6 | 1-(Naphthalen-2-yl)propan-1-ol | 1-(Naphthalen-2-yl)propan-1-one | 91 |
| 7 | $4\text{-}OMeC_6H_4CH(OH)CH_3$ | $4\text{-}OMeC_6H_4COCH_3$ | 7 |
| 8 | $C_6H_5CH(OH)(CH_2)_2Cl$ | $C_6H_5CO(CH_2)_2Cl$ | 93 |
| 9 | $(C_6H_5)_2CHOH$ | $(C_6H_5)_2CO$ | 98 |
| 10 | 1-Naphthyl(phenyl)methanol | 1-Benzoylnaphthalene | 90 |
| 11 | 9-Fluorenol | 9-Fluorenone | 97 |

[a] Reaction conditions: Substrate (0.5 mmol), NaBr **135** (10 mol%), $H_2O_2$ (2 equiv.), $CH_3COOH$ (1 mL), 60 °C, 2 h.

Under optimized conditions, substrate having *p*-methoxypenyl group provided the oxidation product **c′** with poor yield due to formation of major product 4-methoxy-α-methylbenzyl acetate (**d**) and 4-methoxy-α-methylbenzyl hydroperoxide (**e**). Without NaBr catalyst, **d** and **e** products were largely prepared which indicated that the formation of these products occurred independently from the oxidation reaction pathway. With more experience, the best optimized condition was performed in AcOH-EtOAc (3:7, 0.25 M), giving the desired ketone **c′** in 86% isolated yield (Table 106, Entries 1–5).

**Table 106.** Optimization of conditions for substrate **c** [a].

| Entry | Solvent | Yield of c′ (%) [b] | Yield of d (%) [b] | Yield of e (%) [b] |
|-------|---------|---------------------|--------------------|--------------------|
| 1 | AcOH | 7 | 27 | 15 |
| 2 | AcOH-EtOAc (7:3) | 31 | 31 | 10 |
| 3 | AcOH-EtOAc (5:5) | 47 | 13 | 7 |
| 4 | AcOH-EtOAc (3:7) | 60 | 5 | 4 |
| 5 [c] | AcOH-EtOAc (3:7) | 86 [d] | Trace | Trace |

[a] Reaction conditions: *p*-Methoxybenzyl alcohol (0.5 mmol), NaBr **135** (10 mol%), solvent (1 mL), 60 °C for 2 h.
[b] Determined by $^1$H NMR analysis of crude material. [c] Volume of solvent used was 2 mL. [d] Isolated yield.

As illustrated in Table 107, this protocol was successfully applied for the preparation of corresponding oxidative products **f′–j′** a variety of substrates such as 1-*o*-methoxyphenyl-1-ethanol (**f**), cyclopropylphenylmethanol (**g**), and various cyclic substrates **h–j** with good yields (Table 107) [150,151].

**Table 107.** The preparation of corresponding oxidative products **f′–j′** [a].

| Entry | Substrate | Product | Yield (%) |
|-------|-----------|---------|-----------|
| 1 | 4-OMeC$_6$H$_4$CH(OH)CH$_3$ | 4-OMeC$_6$H$_4$COCH$_3$ (**c′**) | 86 |
| 2 | 2-OMeC$_6$H$_4$CH(OH)CH$_3$ | 2-OMeC$_6$H$_4$COCH$_3$ (**f′**) | 73 |
| 3 [b] | C$_6$H$_5$CH(OH)*c*-Pr | C$_6$H$_5$CH(OH)*c*-Pr (**g′**) | 61 |
| 4 | 3,3-Dimethyl-1-indanol | 3,3-Dimethyl-1-indanone (**h′**) | 70 |
| 5 | 4,4-Dimethyl-1-tetralol | 4,4-Dimethyl-1-tetralone (**i′**) | 73 |
| 6 | 2,3,4,5-Tetrahydrobenzo[*b*]oxepin-5-ol | 3,4-Dihydrobenzo[*b*]oxepin-5(2*H*)-one (**j′**) | 66 |

[a] Reaction conditions: Substrate (0.5 mmol), NaBr **135** (10 mol%), CH$_3$COOH (2 mL), 60 °C, 2 h. [b] NaBr **135** (20 mol%).

Heterogeneous catalytic oxidation of preyssler type heteropolyacid supported onto silica gel, H$_{14}$[NaP$_5$W$_{30}$O$_{110}$]/SiO$_2$ (**136**) was reported for the converting of substituted benzyl alcohols converted to the corresponding benzaldehyde derivatives, as shown in Table 108 [26]. Using 0.03 g H$_{14}$-P$_5$ supported onto silica gel in refluxing acetic acid represented the best results with excellent yields. After extended reaction times, overoxidation products such as carboxylic acids were not observed and the catalytic activity remained after three runs.

**Table 108.** Oxidation of alcohols using $H_{14}$-$P_5$ supported onto silica gel **136** [a].

$$R\text{-}C_6H_4\text{-}CH_2OH \xrightarrow[\text{CH}_3\text{COOH (5 mL), reflux, 10-30 min}]{H_{14}[NaP_5W_{30}O_{110}]/SiO_2 \ \textbf{136} \ (0.03 \ g)} R\text{-}C_6H_4\text{-}CHO$$

| Entry | Substrate | Product | t/min | Yield (%) [a] |
|---|---|---|---|---|
| 1 | $C_6H_5CH_2OH$ | $C_6H_5CHO$ | 13 | 99 |
| 2 | $2\text{-}OHC_6H_4CH_2OH$ | $2\text{-}OHC_6H_4CHO$ | 10 | 98 |
| 3 | $2\text{-}MeC_6H_4CH_2OH$ | $2\text{-}MeC_6H_4CHO$ | 12 | 99 |
| 4 | $4\text{-}BrC_6H_4CH_2OH$ | $4\text{-}BrC_6H_4CHO$ | 14 | 99 |
| 5 | $4\text{-}NO_2C_6H_4CH_2OH$ | $4\text{-}NO_2C_6H_4CHO$ | 15 | 98 |
| 6 | Furfuryl alcohol | Furfural | 30 | 97 |

[a] Reaction conditions: Substrate (1 mmol), $H_{14}[NaP_5W_{30}O_{110}]/SiO_2$ **136** (0.03 g), $CH_3COOH$ (5 mL), reflux, 10 to 30 min. [b] Yields analyzed by GC.

### 2.3.4. Dimethylacetamide Solvent

As shown in Table 109, a series of heteropolytungstates catalysts, $Q_3PW_{12}O_{40}$ (**137**), $Q_3PMo_{12}O_{40}$ (**138**), and $Q_3PMo_4O_{24}$ (**139**) ($Q^+$ represents $[C_7H_7N(CH_3)_3]^+$), were applied for the BzOH oxidation reactions by Jian [152]. Using **137** and **138** as catalyst needed no feature of the reaction-controlled phasetransfer catalysis, within the determined reaction time, the catalysts were still soluble in the solvent, and the reaction conversions were poor (38.5% and 24.6%, respectively). Applying $[C_7H_7N(CH_3)_3]_9PW_9O_{34}$ (**140**) as catalyst under optimal conditions, DMAc, $H_2O_2$/BzOH:1, 95.0% conversion of benzyl alcohol (based on $H_2O_2$) and over 99% selectivity of benzaldehyde was achieved. In addition, the catalyst could be reused and recycled three times.

**Table 109.** Catalytic oxidation of BzOH by various heteropolyoxometalates with $H_2O_2$ [a].

$$R\text{-}C_6H_4\text{-}CH_2OH \xrightarrow[\text{DMAc (10 mL), 80 °C, 0.5-3.5 h}]{\text{catalyst}/H_2O_2/\text{alcohol molar ratio} \ (1:225:250)} R\text{-}C_6H_4\text{-}CHO$$

| Entry | Catalyst | Solvent | t/h | Solubility of Catalyst | | Conv. (%) | TON (mmol)/ Catalyst (mmol) | TOF ($h^{-1}$) |
|---|---|---|---|---|---|---|---|---|
| | | | | During Reaction | After Reaction | | | |
| 1 | $Q_3PW_{12}O_{40}$ | DMAc | 3.5 | soluble | soluble | 38.5 | 87 | 25 |
| 2 | $Q_9PW_9O_{34}$ | DMAc | 0.5 | soluble | insoluble | 95 | 214 | 428 |
| 3 | $Q_3PMo_{12}O_{40}$ | $CH_3CN$ | 3.5 | soluble | soluble | 24.6 | 55 | 16 |
| 4 | $Q_3PMo_4O_{24}$ | $CH_3CN$ | 3.5 | soluble | insoluble | 92.8 | 209 | 60 |

[a] Reaction conditions: Catalyst/$H_2O_2$/alcohol molar ratio (1:225:250), DMAc (10 mL), 80 °C, 0.5 to 3.5 h.

### 2.3.5. Methanol Solvent

As presented in the Table 110, Dabiri et al. [153] reported oxidation of alcohols to the related carbonyl compounds using a vanadium Schiff base complex on nano silica catalyst (VSBC@NS) (**141**). Nano silica $NH_2$-functionalized, 2,4-dihydroxy benzaldehyde, and $VO(acac)_2$ were reacted and provided heterogeneous **141** catalysts which showed good efficiency, chemical stability, and at least five times reusability in selective oxidation of alcohols to aldehydes.

### 2.3.6. *tert*-Buthanol Solvent

Tandon and co-workers [154] reported a quasi-homogenous reaction system included using 1:1.5 *t*-butanol-$H_2O_2$, not expensive PTC tetrabutyl ammonium hydrogen sulfate (TBAHSO$_4$) as the phase

transfer and heteropolyanion $Na_2WO_4$ (**142**) as the catalyst for simple secondary alcohol oxidations to ketones at 90 °C. As shown in Table 111, 100% yield of products in 30 min confirmed the power of this catalytic system.

**Table 110.** Oxidation of alcohols [a].

| Entry | Substrate | Product | Conv. (%) | Yield (%) [b] | TON | TOF (h$^{-1}$) |
|---|---|---|---|---|---|---|
| 1 | $C_6H_5CH_2OH$ | $C_6H_5CHO$ | 98 | 95 | 4.75 | 1.19 |
| 2 | $4\text{-}MeC_6H_4CH_2OH$ | $4\text{-}MeC_6H_4CHO$ | 93 | 89 | 4.45 | 1.11 |
| 3 | $4\text{-}ClC_6H_4CH_2OH$ | $4\text{-}ClC_6H_4CHO$ | 90 | 88 | 4.4 | 1.1 |
| 4 | $4\text{-}NO_2C_6H_4CH_2OH$ | $4\text{-}NO_2C_6H_4CHO$ | 81 | 76 | 3.8 | 0.95 |
| 5 | $3\text{-}NO_2C_6H_4CH_2OH$ | $3\text{-}NO_2C_6H_4CHO$ | 84 | 81 | 4.05 | 1.01 |
| 6 | 1-Indanol | 1-Indanone | 94 | 91 | 4.55 | 1.14 |
| 7 | $C_6H_5CH(OH)CH_3$ | $C_6H_5COCH_3$ | 89 | 86 | 4.3 | 1.08 |

[a] Reaction conditions: Substrate (0.5 mmol), VSBC@NS **141** (20 mol%), $H_2O_2$ (1.2 equiv.), MeOH (2 mL), reflux, 4 h.
[b] GC yield, *n*-dodecane was used as an internal standard.

**Table 111.** Oxidation of secondary alcohols with hydrogen peroxide in *tert*-butanol [a].

| Entry | Substrate | Product | Yield (%) |
|---|---|---|---|
| 1 | 1-Naphthaleneethanol | 1-Acetonaphthone | 98 |
| 2 | $(C_6H_5)_2CHOH$ | $(C_6H_5)_2CO$ | 100 |
| 3 | $C_6H_5CH(OH)CH_3$ | $C_6H_5COCH_3$ | 100 |

[a] Reaction conditions: Substrate (1.84 g, 10 mmol), $Na_2WO_4$ **142** (164 mg, 0.5 mmol), TBAHSO$_4$ (169 mg, 0.5 mmol), $H_2O_2$ (30%, 1.72 mL, 15 mmol), *t*-BuOH (2 mL), 90 °C, 30 min.

Oxidation of benzylic alcohols also provided corresponding aldehydes or acids with good yields (Table 112) [154]. Substitution of the aromatic ring influenced the state of oxidation like *p*-methoxybenzaldehyde was produced with 87% yield from *p*-methoxybenzyl alcohol in four hours and further oxidation did not occur; whereas *p*-nitrobenzaldehyde (62%) and *p*-nitrobenzoic acid (30%) were produced from *p*-nitrobenzyl alcohol oxidation. In addition, non-benzylic primary alcohols were directly converted to their corresponding carboxylic acids (Table 112, Entry 3). The results showed that, generally, secondary alcohols were more reactive towards oxidation than primary alcohols.

2.3.7. Xylene Solvent

Hayashi, in 2012 [155], reported using 30% $H_2O_2$ in the presence of activated carbon (**143**) for the oxidation of secondary benzylic alcohols to ketones (Table 113). At first, 9-fluorenol was selected as a substrate (Table 114) and 9-fluorenone was obtained with 77% yield (100 weight% of activated carbon, *m*-xylene, 95 °C, 18 h). When ten equivalents of 30% $H_2O_2$ were used, less (4 equiv.) and more amounts (15 equiv.) of 30% $H_2O_2$ caused a decrease of yield (Table 114, Entries 1 (68%) and 3 (35%), respectively). After drying, the activated carbon was recovered and reused without deactivation.

**Table 112.** Oxidation of primary alcohols with hydrogen peroxide in *tert*-butanol.

| Entry | Substrate | Product | Yield (%) |
|-------|-----------|---------|-----------|
| 1 | $C_6H_5CH_2OH$ | $C_6H_5CHO$ [a] | 75 [g] |
| 2 | $C_6H_5CH_2OH$ | $C_6H_5CO_2H$ [b] | 100 |
| 3 | $C_6H_5CH_2OH$ | $C_6H_5CH_2CO_2H$ [b] | 84 |
| 4 | $4\text{-}OMeC_6H_4CH_2OH$ | $4\text{-}OMeC_6H_4CHO$ [c] | 87 [h] |
| 5 | $4\text{-}OMeC_6H_4CH_2OH$ | $4\text{-}OMeC_6H_4CH_2CO_2H$ [d] | 87 |
| 6 | $4\text{-}NO_2C_6H_4CH_2OH$ | $4\text{-}NO_2C_6H_4CHO$ [b,e] | 62 |
| 7 | $4\text{-}NO_2C_6H_4CH_2OH$ | $4\text{-}NO_2C_6H_4CHO$ [e,f] | 52 |

Reaction conditions: Alcohol/$H_2O_2$/$Na_2WO_4$/PTC = [a] 20:30:1:1 (8 h), [b] 20:40:1:1, [c] 20:30:1:1, [d] 20:50:1:1, [e] 18 h, [f] 100:200:1:2 (18 h). [g] Remaining 25%: benzoic acid. [h] Rest is unreacted alcohol.

**Table 113.** Oxidation of various alcohols to the corresponding carbonyl compounds [a].

| Entry | Substrate | Product | Yield (%) |
|-------|-----------|---------|-----------|
| 1 | Phenyl(2-quinolinyl)methanol | Phenyl(2-quinolinyl)methanone | 74 |
| 2 | α-Quinolyl-ethanol | 1-(2-Quinolinyl)ethanone | 67 |
| 3 | Phenyl(2-pyridyl)methanol | Phenyl 2-pyridyl ketone | 64 |
| 4 | 9-Fluorenol | 9-Fluorenone | 77 |
| 5 | $C_6H_5CH(OH)COC_6H_5$ | $(C_6H_5)_2(CO)_2$ | 91 |

[a] Reaction conditions: Substrate (1 mmol), activated carbon **143** (100 wt%), $H_2O_2$ (30%, 10 mmol), anhydrous xylene (10 mL), 95 °C, 18 h.

**Table 114.** Oxidation of 9-fluorenol [a].

| Entry | X (equiv.) | Yield (%) [b] |
|-------|-----------|---------------|
| 1 | 4 | 68 |
| 2 | 10 | 77 |
| 3 | 15 | 35 |

[a] Reaction conditions: 9-Fluorenol (1 mmol), activated carbon **143** (100 wt%), $H_2O_2$ (30%, 4, 10, 15 mmol), anhydrous xylene (10 mL), 95 °C, 18 h. [b] Isolated yield after silica gel column chromatography.

### 2.3.8. Chloroform Solvent

A solvent-controlled selective oxidation of benzyl alcohol (BnOH) by 30 wt% $H_2O_2$ with iron(III) tosylate ($Fe(OTs)_3 \cdot 6H_2O$) (**144**) was reported (Table 115) [156]. The results confirmed that using different solvents provided dissimilar products such as in chloroform, quantitative conversion to benzaldehyde (BzH) was shown, while in acetonitrile, benzoic acid was detected with high yield.

**Table 115.** Oxidation of benzyl alcohol catalyzed by $Fe(OTs)_3.6H_2O$ **144** with $H_2O_2$ [a].

| | | | | | | |
|---|---|---|---|---|---|---|
| | $C_6H_5CH_2OH$ | $\xrightarrow{\substack{Fe(OTs)_3.6H_2O\ 144\ (0.1\ mmol) \\ n(H_2O_2)/n(BnOH)=6/1 \\ CHCl_3\ (10\ g),\ 60\ ^\circ C,\ 16\ h.}}$ | $C_6H_5CHO$ | | | |

| Entry | Substrate | Product | Conv. (%) | Yield (%) | Sel. (%) | TON (mol/mol$_{cat}$) |
|---|---|---|---|---|---|---|
| 1 | $C_6H_5CH_2OH$ | $C_6H_5CHO$ | 89.3 | 85.6 | 94.1 | 85.5 |

[a] Reaction conditions: Benzyl alcohol (1.08 g, 10 mmol), $Fe(OTs)_3.6H_2O$ **144** (0.1 mmol), $H_2O_2$ (30%, molar ratio $n(H_2O_2)/n(BnOH)$ = 6/1), $CHCl_3$ (10 g), 60 °C, 16 h.

### 2.4. Oxidation of Benzylic and Heterocyclic Alcohols in Dual-Phase System

### 2.4.1. Acetonitrile-Water

Tungstate ions immobilized on the homemade periodic mesoporous organosilica with imidazolium ionic liquid framework which was denoted as $WO_4^=@PMO-IL$ (**145**) were prepared and applied with hydrogen peroxide in the oxidation of primary and secondary alcohols to the corresponding carbonyl compounds (Table 116) [157]. Various primary even pretty hindered substrate 2-nitro-benzyl alcohol were also converted to the corresponding aldehyde with excellent selectivities and good yields without any overoxidation (Table 116, Entry 5). With acceptable to excellent yield, under the same reaction conditions, secondary aromatic alcohols were also converted to the corresponding ketones (Table 116, Entries 7–9). In addition, the catalyst could be also efficiently recovered and reused in seven subsequent reaction cycles without deactivation [157].

**Table 116.** Oxidation of various alcohols with $WO^=_4@PMO-IL$ **145** catalyst [a].

| Entry | Substrate | Product | t/h | Conv. (%) [b] | Sel. (%) |
|---|---|---|---|---|---|
| 1 | $C_6H_5CH_2OH$ | $C_6H_5CHO$ | 12 | 75 | 100 |
| 2 | $4\text{-}OMeC_6H_4CH_2OH$ | $4\text{-}OMeC_6H_4CHO$ | 7.5 | 76 | 100 |
| 3 | $4\text{-}ClC_6H_4CH_2OH$ | $4\text{-}ClC_6H_4CHO$ | 11 | 82 | 100 |
| 4 | $4\text{-}NO_2C_6H_4CH_2OH$ | $4\text{-}NO_2C_6H_4CHO$ | 24 | 65 | 93 |
| 5 | $2\text{-}NO_2C_6H_4CH_2OH$ | $2\text{-}NO_2C_6H_4CHO$ | 33 | 77 | 97 |
| 6 | $3\text{-}ClC_6H_4CH_2OH$ | $3\text{-}ClC_6H_4CHO$ | 21 | 80 | 100 |
| 7 | $C_6H_5CH(OH)CH_3$ | $C_6H_5COCH_3$ | 15 | 94 | 100 |
| 8 | $C_6H_5CH(OH)Et$ | $C_6H_5COEt$ | 15 | 73 | 100 |
| 9 | $C_6H_5CH(OH)CH_2C_6H_5$ | $C_6H_5COCH_2C_6H_5$ | 16 | 42 | 100 |

[a] Reaction condition: Substrate (1 mmol), $CH_3CN:H_2O$ (1:1, 0.5 mL), $WO_4^=@PMO-IL$ **145** (1.5 mol %), $H_2O_2$ (30%, 5 equiv.), 90 °C, 7.5 to 33 h. [b] GC yield using standard addition method.

The mild alcohol oxidations with hydrogen peroxide were catalyzed by heterogeneous catalyst iron(III) tetrakis(*p*-sulfonatophenylporphyrinato)acetate supported on polyvinylpyridine and Amberlite IRA-400 (**146**). As illustrated in Table 117, benzyl alcohol and substituted benzyl alcohols were efficiently converted to their corresponding aldehydes with no significant effect of substituents on the oxidation process [158]. In addition, secondary alcohols were converted to their corresponding ketones with good yields. The catalysts could also be applied in four regeneration cycles without significant loss of their activity.

**Table 117.** Performance of catalysts in the oxidation of alcohols with $H_2O_2$ [a].

| Entry | Substrate | Product | Fe(TPPS)-Ad IRA-400 | | Fe(TPPS)-PVP | |
|---|---|---|---|---|---|---|
| | | | t/h | Yield (%) [b] | t/h | Yield (%) [b] |
| 1 | $C_6H_5CH_2OH$ | $C_6H_5CHO$ | 4 | 91 | 4 | 90 |
| 2 | $4\text{-}ClC_6H_4CH_2OH$ | $4\text{-}ClC_6H_4CHO$ | 4 | 90 | 3 | 90 |
| 3 | $4\text{-}BrC_6H_4CH_2OH$ | $4\text{-}BrC_6H_4CHO$ | 3 | 89 | 3 | 90 |
| 4 | $4\text{-}OMeC_6H_4CH_2OH$ | $4\text{-}OMeC_6H_4CHO$ | 4 | 92 | 4 | 92 |
| 5 | $2\text{-}ClC_6H_4CH_2OH$ | $2\text{-}ClC_6H_4CHO$ | 4 | 90 | 4 | 93 |
| 6 | $3\text{-}OMeC_6H_4CH_2OH$ | $3\text{-}OMeC_6H_4CHO$ | 5 | 91 | 5 | 90 |
| 7 | 1-Indanol | 1-Indanone | 8 | 76 | 7 | 74 |
| 8 | $C_6H_5CH(OH)CH_3$ | $C_6H_5COCH_3$ | 5 | 90 | 4 | 86 |

[a] Reaction conditions: Substrate (1 mmol), supported Fe(III) porphyrins **146** (20 μmol), imidazole (0.4 mmol), $H_2O_2$ (6 mmol), $CH_3CN$:$H_2O$ (1:1, 10 mL), room temperature, 3 to 8 h. [b] GC yield based on starting alcohol.

Tetrabutylammonium decatungstate (VI) (TBADT) (**147**) as catalyst was applied in the oxidation of selected alcohols with hydrogen peroxide as a green oxidant using acetonitrile/water 1,2- or dichloroethane/water as a solvent system. Microwave irradiation combined with elevated pressure in two phase system acetonitrile/water resulted in the highest conversions of substrates in the range of 80% to 100% (Table 118) [159]. TBADT also conserved its catalytic activity for two regeneration cycles.

**Table 118.** Oxidation of selected alcohols in the acetonitrile/water system [a].

| Entry | Substrate | Product | Conv. (mol%) | | Products Proportion (mol%) | |
|---|---|---|---|---|---|---|
| | | | MW [b] | MW [c] | MW [b] | MWP [c] |
| 1 | $C_6H_5CH_2OH$ | $C_6H_5CHO$ $C_6H_5CO_2H$ | 64 | 100 | 955 | 964 |
| 2 | $C_6H_5CH(OH)CH_3$ | $C_6H_5COCH_3$ | 81 | 96 | 100 | 100 |

[a] Reaction conditions: Substrate (6 mmol), TBADT **147** (1 mol%), $H_2O_2$ (30%, 18 mmol), $CH_3CN$:$H_2O$ (3 mL), 80 °C, 10 to 15 min. [b] MW, microwave heating for 30 min. [c] MWP, microwave heating under higher pressure for 15 min.

### 2.4.2. Water-Toluene

$H_2O_2$ in the presence of catalytic amounts of titanium oxide loaded $SiO_2$-based phase-boundary (**148**) was used to investigate oxidation of various hydrophobic alcohols, especially relatively bulky alcohols with their oxidation limited by well-known titanium-containing microporous materials due to their limitations of pore size (Table 119) [160,161]. Primary benzylic alcohols were efficiently converted into corresponding aldehydes. The rate of their oxidation depended not only on substituents (–Cl, –$OCH_3$, and –$CH_3$) but also on their positions (*p-*, *m-*, and *o-*) (Entries 1–6). Correspondingly, in relatively high selectivity, secondary benzylic alcohols were oxidized to corresponding ketones (Entries 7 and 8).

**Table 119.** Oxidation of various alcohols catalyzed by w/o-Ti-SiO$_2$ **148** using $H_2O_2$ as an oxidant [a].

| Entry | Substrate | Product | Yield (mmol) [b] | Sel. (%) [c] |
|-------|-----------|---------|------------------|--------------|
| 1 | $C_6H_5CH_2OH$ | $C_6H_5CHO$ | 1.29 | 97 |
| 2 | $4\text{-}ClC_6H_4CH_2OH$ | $4\text{-}ClC_6H_4CHO$ | 1.33 | 92 |
| 3 | $4\text{-}OMeC_6H_4CH_2OH$ | $4\text{-}OMeC_6H_4CHO$ | 1.20 | 74 |
| 4 | $4\text{-}MeC_6H_4CH_2OH$ | $4\text{-}MeC_6H_4CHO$ | 1.81 | 80 |
| 5 | $3\text{-}MeC_6H_4CH_2OH$ | $3\text{-}MeC_6H_4CHO$ | 1.48 | 92 |
| 6 | $2\text{-}MeC_6H_4CH_2OH$ | $2\text{-}MeC_6H_4CHO$ | 1.15 | 99 |
| 7 | $C_6H_5CH(OH)CH_3$ | $C_6H_5COCH_3$ | 0.85 | 100 |
| 8 | $(C_6H_5)_2CHOH$ | $(C_6H_5)_2CO$ | 0.91 | 100 |

[a] Reaction conditions: Substrate (8 mmol), w/o-Ti-SiO$_2$ **148** (67 mg), $H_2O_2$ (30%, 1.8 mL), toluene (3 mL), 60 °C, 16 h. [b] Yield of the produced aldehyde determined by GC using internal standard technique. [c] Selectivity for aldehyde production is defined as the percentage of the aldehyde yield to the total amount of products.

### 2.4.3. Dichloromethane-Water

A tetrameric DABCO–bromine complex (**149**) as a novel active bromine complex was introduced for the fast oxidation of allylic, benzylic, primary, and secondary alcohols to carbonyl compounds with excellent yields and without overoxidation (Table 120) [27].

**Table 120.** Oxidation of alcohols to aldehydes and ketones with the DABCO-bromine complex **149** [a].

| Entry | Substrate | Product | t/h | Yield (%) |
|-------|-----------|---------|-----|-----------|
| 1 | $C_6H_5CH_2OH$ | $C_6H_5CHO$ | 4 | 97 |
| 2 | $4\text{-}ClC_6H_4CH_2OH$ | $4\text{-}ClC_6H_4CHO$ | 5 | 98 |
| 3 | $4\text{-}OMeC_6H_4CH_2OH$ | $4\text{-}OMeC_6H_4CHO$ | 4 | 97 |
| 4 | $4\text{-}NO_2C_6H_4CH_2OH$ | $4\text{-}NO_2C_6H_4CHO$ | 5 | 87 |
| 5 | $2\text{-}MeC_6H_4CH_2OH$ | $2\text{-}MeC_6H_4CHO$ | 3 | 95 |
| 6 | $4\text{-}MeC_6H_4CH_2OH$ | $4\text{-}MeC_6H_4CHO$ | 3 | 96 |

[a] Reaction conditions: Substrate (1 mmol), DABCO–Br$_2$ **149** (0.262 g, 0.166 mol), CH$_2$Cl$_2$:H$_2$O (6 mL:2 mL), room temperature, 3 to 5 h.

## 3. Conclusions

In this review, an attempt has been made to summarize some efficient catalytic systems for the selective oxidation of primary and secondary alcoholic OH groups to carbonyl or carboxyl functional groups, using $H_2O_2$ as the green oxidant. Different chemical environments such as an aqueous media, a solvent-free system, and various organic solvent and dual-phase systems in the present of hydrogen peroxide oxidant indicated good selectivity and in most cases no overoxidation to acid were observed. Moreover, in many articles, electron-donating were compared with electron-withdrawing functional groups in benzyl alcohol derivative oxidations. In most literature, it was found that there were different yields and conversions of primary and secondary benzylic alcohols with electron-donating and -withdrawing functional groups. It was also shown that electron-donating or -withdrawing functional groups had the same effect on the oxidation results of many researches. In light of the above, it is hoped that this review can assist as a valuable critical overview of the area and the contributions help in further research in this field.

**Author Contributions:** This work has been completed in continuation of M.M.H., N.G. and E.H. activities in the field of oxidation of alcohols. All authors have read and agreed to the published version of the manuscript.

**Funding:** This work supported by five year granted research chair by INSF as well as Alzahra Universoty Research Council.

**Acknowledgments:** The authors are thankful to Alzahra University research Council for partial financial assistance. MMH also appreciates financial support from Iran National Science Foundation (INSF), granted via an individual granted research chair.

**Conflicts of Interest:** The authors declare no conflict of interest.

## Abbreviations

| | |
|---|---|
| PCC | pyridinium chlorochromate |
| BzOH | benzyl alcohol |
| BzH | benzaldehyde |
| RSM | response surface methodology |
| PTA | phosphotungstic acid |
| APTES | 3-aminopropyltriethoxysilane |
| TLC | thin layer chromatography |
| MNPs | magnetic nanoparticles |
| EPI | epichlorohydrin |
| MEG | ethylene glycol |
| EPR | electron paramagnetic resonance spectroscopy |
| TEA | trimethylamine |
| TBHP | *tert*-butyl hydroperoxide |
| Fe-BTC | iron(III)-benzenetricarboxylate |
| BET | Brunauer–Emmett–Teller |
| TEDA | 1,4-diazabicyclo[2.2.2]octane |
| OKR | oxidative kinetic resolution |
| POM | Polyoxometalate |
| AC | activated carbon |
| ABTSC | 1-(4-dimethylaminobenzylidene) thiosemicarbazide |
| TCTS | 1-(2-pyridincarboxyl-idene) thiosemicarbazide |
| TEOS | tetraethyl orthosilicate |
| TBA | tetra-*n*-butylammonium |
| PVP | polyvinylpolypyrrolidone |
| VPO | vanadium phosphorus oxide |
| DMAc | dimethylacetamide |
| UHP | urea-hydrogen peroxide |
| PEGDME250 | polyethylene glycol dimethyl ether 250 |
| TBAHSO$_4$ | tetrabutyl ammonium hydrogen sulfate |

PTC         phase transfer catalyst
TBADT      tetrabutylammonium decatungstate
DABCO     1,4-diazabicyclo[2.2.2] octane

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
