# Peer review of "Hydrogen Peroxide as a Green Oxidant for the Selective Catalytic Oxidation of Benzylic and Heterocyclic Alcohols in Different Media: An Overview"

_chemistry, doi:10.3390/chemistry2010010_

Round 1
Reviewer 1 Report
In general, the paper was well done and have described a plethora of reactions where hydrogen peroxide was the oxidant with different catalysts.
I have some suggestions that may improve the review.
The results of yield and conversion should be written as 47 % and not as 46.5%. Change it ai all the Tables. The substrate and product should be shown as a structural formula and not as molecular formulae (see Table 3) In general, there are Tables too longer. The authors should to try to short these Tables, It is not necessary to do an extense list if along with the text the authors did not discuss the results. A number of six entries its is enough, in my opinion. The authors have presented all the results as Tables. There is no curves or graphics. For instance, Table 49 can be easily converted to kinetic curves Conversion versus time Table 55. I don't believe in those conversions; 42.95 % should be considered 43 %. Finally, this review should be strongly shortened.
Author Response
Dear Prof. Yu Wang
Thank you very much for given opportunity to allow us revising our manuscript entitled “Hydrogen peroxide as a green agent for the selective catalytic oxidation of benzylic and heterocyclic alcohols in different chemical media: a review” in “Chemistry journal”.
We carefully read the instructions and corrected the errors in the article accordingly and highlighted them. Regarding the first editor's question, we are not really allowed to change the percentage of conversions because our review is a report of the work of other authors that we have referenced.
Looking forward to hearing from, you in the near future.
If you need any assistance from this side please do not hesitate to contact me
.
Best Regards
Majid M Heravi
Department of Chemistry
Alzahra University
Tehran, Iran
Reviewer 2 Report
Authors reported a review about water peroxide mediated oxidation of alcohols. Despite thei intentions, the work is poorly written and it is hard to understend in several sections. A deep language revision is needed before considered it for pubblicaiton.
Some of the issues detected are reported here:
line 86-87: This sentence does not have any sense
line 96: Systems not system
line 105: On not in
line 111-112: This sentence does not have any sense. Rewritten more clearly.
line 116: The improving no the better.
line 152: Conversion not conversions
line 157: Incipient not SIncipient
line 166: If you used di- use the same notaiton everytime
line 170: Define TON
line 204: Several instead of diverse
line 212-213: This sentence does not have any sense
line 220-222: This sentence does not have any sense
line 230:By instead of on
line 246: Although,
line 251: Reported instead of indicated
line 271: was obtained
lines 276-278: This sentence does not have any sense
line 283 What does it means "aromatic secondary"?
line 283-285: This sentence does not have any sense
line 287-290: Unclear sentence. Rewritten it.
line 312: Were not was
line 319: were converted
line 394-396:This sentence does not have any sense
line 401: IN table 32 footnotes are to bigger. Incorporated them into the discussion
line 453: Citation is missed
line 520: Catpital C was missed
line 577: Treated not treatment
line 716: Where is the experimental section?
line 916: What doe it mean "ACHTUNGTRENUNG"?
line 924: You should define TON in the line 170 not here
line 1129: were instes of was
line 1884-1886: This is a review work submitted to a reputated peer-reviewed journal. It MUST be comprehensive. If you think that it is not you should not submitt or withdraw it.
Furthermore, a rieplilogative table with all catalyst cited and relative references will be very usefully and should be added.
Additionally, all the data are mainly reported without further comments by the authors. Authors must add a section where they discuss in detailed about mechanisms of the catalysts and solvent effects comapring them.
In the present form this manuscript does not fit the requirements for publication in Chemistry. Due the interest of the topic and the amount of work required by a review article production, i suggest to reconsider it after major revisions.
Author Response
Response to Reviewer 2 Comments
line 86-87: This sentence does not have any sense
The sentence was corrected.
line 96: Systems not system
It was corrected.
line 105: On not in
It was corrected.
line 111-112: This sentence does not have any sense. Rewritten more clearly.
We checked the sentence and corrected it.
line 116: The improving no the better.
It was corrected.
line 152: Conversion not conversions
It was corrected.
line 157: Incipient not SIncipient
It was corrected.
line 166: If you used di- use the same notaiton everytime
It was corrected.
line 170: Define TON
TON's definition was written.
line 204: Several instead of diverse
It was corrected.
line 212-213: This sentence does not have any sense
We checked the sentence and corrected it.
line 220-222: This sentence does not have any sense
The sentence was corrected.
line 230: By instead of on
It was corrected.
line 246: Although,
It was corrected.
line 251: Reported instead of indicated
It was corrected.
line 271: was obtained
It was corrected.
lines 276-278: This sentence does not have any sense
The sentence was corrected.
line 283 What does it means "aromatic secondary"?
The aromatic secondary changed to the aromatic secondary alcohols.
line 283-285: This sentence does not have any sense
The sentence was conceptually reviewed and refined.
line 287-290: Unclear sentence. Rewritten it.
The sentence was corrected.
line 312: Were not was
It was corrected.
line 319: were converted
It was corrected.
line 394-396: This sentence does not have any sense
The sentence was conceptually reviewed and refined.
line 401: IN table 32 footnotes are to bigger. Incorporated them into the discussion
It was corrected.
line 453: Citation is missed
It was corrected.
line 520: Catpital C was missed
It was corrected.
line 577: Treated not treatment
It was corrected.
line 716: Where is the experimental section?
Table 71 refers to the experimental section.
line 916: What doe it mean "ACHTUNGTRENUNG"?
ACHTUNGTRENUNG was deleted.
line 924: You should define TON in the line 170 not here
It was corrected.
line 1129: were instes of was
It was corrected.
line 1884-1886: This is a review work submitted to a reputated peer-reviewed journal. It MUST be comprehensive. If you think that it is not you should not submitt or withdraw it.
It was corrected.
Round 2
Reviewer 2 Report
Authors fixed teh amin typos and improve some of the points of discussion.
Nonetheless, i do not consider appropriate "We also would like to apologize
to anyone who finds this description of her or his work inadequate or whose work has been omitted."
As, i said befor, this is a review work submitted to a reputated peer-reviewed journal. It MUST be comprehensive. If you think that it is not you should not submitt or withdraw it.
Author Response
Dear Prof. Yu Wang
Thank you very much for given opportunity to allow us revising our manuscript entitled “Hydrogen peroxide as a green agent for the selective catalytic oxidation of benzylic and heterocyclic alcohols in different chemical media: a review” in “Chemistry journal”.
We carefully read the instructions and removed the unrelated and extra articles.
Looking forward to hearing from, you in the near future.
If you need any assistance from this side, please do not hesitate to contact me
.
Best Regards
Majid M Heravi
Department of Chemistry
Alzahra University
Tehran, Iran